# LION: A BIDIRECTIONAL FRAMEWORK THAT TRAINS LIKE A TRANSFORMER AND INFERS LIKE AN RNN

## ABSTRACT

We introduce LION, a novel sequence-to-sequence framework that unifies the bidirectionality and parallelized training of Transformers with the fast inference of recurrent neural networks. LION is built upon a mathematical formulation where full kernelized attention with a learnable mask is efficiently computed using a bidirectional selective recurrent model, matching the effectiveness of softmax-based attention with constant-time inference. Our framework naturally accounts for spatial and temporal relationships within input sequences, reducing reliance on heuristic positional embeddings and facilitating straightforward scalability in context length and resolution. Using our framework and inspired by the recent state-space models, we propose three main running examples LION-LIT , LION-RETNET , and LION-S , a transformer with selective mask and recurrent inference. Numerical evaluations on tasks such as language modeling, the Long-Range Arena, and image classification show that LION framework achieves performance on par with state-of-the-art models while delivering fast training and inference efficiency.

## 1 INTRODUCTION

The new material added for the rebuttal is in blue.

Transformers (Vaswani et al., 2017) have become a key pillar for large language models (LLMs), with different variants tailored to specific applications (Brown et al., 2020; Achiam et al., 2023; Team et al., 2023). A key distinction lies in the use of causal (autoregressive) Transformers for language modeling (Kojima et al., 2022; Dubey et al., 2024), which predict tokens sequentially based on prior context. In contrast, bidirectional Transformers are central to large vision-language models (Liu et al., 2023; Zhu et al., 2024a; Wu et al., 2024), such as the Vision Transformer (ViT), which encodes image data in models like CLIP (Radford et al., 2021) and acts as the decoder in diffusion models for image generation (Ho et al., 2020).

Despite the success of autoregressive Transformers, they face significant resource challenges, particularly in the need to store key and value information, known as KV-cache (Pope et al., 2023), during inference. This leads to increased memory consumption, especially when processing long sequences in stark contrast to earlier recurrent neural networks (RNNs) (Elman, 1990), which has long offered a more memory-efficient alternative by maintaining a hidden state.

To address the resource bottlenecks in Transformers, Linear Transformer (Katharopoulos et al., 2020) has been proposed, expressing attention as a linear dot-product of kernel feature maps. This allows Transformers to be reformulated as RNNs, enabling the processing of longer sequences with reduced memory demands. Given the popularity of bidirectional Transformers across various fields and the efficiency of RNNs, a natural question arises:

*Is a bidirectional Transformer actually a bidirectional RNN?*

In this paper, we answer this question affirmatively. Indeed, we demonstrate that applying two linear attention mechanisms simply in opposite directions and then summing them does not recover the original bidirectional Transformer (*cf.*, Observation 3.1). Instead, we propose a novel design, LION, that allows the bidirectional Transformer to be expressed as a bidirectional RNN. Our framework retains the advantages of parallel training found in Transformers, offering bidirectionality in inference

while addressing the memory issues inherent in traditional Transformer models. A schematic of the proposed framework LION is visualized in Figure 1.

Besides the popularity of Transformers and their variants, state space models (SSMs) have emerged as another family of architecture for sequence modeling due to their efficient inference capabilities (Gu et al., 2022; Smith et al., 2023; Gu et al., 2020). The representative works Mamba (Gu & Dao, 2024) and Mamba-2 (Dao & Gu, 2024) have also demonstrated strong performance in language modeling. Building on our bidirectional Transformer theory, LION framework combines the expressive power of bidirectional Transformers with the selective mechanism of Mamba, further enhancing the model's capability to process long sequences while maintaining computational efficiency. Through this approach, we aim to provide a scalable and efficient solution for tasks that demand both long-range dependency modeling and dense information processing. Overall, our main contributions can be summarized as follows:

- We propose a theoretical framework LION (Theorem 3.3), which expresses bidirectional Transformers as bidirectional RNNs, enabling efficient inference for long sequences while benefiting from well-established Transformer training (*cf.*, Table 1).

- Our theoretical framework offers the foundations to transform a wide class of autoregressive recurrent models (*cf.*, Appendix B) into their bidirectional counterparts.

- We propose three main running examples of our framework, inspired by prior work, namely:

  1. LION-LIT : Scaled attention without masking, a bidirectional extension of Linear Transformer Katharopoulos et al. (2020).

  2. LION-RETNET : Fixed masked scaled attention with scalar and learnable state parameter $\gamma$, an extension of RETNET Sun et al. (2023) into the bidirectional setting.

  3. LION-S : Selective masked scaled attention with input-dependent mask $\lambda_i$, inspired by the selectivity of Mamba-2 Dao & Gu (2024).

- Through extensive experiments in the Long Range Arena, Vision Tasks, and Masked Language Modeling, we have demonstrated the capabilities of the LION framework and the models built upon it, as outlined above.

Due to the space constraints, a detailed overview of related work is deferred to Appendix B. Section 2 in the sequel provides the necessary preliminaries on attention, state space model, and linear recurrent network. Section 3 then explains our framework LION, and mathematically grounds our concrete contributions. Section 4 describes how to build LION-S by introducing selectivity via discretization of continuous state-space models, which is then followed by numerical evidence in Section 5 and the conclusions in Section 6.

## 2 PRELIMINARIES AND BACKGROUND

**Notation.** Matrices (vectors) are symbolized by uppercase (lowercase) boldface letters, e.g., $\mathbf{Y}$ and $\mathbf{y}$. The Hadamard product is denoted by $\odot$ and $*$ signifies the scalar product.

**Attention.** Attention have been a cornerstone of foundation models for several years (Vaswani et al., 2017; Kojima et al., 2022). Given a data sequence $\mathbf{x}_1, \mathbf{x}_2, \ldots, \mathbf{x}_L$, a single-head softmax-attention uses a softmax function to define the attention weights:

$$(\mathbf{q}_i, \mathbf{k}_i, \mathbf{v}_i) = (\mathbf{W_q}\mathbf{x}_i, \mathbf{W_k}\mathbf{x}_i, \mathbf{W_v}\mathbf{x}_i), \qquad \mathbf{y}_i = \sum_{j=1}^{i} \frac{\exp(\mathbf{q}_i^\top \mathbf{k}_j)}{\sum_{p=1}^{i} \exp(\mathbf{q}_i^\top \mathbf{k}_p)} \mathbf{v}_j, \qquad (1)$$

where $\mathbf{q}_i, \mathbf{k}_i, \mathbf{v}_i, \mathbf{y}_i \in \mathbb{R}^d$ and the weights $\mathbf{W_q}, \mathbf{W_k}, \mathbf{W_v} \in \mathbb{R}^{d \times d}$ with $d$ being the projection dimension. With $\mathbf{Q} := [\mathbf{q}_1, \ldots, \mathbf{q}_L]^\top, \mathbf{K} := [\mathbf{k}_1, \ldots, \mathbf{k}_L]^\top, \mathbf{V} := [\mathbf{v}_1, \ldots, \mathbf{v}_L]^\top \in \mathbb{R}^{L \times d}$, we can then express the attention as the following matrix form: $\mathbf{Y} = \text{softmax}\left(\mathbf{Q}\mathbf{K}^\top\right)\mathbf{V}$. Such matrix form is crucial for parallelized training over the sequence length. In contrast, (1) is used during inference for generating or processing tokens. However, for autoregressive transformers (Kojima et al., 2022), employing (1) requires storing the previous $L$ tokens to attend to the latest token during inference. This approach is less efficient than RNNs, where only the state is stored regardless of the previous sequence (*cf.*, Orvieto et al. (2023)).

Attention can be generalized via a kernel function $\kappa : \mathbb{R}^d \times \mathbb{R}^d \to \mathbb{R}$ (Tsai et al., 2019) as

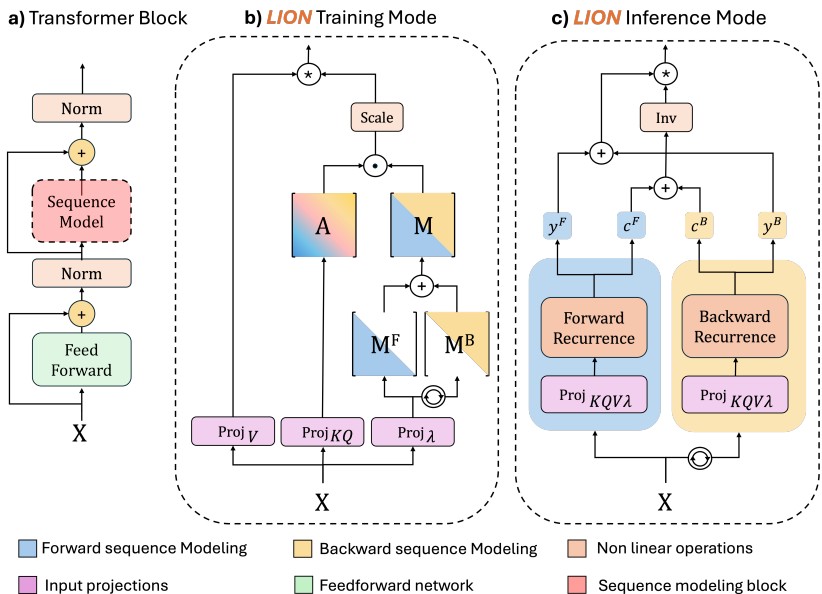

Figure 1: (*Left*) standard Transformer block. (*Middle*) training mode of LION with the bidirectional Transformer. (*Right*) inference mode of LION with the bidirectional RNN. Norm refers to Layer normalization, Proj is the projection operation to calculate $\mathbf{Q}, \mathbf{K}, \mathbf{V}$ and $\boldsymbol{\lambda}$ values, Scale is the scaling operation in Eq. (4), Inv is the inversion operation, $\mathbf{A}$ is the linear attention matrix, $\mathbf{A} = \mathbf{Q}\mathbf{K}^T$, $\mathbf{M}^{F/B}$ are forward/backward recurrence masks, $\mathbf{y}^{F/B}$ are forward/backward outputs and $c^{F/B}$ are forward/backward are the scaling coefficients. For further definitions of the architectural elements in LION, please refer to Sections 2 and 3.

Table 1: *Summary of training and inference strategies.* $\rightleftarrows$ represents bidirectionality of the method. Complexity indicates the computational and memory requirements during inference for processing $L$ tokens and $d$ is the model dimension. LION (Theorem **3.3**) is designed to parallelize training using masked attention while employing recurrence during inference, specifically for bidirectional sequence modeling. ▢ denotes the adaptation of auto regressive recurrent models with LION to *truly* exploit bidirectionality and attention, such as Linear Transformer as LION-LIT. ▢ represents injecting selectivity into attention while inferring with bidirectional recurrence as well as benefiting from the transformer training system pipeline, such as LION-S.

| Train Strategy | Inference Strategy | Method Instantiations | Train sequential operations | Complexity | Inference Memory | $\rightleftarrows$ |
|---|---|---|---|---|---|---|
| Recurrence | Recurrence | LSTM, GRU | $\mathcal{O}(L)$ | $\mathcal{O}(Ld)$ | $\mathcal{O}(d)$ | ✗ |
| Recurrence | Recurrence | ELMO | $\mathcal{O}(L)$ | $\mathcal{O}(Ld)$ | $\mathcal{O}(Ld)$ | ✓ |
| Attention | Attention | Transformer, Vit, BERT | $\mathcal{O}(1)$ | $\mathcal{O}(L^2d^2)$ | $\mathcal{O}(L^2d^2)$ | ✓ |
| Causal Attention | KV Cache | GPT-x, Llama | $\mathcal{O}(1)$ | $\mathcal{O}(L^2d^2)$ | $\mathcal{O}(Ld^2)$ | ✗ |
| Causal Attention | Recurrence | LinearTrans, RetNet | $\mathcal{O}(1)$ | $\mathcal{O}(Ld^2)$ | $\mathcal{O}(d^2)$ | ✗ |
| Parallel Scan | Recurrence | Mamba, Mamba-2, S5 | $\mathcal{O}(1)$ | $\mathcal{O}(Ld)$ | $\mathcal{O}(d)$ | ✗ |
| Parallel Scan | Recurrence | Vim | $\mathcal{O}(1)$ | $\mathcal{O}(Ld^2)$ | $\mathcal{O}(Ld)$ | ✓ |
| Attention | LION (**3.3**) | LION-LIT | $\mathcal{O}(1)$ | $\mathcal{O}(Ld^2)$ | $\mathcal{O}(Ld)$ | ✓ |
| Attention | LION (**3.3**) | LION-RETNET | $\mathcal{O}(1)$ | $\mathcal{O}(Ld^2)$ | $\mathcal{O}(Ld)$ | ✓ |
| Attention | LION (**3.3**) | LION-S | $\mathcal{O}(1)$ | $\mathcal{O}(Ld^2)$ | $\mathcal{O}(Ld)$ | ✓ |

$$\mathbf{y}_i = \sum_{j=1}^{i} \frac{\kappa(\mathbf{q}_i, \mathbf{k}_j)}{\sum_{p=1}^{i} \kappa(\mathbf{q}_i, \mathbf{k}_p)} \mathbf{v}_j . \tag{2}$$

Katharopoulos et al. (2020) introduces Linear Attention which replaces the exponential kernel $\kappa(\mathbf{q}_i, \mathbf{k}_j) = \exp(\mathbf{q}_i^\top \mathbf{k}_j)$ with feature map function $\phi(\mathbf{q}_i)^\top \phi(\mathbf{k}_j)$ where $\phi(\cdot) : \mathbb{R}^d \to \mathbb{R}^n$ maps to a higher-dimensional space. For simplicity of notation, we use $\mathbf{q}_i := \phi(\mathbf{W_q}\mathbf{x}_i)$ and similarly for $\mathbf{k}_i := \phi(\mathbf{W_k}\mathbf{x}_i)$ in the sequel. This approach enables the transformer to be framed as an RNN with

linear recurrence [1], as shown in (4). This formulation eliminates the need to store previous tokens during inference, while still maintaining a parallelized form for training.

**State Space Models.** Inspired by continuous-time systems, state space models (SSMs) have emerged as alternative sequence models. These models project tokens into a state space representation, and learn the discretized parameters ($\bar{\mathbf{A}}$, $\bar{\mathbf{B}}$, and $\bar{\mathbf{C}}$) of the continuous SSM ($\mathbf{A}_{(t)}$, $\mathbf{B}_{(t)}$, and $\mathbf{C}_{(t)}$) (Gu et al., 2022; Smith et al., 2023; Gu et al., 2020). Recent SSMs designed for language modeling, such as Mamba (Gu & Dao, 2024), use input-dependent matrices $\bar{\mathbf{A}}_i$, $\bar{\mathbf{B}}_i$, and $\bar{\mathbf{C}}_i$, showing strong performance and competitiveness with Transformers. Recently, Mamba-2 (Dao & Gu, 2024) has demonstrated a strong connection between transformers and SSMs through the theory of State Space Duality (where $\bar{\mathbf{C}}_i$, $\bar{\mathbf{B}}_i$ can be considered as $\mathbf{q}_i, \mathbf{k}_j$).

**Linear Recurrent Models.** The Transformer and SSMs motivate examining all these architectures through the lens of the linear recurrent models (Yang et al., 2024):

**STATE SPACE MODEL**

CONTINUOUS

$$\mathbf{S}'_{(t)} = \mathbf{A}_{(t)}\mathbf{S}_{(t)} + \mathbf{B}_{(t)}\mathbf{x}_{(t)}, \quad (3a)$$

$$\mathbf{y}_{(t)} = \mathbf{C}_{(t)}\mathbf{S}_{(t)} \quad (3b)$$

DISCRETE

$$\mathbf{S}_i = \bar{\mathbf{A}}_i\mathbf{S}_{i-1} + \bar{\mathbf{B}}_i\mathbf{x}_i, \quad (3c)$$

$$\mathbf{y}_i = \mathbf{C}_i\mathbf{S}_i \quad (3d)$$

**LINEAR ATTENTION**

$$\mathbf{S}_i = \mathbf{S}_{i-1} + \mathbf{k}_i\mathbf{v}_i^\top, \quad (4a)$$

$$\mathbf{z}_i = \mathbf{z}_{i-1} + \mathbf{k}_i, \quad (4b)$$

$$\text{SCALED} : \mathbf{y}_i = \frac{\mathbf{q}_i^\top\mathbf{S}_i}{\mathbf{q}_i^\top\mathbf{z}_i} \quad (4c)$$

$$\text{NON-SCALED} : \mathbf{y}_i = \mathbf{q}_i^\top\mathbf{S}_i \quad (4d)$$

**LINEAR RECURRENT MODEL**

$$\mathbf{S}_i = \mathbf{\Lambda}_i \star \mathbf{S}_{i-1} + \boldsymbol{\gamma}_i \bullet \mathbf{k}_i\mathbf{v}_i^\top, \quad (5a)$$

$$\mathbf{z}_i = \alpha_i \bullet \mathbf{z}_{i-1} + \beta_i \bullet \mathbf{k}_i, \quad (5b)$$

$$\text{SCALED} : \mathbf{y}_i = \frac{\mathbf{q}_i^\top\mathbf{S}_i}{\mathbf{q}_i^\top\mathbf{z}_i} \quad (5c)$$

$$\text{NON-SCALED} : \mathbf{y}_i = \mathbf{q}_i^\top\mathbf{S}_i \quad (5d)$$

where $\mathbf{S}_i \in \mathbb{R}^{d\times d}$ and $\mathbf{z}_i \in \mathbb{R}^d$ are the hidden state matrix and the vector used for scaling.

Linear recurrent models provide another general framework for sequence modeling in addition to standard transformers (Vaswani et al., 2017; Kojima et al., 2022). These models however introduce four additional parameters, $\mathbf{\Lambda}_i, \boldsymbol{\gamma}_i, \beta_i, \alpha_i$, along with their corresponding operation functions $\bullet$ and $\star$; please refer to Table 5 for detailed choices for different architectures.

When chosen wisely (e.g., using the *HIPPO* theory (Gu et al., 2020)), these parameters can significantly enhance the model's ability to capture long-range dependencies within sequences (Gu et al., 2022; Gu & Dao, 2024; Yang et al., 2024). For linear recurrent models, efficient training is achieved either by employing a form similar to $\mathbf{Y} = \text{softmax}\left(\mathbf{Q}\mathbf{K}^\top\right)\mathbf{V}$ or by using techniques like parallel scan (Blelloch, 1990), as utilized by many SSMs (e.g., Mamba, S5). Table 1 summarizes the training strategy and inference complexity of different sequence models.

## 3 LION: EXPANDING FULL ATTENTION TO BIDIRECTIONAL RNN

We first develop the theoretical foundation for extending the autoregressive case to the bidirectional setting with equivalence to the scaled attention. We introduce LION, a bidirectional sequence-to-sequence framework equivalent to attention that benefits from attention parallelization during training and achieves fast linear recurrence during inference.

**Observation 3.1.** First, we observe that the combination of the forward and backward recurrences of the linear recurrent model cannot yield the attention. Consider the following bidirectional recurrence equations:

$$\boxed{\textbf{a1) } \mathbf{S}_i^{F/B} = \mathbf{S}_{i-1}^{F/B} + \mathbf{k}_i\mathbf{v}_i^\top, \ \mathbf{z}_i^{F/B} = \mathbf{z}_{i-1}^{F/B} + \mathbf{k}_i, \ \mathbf{y}_i^{F/B} = \frac{\mathbf{q}_i^\top\mathbf{S}_i^{F/B}}{\mathbf{q}_i^\top\mathbf{z}_i^{F/B}}.} \neq \boxed{\textbf{a2) } \mathbf{Y} = \text{SCALE}(\mathbf{Q}\mathbf{K}^\top)\mathbf{V}} \quad (6)$$

$$\boxed{\textbf{b1) } \mathbf{S}_i^{F/B} = \lambda_i\mathbf{S}_{i-1}^{F/B} + \mathbf{k}_i\mathbf{v}_i^\top, \ \mathbf{z}_i^{F/B} = \lambda_i\mathbf{z}_{i-1}^{F/B} + \mathbf{k}_i, \ \mathbf{y}_i^{F/B} = \frac{\mathbf{q}_i^\top\mathbf{S}_i^{F/B}}{\mathbf{q}_i^\top\mathbf{z}_i^{F/B}}.} \neq \boxed{\textbf{b2) } \mathbf{Y} = \text{SCALE}(\mathbf{Q}\mathbf{K}^\top \odot \mathbf{M})\mathbf{V}} \quad (7)$$

$F/B$ indicates that the same recurrent model is applied in both forward and backward recurrence directions, and $\text{SCALE}(\cdot)$ denotes the scaling of the attention matrix across its rows ($\text{SCALE}(\mathbf{A})_{ij} = \mathbf{A}_{ij}/\sum_{j=1}^L \mathbf{A}_{ij}$). Note that in Eqs. (6) and (7) and the following content, when doing backward recurrence, the subscript of $\mathbf{S}, \mathbf{z}, \mathbf{y}, \mathbf{q}, \mathbf{k}, \mathbf{v}$ should be flipped by the rule of $i := L - i + 1$. The final output is the addition of the forward and the backward recurrences, i.e., $\mathbf{y}_i = \mathbf{y}_i^F + \mathbf{y}_i^B, \forall i \in [L]$.

---

[1] However, softmax based attention due to applying non-linearity into the attention formulation can not be linearized in this form

Figure 2: *Differences between attention and the addition of two linear recurrent models.* **a1**) Addition of two linear transformers, **a2**) Attention with scaling, **b1**) Addition of two linear recurrent models, **b2**) Masked attention with scaling. The red text highlights the differences between attention and the summed recurrent models. We use blue for the causal (forward recurrence), yellow for the non-causal (backward recurrence), and pink for the diagonal part of the attention.

While **a1** and **a2** in Eq. (6) represents the attention without the mask, **b1** and **b2** in Eq. (7) corresponds to masked attention. Moreover, $\lambda_i$ corresponds to the scalar version of $\mathbf{\Lambda}_i$ in Eq. (5a).

We show in Figure 2 that this recurrence does not equal the attention matrix, regardless of whether scaling is applied before (6) or after (7) the mask, as the naive addition of two linear recurrent models for forward and backward recurrences fails to produce an attention matrix (more details of proofs are at Appendix C.1). These key differences can be described as follows: $(i)$ The diagonal elements representing attention for each token appear in both recurrences, leading to twice the attention score for a token and itself compared to others. $(ii)$ Causal (forward recurrence) and non-causal (backward recurrence) attention scores are scaled individually, resulting in tokens not being properly scaled relative to the keys of other tokens in the sequence, unlike attention shown in Figure 2, parts **a2** and **b2**.

We precede our main result with a proposition from Sun et al. (2023), which states that an autoregressive transformer can be expressed as a linear recurrent model:

**Proposition 3.2.** *Considering the following forward recurrence:*

$$\mathbf{S}_i^F = \lambda_i \mathbf{S}_{i-1}^F + \mathbf{k}_i \mathbf{v}_i^\top, \quad \mathbf{z}_i^F = \lambda_i \mathbf{z}_{i-1}^F + \mathbf{k}_i, \quad \mathbf{y}_i = \frac{\mathbf{q}_i^\top \mathbf{S}_i^F}{\mathbf{q}_i^\top \mathbf{z}_i^F}. \tag{8}$$

*The vectorized output takes the following form:*

$$\mathbf{Y} = \left(\text{SCALE}(\mathbf{Q}\mathbf{K}^\top \odot \mathbf{M}^C)\right)\mathbf{V}, \quad \mathbf{M}_{ij}^C = \begin{cases} \Pi_{k=i}^{j+1}\lambda_k, & i \geq j; \\ 0, & i < j, \end{cases} \tag{9}$$

*with $\mathbf{M}^C$ being the selective causal mask.*

Our goal is to derive a bidirectional linear recurrence for attention with scaling and $(\text{SCALE}(\mathbf{Q}\mathbf{K}^\top \odot \mathbf{M}))$, as this framework is more generalized and can be adapted to various linear recurrent models (more detail on different variation like scaling prior to masking $\text{SCALE}(\mathbf{Q}\mathbf{K}^\top) \odot \mathbf{M}$ are provided at Appendix C.1). Motivated by (9) and the observation of how the attention matrix is divided into causal and non-causal components, we begin our method by splitting the attention matrix and the mask into upper and lower triangular parts.

$$\mathbf{Y} = \text{SCALE}\left(\underbrace{\begin{pmatrix} \mathbf{q}_1^\top \mathbf{k}_1 & \mathbf{q}_1^\top \mathbf{k}_2 & \cdots & \mathbf{q}_1^\top \mathbf{k}_L \\ \mathbf{q}_2^\top \mathbf{k}_1 & \mathbf{q}_2^\top \mathbf{k}_2 & \cdots & \mathbf{q}_2^\top \mathbf{k}_L \\ \vdots & \vdots & \ddots & \vdots \\ \mathbf{q}_L^\top \mathbf{k}_1 & \mathbf{q}_L^\top \mathbf{k}_2 & \cdots & \mathbf{q}_L^\top \mathbf{k}_L \end{pmatrix}}_{\mathbf{A} = \mathbf{Q}\mathbf{K}^\top} \odot \underbrace{\begin{pmatrix} 1 & \lambda_2 & \lambda_2\lambda_3 & \cdots & \lambda_2\cdots\lambda_L \\ \lambda_1 & 1 & \lambda_3 & \cdots & \lambda_3\cdots\lambda_L \\ \lambda_1\lambda_2 & \lambda_2 & 1 & \cdots & \lambda_4\cdots\lambda_L \\ \vdots & \vdots & \vdots & \ddots & \vdots \\ \lambda_{L-1}\cdots\lambda_1 & \lambda_{L-1}\cdots\lambda_2 & \lambda_{L-1}\cdots\lambda_3 & \cdots & 1 \end{pmatrix}}_{\mathbf{M}}\right)\begin{pmatrix} \mathbf{v}_1^\top \\ \mathbf{v}_2^\top \\ \mathbf{v}_3^\top \\ \vdots \\ \mathbf{v}_L^\top \end{pmatrix}, \tag{10}$$

where we use yellow for upper triangular elements, blue for lower triangular elements, and pink for the diagonal elements of the attention matrix and the mask. By splitting (10) into upper and lower triangular forms, we obtain the following:

$$\left(\begin{array}{cccc} \mathbf{q}_1^\top \mathbf{k}_1 & \mathbf{q}_1^\top \mathbf{k}_2 & \cdots & \mathbf{q}_1^\top \mathbf{k}_L \\ \mathbf{q}_2^\top \mathbf{k}_1 & \mathbf{q}_2^\top \mathbf{k}_2 & \cdots & \mathbf{q}_2^\top \mathbf{k}_L \\ \vdots & \vdots & \ddots & \vdots \\ \mathbf{q}_L^\top \mathbf{k}_1 & \mathbf{q}_L^\top \mathbf{k}_2 & \cdots & \mathbf{q}_L^\top \mathbf{k}_L \end{array}\right) = \underbrace{\left(\begin{array}{cccc} \frac{1}{2}\mathbf{q}_1^\top \mathbf{k}_1 & & & \\ \mathbf{q}_2^\top \mathbf{k}_1 & \frac{1}{2}\mathbf{q}_2^\top \mathbf{k}_2 & & \\ \vdots & \vdots & \ddots & \\ \mathbf{q}_L^\top \mathbf{k}_1 & \mathbf{q}_L^\top \mathbf{k}_2 & \cdots & \frac{1}{2}\mathbf{q}_L^\top \mathbf{k}_L \end{array}\right)}_{\mathbf{A}^F} + \underbrace{\left(\begin{array}{cccc} \frac{1}{2}\mathbf{q}_1^\top \mathbf{k}_1 & \mathbf{q}_1^\top \mathbf{k}_2 & \cdots & \mathbf{q}_1^\top \mathbf{k}_L \\ & \frac{1}{2}\mathbf{q}_2^\top \mathbf{k}_2 & \cdots & \mathbf{q}_2^\top \mathbf{k}_L \\ & & \ddots & \\ & & & \frac{1}{2}\mathbf{q}_L^\top \mathbf{k}_L \end{array}\right)}_{\mathbf{A}^B} \quad (11)$$

$$\underbrace{\left(\begin{array}{ccccc} 1 & \lambda_2 & \lambda_2\lambda_3 & \cdots & \lambda_2\cdots\lambda_L \\ \lambda_1 & 1 & \lambda_3 & \cdots & \lambda_3\cdots\lambda_L \\ \lambda_1\lambda_2 & \lambda_2 & 1 & \cdots & \lambda_4\cdots\lambda_L \\ \vdots & \vdots & \vdots & \ddots & \vdots \\ \lambda_{L-1}\cdots\lambda_1 & \lambda_{L-1}\cdots\lambda_2 & \lambda_{L-1}\cdots\lambda_3 & \cdots & 1 \end{array}\right)}_{\mathbf{M}} = \underbrace{\left(\begin{array}{ccccc} 1 & & & & \\ \lambda_1 & 1 & & & \\ \lambda_1\lambda_2 & \lambda_2 & 1 & & \\ \vdots & & & \ddots & \\ \lambda_{L-1}\cdots\lambda_1 & \lambda_{L-1}\cdots\lambda_2 & \lambda_{L-1}\cdots\lambda_3 & \cdots & 1 \end{array}\right)}_{\mathbf{M}^F} + \underbrace{\left(\begin{array}{ccccc} 1 & \lambda_2 & \lambda_2\lambda_3 & \cdots & \lambda_2\cdots\lambda_L \\ & 1 & \lambda_3 & \cdots & \lambda_3\cdots\lambda_L \\ & & 1 & \cdots & \lambda_4\cdots\lambda_L \\ & & & \ddots & \\ & & & & 1 \end{array}\right)}_{\mathbf{M}^B} - \mathbf{I} \quad (12)$$

As in (11) and (12), the attention matrix and mask are split into lower ($\mathbf{A}^F, \mathbf{M}^F$) and upper triangular ($\mathbf{A}^B, \mathbf{M}^B$) matrices. The scaling operator divides each row of the attention matrix to its summed value, and hence equals to a diagonal matrix $\mathbf{C}^{-1}$ multiplied by the attention:

$$\mathbf{Y} = \big(\text{SCALE}(\mathbf{Q}\mathbf{K}^\top \odot \mathbf{M})\big)\mathbf{V} = (\mathbf{C}^{-1}(\mathbf{Q}\mathbf{K}^\top \odot \mathbf{M}))\mathbf{V}, \quad \mathbf{C}_i = \mathbf{q}_i^\top \sum_{j=1}^{L} \mathbf{M}_{ij}\mathbf{k}_j. \quad (13)$$

Decomposing $\mathbf{C}$ into causal and non-causal parts as $\mathbf{C}_i = \mathbf{q}_i^\top \sum_{j=1}^{i} \mathbf{M}_{ij}\mathbf{k}_j + \mathbf{q}_i^\top \sum_{j=i}^{L} \mathbf{M}_{ij}\mathbf{k}_j - \mathbf{q}_i^\top \mathbf{k}_i$, we can similarly split the scaling matrix into two parts as follows:

$$\mathbf{C}_i = \underbrace{\mathbf{q}_i^\top \sum_{j=1}^{i} \mathbf{M}_{ij}\mathbf{k}_j - \frac{1}{2}\mathbf{q}_i^\top \mathbf{k}_i}_{\mathbf{C}_i^F} + \underbrace{\mathbf{q}_i^\top \sum_{j=i}^{L} \mathbf{M}_{ij}\mathbf{k}_j - \frac{1}{2}\mathbf{q}_i^\top \mathbf{k}_i}_{\mathbf{C}_i^B} \quad (14)$$

Therefore, matrix $\mathbf{C}$ can be decomposed into $\mathbf{C} = \mathbf{C}^F + \mathbf{C}^B$. Since we have $\mathbf{A} = \mathbf{A}^F + \mathbf{A}^B$ and $\mathbf{M} = \mathbf{M}^F + \mathbf{M}^B - \mathbf{I}$, we can proceed to rewrite the output of the scaled, masked attention as

$$\begin{aligned} \mathbf{Y} &= \big(\text{SCALE}(\mathbf{Q}\mathbf{K}^\top \odot \mathbf{M})\big)\mathbf{V} = (\mathbf{C}^{-1}(\mathbf{Q}\mathbf{K}^\top \odot \mathbf{M}))\mathbf{V} \\ &= \big(\mathbf{C}^F + \mathbf{C}^B\big)^{-1}\big((\mathbf{A}^F + \mathbf{A}^B) \odot (\mathbf{M}^F + \mathbf{M}^B - \mathbf{I})\big)\mathbf{V} \qquad (15) \\ &= (\mathbf{C}^F + \mathbf{C}^B)^{-1}\big(\mathbf{A}^F \odot \mathbf{M}^F + \mathbf{A}^F \odot \mathbf{M}^B + \mathbf{A}^B \odot \mathbf{M}^F + \mathbf{A}^B \odot \mathbf{M}^B - \mathbf{A}^F \odot \mathbf{I} - \mathbf{A}^B \odot \mathbf{I}\big)\mathbf{V}. \end{aligned}$$

Since the forward and backward recurrence matrices ($\mathbf{A}^F, \mathbf{A}^B$ for attention and $\mathbf{M}^F, \mathbf{M}^B$ for mask) only share the diagonal with each other, and the diagonal of both forward and backward recurrence masks consists entirely of ones, we can simplify the above equation as follows:

$$\mathbf{Y} = (\mathbf{C}^F + \mathbf{C}^B)^{-1}\big(\mathbf{A}^F \odot \mathbf{M}^F + \underbrace{\mathbf{A}^F \odot \mathbf{M}^B}_{\mathbf{A}^F \odot \mathbf{I}} + \underbrace{\mathbf{A}^B \odot \mathbf{M}^F}_{\mathbf{A}^B \odot \mathbf{I}} + \mathbf{A}^B \odot \mathbf{M}^B - \mathbf{A}^F \odot \mathbf{I} - \mathbf{A}^B \odot \mathbf{I}\big)\mathbf{V}$$

$$= (\mathbf{C}^F + \mathbf{C}^B)^{-1}\big(\underbrace{(\mathbf{A}^F \odot \mathbf{M}^F)\mathbf{V}}_{\text{FORWARD}} + \underbrace{(\mathbf{A}^B \odot \mathbf{M}^B)\mathbf{V}}_{\text{BACKWARD}}\big). \quad (16)$$

As seen from Proposition 3.2, the FORWARD part above can be expressed as a linear recurrence. We now demonstrate that the BACKWARD recurrence term can also be represented by the same recurrence in reverse. We re-write the equation (16) by flipping the vector $\mathbf{V}$ as:

$$\underbrace{\left(\begin{array}{cccc} \frac{1}{2}\frac{\mathbf{q}_L^\top \mathbf{k}_L}{\mathbf{q}_L^\top \mathbf{z}_L} & & & \\ \frac{\mathbf{q}_{L-1}^\top \mathbf{k}_L}{\mathbf{q}_2^\top \mathbf{z}_L} & \frac{1}{2}\frac{\mathbf{q}_{L-1}^\top \mathbf{k}_{L-1}}{\mathbf{q}_2^\top \mathbf{z}_L} & & \\ \vdots & \vdots & \ddots & \\ \frac{\mathbf{q}_1^\top \mathbf{k}_L}{\mathbf{q}_1^\top \mathbf{z}_L} & \frac{\mathbf{q}_1^\top \mathbf{k}_{L-1}}{\mathbf{q}_1^\top \mathbf{z}_L} & \cdots & \frac{1}{2}\frac{\mathbf{q}_1^\top \mathbf{k}_1}{\mathbf{q}_1^\top \mathbf{z}_L} \end{array}\right)}_{F(\mathbf{A}^B)} \odot \underbrace{\left(\begin{array}{ccccc} 1 & & & & \\ \lambda_L & 1 & & & \\ \lambda_L\lambda_{L-1} & \lambda_{L-1} & 1 & & \\ \vdots & \vdots & \vdots & \ddots & \\ \lambda_L\cdots\lambda_2 & \lambda_L\cdots\lambda_3 & \lambda_L\cdots\lambda_4 & \cdots & 1 \end{array}\right)}_{F(\mathbf{M}^B)} \left(\begin{array}{c} \mathbf{v}_L^\top \\ \mathbf{v}_{L-1}^\top \\ \mathbf{v}_{L-2}^\top \\ \vdots \\ \mathbf{v}_1^\top \end{array}\right) \quad (17)$$

The equations above are the exact representations for the forward pass, as shown in (16), but with the tokens in reverse order. The matrices $\mathbf{A}^B$ and $\mathbf{M}^B$ are also modified to match the final flipped output

using flipped input values $\mathbf{V}$ using functions $F(\mathbf{X}) = \mathbf{J}_L \mathbf{X} \mathbf{J}_L$ and $\text{FLIP}(\mathbf{X}) = \mathbf{J}_L \mathbf{X}$, where $\mathbf{J}_L$ is an $L$-dimensional exchange matrix, as detailed in Appendix C.4. Thus, the outputs of the forward and backward recurrences can be expressed as follows:

$$\mathbf{Y} = (\mathbf{C}^F + \mathbf{C}^B)^{-1}(\ \mathbf{Y}^F\ +\ \mathbf{Y}^B\ ), \text{where} \tag{18}$$

$$\mathbf{Y}^F = (\mathbf{A}^F \odot \mathbf{M}^F)\mathbf{V}, \quad \mathbf{Y}^B = (\mathbf{A}^B \odot \mathbf{M}^B)\mathbf{V} = \text{FLIP}\Big(\big(F(\mathbf{A}^B) \odot F(\mathbf{M}^B)\big)\text{FLIP}(\mathbf{V})\Big). \tag{19}$$

**Theorem 3.3.** *(*LION*) Since* (18) *is the vectorized form of the recurrence presented in* (3.2)*, we can therefore express the equivalent recurrence for the scaled attention as follows:*

$$\mathbf{S}_i^{F/B} = \lambda_i \mathbf{S}_{i-1}^{F/B} + \mathbf{k}_i \mathbf{v}_i^\top, \quad (20) \qquad \mathbf{y}_i^{F/B} = \mathbf{q}_i^\top \mathbf{S}_i^{F/B} - \frac{1}{2}\mathbf{q}_i^\top \mathbf{k}_i \mathbf{v}_i, \quad (23)$$

$$\mathbf{z}_i^{F/B} = \lambda_i \mathbf{z}_{i-1}^{F/B} + \mathbf{k}_i, \quad (21) \qquad \text{OUTPUT: } \mathbf{y}_i = \frac{\mathbf{y}_i^F + \mathbf{y}_i^B}{c_i^F + c_i^B} \quad (24) \quad = \quad \boxed{\mathbf{Y} = \text{SCALE}(\mathbf{Q}\mathbf{K}^\top \odot \mathbf{M})\mathbf{V} \quad (25)}$$

$$c_i^{F/B} = \mathbf{q}_i^\top \mathbf{z}_i^{F/B} - \frac{1}{2}\mathbf{q}_i^\top \mathbf{k}_i, \quad (22)$$

*The terms* $\frac{1}{2}\mathbf{q}_i^\top \mathbf{k}_i \mathbf{v}_i$ *and* $\frac{1}{2}\mathbf{q}_i^\top \mathbf{k}_i$ *are subtracted because the diagonal of the attention in the forward and backward recurrences is half of the other attention scores. This recurrence is equivalent to scaled and masked attention, represented as* $\mathbf{Y} = \big(\text{SCALE}(\mathbf{Q}\mathbf{K}^\top \odot \mathbf{M})\big)\mathbf{V}$.

Many recurrent models trained with attention in autoregressive tasks can be generalized within this framework as an example by simply fixing the $\lambda_i = 1$ we can have a bidirectional version of Linear Transformer (Katharopoulos et al., 2020) which we refer to as LION-LIT (*cf.*, Appendix C.5, where we adapt various causal recurrent models to the bidirectional setting). Since the forward and backward recurrences operate independently, they can process the sequence in parallel, requiring only $L$ time points for $L$ tokens, similar to autoregressive models. For any token, both outputs $\mathbf{y}_i^F$ and $\mathbf{y}_i^B$ along with the scaling parameters $c_i^F$ and $c_i^B$ are extracted, allowing the final output to be stored directly in the same memory cell, only requiring $L$ memory units, akin to autoregressive models, as shown in Appendix B.5. Additionally, it is important to note that by saving the states $c_i^{F/B}$ and $\mathbf{y}_i^{F/B}$, the memory required scales linearly with the model dimension, as the first state is scalar and the second is a vector, leading to a $\mathcal{O}(Ld)$ memory requirement. In contrast, if we were to naively store the matrix-valued hidden states for each token, $\mathbf{S}_i^{F/B}$, this would result in a $\mathcal{O}(Ld^2)$ memory requirement, which grows quadratically with $d$.

## 4 LION-S: SELECTIVITY INSPIRED FROM CONTINUOUS SYSTEMS

This section outlines the selectivity for the bidirectional recurrent model and proposes LION-S. As shown in Dao & Gu (2024), transformers can be represented as SSMs through a state-space duality, where the parameters $\mathbf{C}_i$ and $\mathbf{B}_i$ in the SSM correspond to $\mathbf{q}_i$ and $\mathbf{k}_i$. However, this connection was established in the discrete domain. In our work, we explore the transformer recurrence with scaling in the continuous domain before discretizing it, which leads to the recurrence parameter $\lambda_i$. By considering the transformer recurrence in the continuous domain and applying zero-order hold discretization (Kalman, 1960), we obtain

CONTINUOUS                                                    DISCRETE

$$\mathbf{S}'_{(t)} = \mathbf{S}_{(t)} + \mathbf{k}_{(t)}\mathbf{v}_{(t)}^\top, \tag{26a} \qquad \mathbf{S}_i = e^{a_i}\mathbf{S}_{i-1} + (e^{a_i} - 1)\mathbf{k}_i\mathbf{v}_i^\top, \tag{27a}$$

$$\mathbf{z}'_{(t)} = \mathbf{z}_{(t)} + \mathbf{k}_{(t)}, \ \mathbf{y}_{(t)} = \frac{\mathbf{q}_{(t)}^\top \mathbf{S}_{(t)}}{\mathbf{q}_{(t)}^\top \mathbf{z}_{(t)}} \tag{26b} \qquad \mathbf{z}_i = e^{a_i}\mathbf{z}_{i-1} + (e^{a_i} - 1)\mathbf{k}_i, \ \mathbf{y}_i = \frac{\mathbf{q}_i^\top \mathbf{S}_i}{\mathbf{q}_i^\top \mathbf{z}_i}. \tag{27b}$$

This leads to the parameter $\lambda_i$ to have an exponential form proven at appendix B.6 $\lambda_i = e^{a_i}$, resulting in the mask $\mathbf{M}$ defined as follows:

$$\mathbf{D}_{ij} = \begin{cases} \sum_{k=i}^{j+1} a_k & \text{if } i > j \\ \sum_{k=i+1}^{j} a_k & \text{if } i < j\ , \\ 0 & \text{if } i = j \end{cases} \qquad \mathbf{M} = \exp(\mathbf{D}), \tag{28}$$

where $\exp(\cdot)$ is applied element-wise and $\mathbf{M}$ can be learned with trainable $a_i$ or with selectivity as $a_i = \log(\sigma(\mathbf{w}_a^\top \mathbf{x}_i + b))$, where $\sigma$ is the sigmoid function. The parameter $a_i$ can also be treated as

Table 2: *Performance on Long Range Arena Tasks.* For each column (dataset), the best and the second best results are highlighted with **bold** and underline respectively. Note that the MEGA architecture has roughly $10\times$ the number of parameters as the other architectures.

| Category | Model (input length) | ListOps 2048 | Text 4096 | Retrieval 4000 | Image 1024 | Pathfinder 1024 | PathX 16K | Avg. |
|---|---|---|---|---|---|---|---|---|
| Transformer | Transformer | 36.37 | 64.27 | 57.46 | 42.44 | 71.40 | ✗ | 54.39 |
| | MEGA ($\mathcal{O}(L^2)$) | **63.14** | **90.43** | 91.25 | **90.44** | 96.01 | 97.98 | **88.21** |
| | MEGA-chunk ($\mathcal{O}(L)$) | 58.76 | 90.19 | 90.97 | 85.80 | 94.41 | 93.81 | 85.66 |
| SSM | DSS | 57.60 | 76.60 | 87.60 | 85.80 | 84.10 | 85.00 | 79.45 |
| | S4 (original) | 58.35 | 86.82 | 89.46 | 88.19 | 93.06 | 96.30 | 85.36 |
| | S5 (v1) | 61.00 | 86.51 | 88.26 | 86.14 | 87.57 | 85.25 | 82.46 |
| | S5 (v2) | 62.15 | 89.31 | **91.40** | 88.00 | 95.33 | **98.58** | 87.46 |
| | Mamba | 38.02 | 82.98 | 72.14 | 69.82 | 69.26 | 67.32 | 66.59 |
| | Mamba (From Beck et al. (2024)) | 32.5 | N/A | 90.2 | 68.9 | **99.2** | N/A | N/A |
| RNN | LRU | 60.2 | 89.4 | 89.9 | 89.0 | 95.1 | 94.2 | 86.3 |
| | xLSTM | 41.1 | N/A | 90.6 | 69.5 | 91.9 | N/A | N/A |
| Transformer as Linear Recurrent Model | Local Att. | 15.82 | 52.98 | 53.39 | 41.46 | 66.63 | ✗ | 46.06 |
| | Sparse Transformer | 17.07 | 63.58 | 59.59 | 44.24 | 71.71 | ✗ | 51.24 |
| | Longformer | 35.63 | 62.85 | 56.89 | 42.22 | 69.71 | ✗ | 53.46 |
| | Linformer | 16.13 | 65.90 | 53.09 | 42.34 | 75.30 | ✗ | 50.55 |
| | Reformer | 37.27 | 56.10 | 53.40 | 38.07 | 68.50 | ✗ | 50.67 |
| | Sinkhorn Trans. | 33.67 | 61.20 | 53.83 | 41.23 | 67.45 | ✗ | 51.48 |
| | BigBird | 36.05 | 64.02 | 59.29 | 40.83 | 74.87 | ✗ | 55.01 |
| | Linear Trans. | 16.13 | 65.90 | 53.09 | 42.34 | 75.30 | ✗ | 50.55 |
| | Performer | 18.01 | 65.40 | 53.82 | 42.77 | 77.05 | ✗ | 51.41 |
| | FNet | 35.33 | 65.11 | 59.61 | 38.67 | 77.80 | ✗ | 55.30 |
| | Nyströmformer | 37.15 | 65.52 | 79.56 | 41.58 | 70.94 | ✗ | 58.95 |
| | Luna-256 | 37.25 | 64.57 | 79.29 | 47.38 | 77.72 | ✗ | 61.24 |
| | H-Transformer-1D | 49.53 | 78.69 | 63.99 | 46.05 | 68.78 | ✗ | 61.41 |
| | LION-LIT | 16.78 | 65.21 | 54.00 | 43.29 | 72.78 | ✗ | 50.41 |
| | LION-S | 62.25 | 88.10 | 90.35 | 86.14 | 91.30 | 97.99 | 86.07 |

a vector, allowing it to be multiplied with the Hadamard product on the state $\mathbf{S}_i$, as discussed in C.7. Adding the selective parameter $a_i$ into the LION framework introduces LION-S a bidirectional selective transformer with recurrence inference. Importantly, due to the use of the recurrence parameter $a_i$, LION-S *does not require any additional positional encoding, enabling it to extrapolate beyond the context length or resolution during inference.*

In addition to its connection to continuous systems, the matrix $\mathbf{D}$ can be computed using a prefix sum algorithm (Blelloch, 1990), allowing for the summation of $a_i$ values in $\mathcal{O}(\log(L))$ time, after which it can be exponentiated to derive the mask $\mathbf{M}$. Note that, as the same parameter $e^{a_i} - 1$ has appeared in (27a) and (27b), we can consider this term as a part of $\mathbf{k}_i$.

## 5 EXPERIMENTS

This section illustrates the performance of LION-LIT and -S on well-established benchmarks: Long Range Arena, masked language modelling, and image classification. Note that thanks to Theorem 3.3, LION-S benefits from the parallelization capabilities built for masked attention during training. We similarly achieve efficient inference through the bidirectional recurrence as also illustrated by Figure 1. Due to the use of $a_i$ from (28), LION-S does not require positional encodings and can extrapolate beyond context length during inference, which we will also demonstrate below.

### 5.1 LONG RANGE ARENA

We assess the performance of LION-S on the Long Range Arena (LRA) (Tay et al., 2020b), a well-established benchmark for efficient transformers. As shown in Table 2, LION-S is the only Transformer employing recurrent inference to achieve an impressive $86.07\%$ on the LRA dataset and is capable of tackling the challenging Path-X problem, where other linear recurrent models shows clear limitations. Our results indicate that LION-S achieves performance comparable to SSMs, which are renowned for their capabilities to capture long-range interactions within data and excel in the LRA task. Furthermore, among transformers, MEGA (Ma et al., 2022) is the only one with roughly ten times the parameter count that demonstrates performance comparable to LION-S. An extensive discussion on the choice of non-linearity, scaling, and dimensions of parameters is presented in Appendix D.2 and D.3. For more information on the LRA benchmarks, see Appendix D.1.

Table 3: *C4 Masked Language Modelling and GLUE results.* For each column (dataset), the best and the second best results for each model size are highlighted with **bold** and underline respectively.

| Model | MLM Acc. | MNLI | RTE | QQP | QNLI | SST2 | STSB | MRPC | COLA | Avg. |
|---|---|---|---|---|---|---|---|---|---|---|
| BERTLARGE | **69.88** | **85.68** | **67.44** | **89.90** | **91.89** | **93.04** | **88.63** | **90.89** | 56.14 | **82.95** |
| LION-LIT LARGE | 67.11 | 83.73 | 57.18 | 89.85 | 89.93 | 91.86 | 88.02 | 90.18 | 55.36 | 80.76 |
| LION-RETNET | 68.64 | 83.82 | 60.72 | 89.72 | 89.79 | 92.93 | 87.29 | 89.66 | 56.83 | 81.34 |
| LION-S LARGE | 69.16 | 84.38 | 57.69 | 89.57 | 90.30 | 92.93 | 87.68 | 90.57 | **59.54** | 81.58 |

## 5.2 MASKED LANGUAGE MODELLING

We assess BERT, LION-S, and a Linear Attention variant of BERT combined with our bidirectional approach (LION-LIT) using the Masked Language Modeling (MLM) task, which is ideally suited for bidirectional models (Devlin et al., 2019; Liu et al., 2019). Our approach involves initially pre-training the models on the C4 dataset (Dodge et al., 2021), followed by fine-tuning and evaluating their downstream performance on the GLUE benchmark (Wang et al., 2018). Both the pre-training and fine-tuning phases employ the M2 hyperparameters (Fu et al., 2023), except for the LARGE models, where learning rates of $2 \cdot 10^{-4}$ and $10^{-5}$ for pretraining and finetuning were employed for stability based on our results in Appendix D.6. For additional experimental details and results with smaller scaled models, we refer to Appendix D.5 and Appendix D.4 respectively.

In Table 3, without extensive tuning, the LION models perform closely follow BERT in both the MLM pretraining task and the GLUE finetuning tasks. However, when we test the models beyond the context length used in training, LION greatly retains or even improves the MLM accuracy in comparison to the BERT baseline, see Section 5.4.

## 5.3 IMAGE CLASSIFICATION

The image classification is an important task for bidirectional models, where Vision Transform-ers (Radford et al., 2021) perform well. We analyze the performance of the LION-S architecture and compare it against Vision Transformer (ViT-T), Linear Transformer in the bidirectional format (LION-LIT) (*c.f* Appendix C.5) , Hydra (Hwang et al., 2024) and LION-S with improved masking locality referenced as LION-S (v2) (details in Appendix C.8) on the CIFAR-10, CIFAR-100 (Krizhevsky, 2009) and ImageNet-1K (Russakovsky et al., 2015) datasets not only in terms of accuracy but also memory used during inference.

During training, we leverage the ViT-Tiny/16 (5.5M) architecture and adapt it for each baseline accordingly. For each dataset, the resolution is fixed to $224 \times 224$, and the models are pre-trained from scratch. We use the original ViT pre-training recipe and only adjust the base learning rate and warmup for our models. Extensive tuning of the pre-training recipe could further improve current results but is outside the scope of this work. For complete details on the training hyperparameters please refer to Appendix D.8. For LION-LIT, we build on the Katharopoulos et al. (2020) approach but use Theorem 3.3 for bidirectionality. In other words, following the implementation, we removed the softmax, added the nonlinearity $\phi(x) = \text{elu}(x) + 1$, and scaling. For Hydra we consider the original hyperparameters and we modify the number of layers to match the parameter size of ViT-T.

Table 4 presents the Top-1 accuracy of models on each dataset. While LION-LIT and LION-S have the same complexity, LION-S architecture significantly outperforms the LION-LIT model in each task.

Table 4: *Image classification task results.* We present the Top-1 accuracy on the validation data. LION-S shows competitive performance against ViT models. * indicates that results are directly copied from paper Zhu et al. (2024b), where the authors are training under a different setup (e.g., with data-augmentation).

| Model | CIFAR-10 | CIFAR-100 | ImageNet |
|---|---|---|---|
| ViT-T | 92.84 | 77.33 | 70.23 |
| HYDRA-T | **96.11** | 77.70 | 69.60 |
| Vim-T* | N/A | N/A | **76.1** |
| LION-LIT | 90.05 | 73.61 | 62.69 |
| LION-RETNET | 93.78 | 75.66 | 67.31 |
| LION-S | 93.25 | 77.56 | 67.95 |
| LION-S (v2) | 94.77 | **80.07** | 69.22 |

| Model | ImageNet |
|---|---|
| ViT-S | 72.19 |
| Vim-S* | **80.3** |
| LION-LIT SMALL | 69.62 |
| LION-RETNET | 71.96 |
| LION-SSMALL | 70.86 |
| LION-S (v2)SMALL | 73.44 |

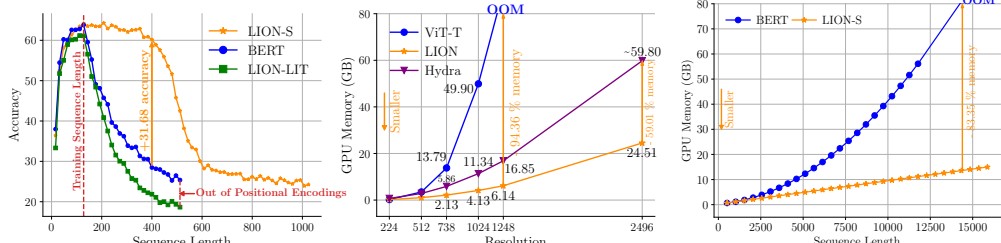

Figure 3: (*Left*) the accuracy of models on MLM task as the sequence length increases. LION-S maintains its high performance for sequences 3.5 times longer than the training sequence length. (*Center/Right*) Inference GPU memory consumption of models at different resolutions for the image classification task on Imagenet with a batch size of 128 (*Center*) and MLM on C4 with a batch size of 4 (*Right*). While ViT goes out of memory (OOM) for resolution 1248, LION-S only needs ∼ 6 GB which is ∼ 94.4% more efficient. Similarly, BERT goes OOM for sequence length 14, 336, while for the same sequence length, LION-S requires less than 15GB of GPU memory.

ViT-T, which has a quadratic complexity performs slightly better than LION-S on ImageNet and worse in other datasets. When considering a larger ViT-Small (21.7M) parameters, this gap between transformer and LION-S on ImageNet gets smaller. LION-S (v2) significantly improves the performance of LION-S in all tested scenarios and over ViT-T on CIFAR-100. For further ablations, *cf.*, Appendix D.7.

## 5.4 CONTEXT EXTENSION AND MEMORY DURING INFERENCE

When considering the number of tokens (analogous to resolution in images) that Transformer-like architectures can effectively process, two primary limitations emerge: ($i$) positional embeddings and ($ii$) memory constraints. Transformers are typically trained up to a specific sequence length and lack predefined positional encodings for tokens that exceed this limit, which can hurt their ability to recognize token positions beyond trained lengths. Furthermore, the quadratic complexity of these models during inference places significant demands on memory resources, often leading to constraints that reduce processing efficiency.

LION-S architecture is free of these two limitations. In Figure 3 (*Left*), we test the LION-S model trained on 128 tokens tested on different lengths. While BERT and LION-LIT models peak at the training length, afterwards they experience a sharp decrease. LION-S, on the other hand, maintains its high performance at 3.5 times of training sequence length. Additionally, LION-S can infer beyond the 512 tokens, as compared to the other models.

For memory usage during inference, as Figure 3 (*Center/Right*) illustrates, LION-S demands significantly lower memory than ViT on image classification or BERT on MLM tasks. Due to the quadratic complexity of Transformers, as the resolution of the image increases, the memory consumption also drastically changes. As a result, ViT and BERT go out of memory (OOM) even with small batch sizes. With the LION-S architecture, thanks to the linear complexity during inference, the change in memory consumption is minimal. At 1248 resolution, LION-S is ∼ 94.4% more efficient than the ViT-T model. Similarly, at sequence length 14, 336, LION-S is 83.35% more efficient than BERT. These two strengths combined, i.e., usage of the recurrence parameters and linear inference complexity, allow LION-S to efficiently extrapolate beyond the context length (or resolution) during inference. For further results on length expansion, see Appendices A and D.7.

## 6 CONCLUSIONS

This paper presents the LION framework, which casts bidirectional Transformers as bidirectional RNNs. Notably, LION allows popular linear recurrent models, such as Linear Transformers, to leverage the well-established transformer training pipeline during training, while benefiting from efficient recurrence during inference in the bidirectional setting. The main examples of the LION framework include LION-LIT (without masking the attention), LION-RETNET (which uses a fixed mask inspired by Sun et al. (2023)), and LION-S (which employs a selective mask similar to Dao & Gu (2024); Yang et al. (2023)). Our experiments show that the LION framework facilitates efficient inference and parallel training in the bidirectional setting, while models built upon LION excel at handling complex tasks, such as long-range dependencies in vision and bidirectional language modeling, all while using modest computational and memory resources.

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

## APPENDIX

In and this following sections we include additional experiments, further insights on related work, proofs, and theoretical details. The sections are organized as follows:

- In Appendix A, we provide additional experiments on causal language modeling.
- In Appendix B, we include an extension of related work.
- In Appendix C, we present proofs and theoretical details.
- In Appendix D, we explore ablation studies and parameter configurations for LRA and Image Classification tasks.

## A  CAUSAL LANGUAGE MODELLING

Efficient causal language modelling with linearized attention was previously studied by Katharopoulos et al. (2020) and Sun et al. (2023). In this setup, our formulation becomes similar to the retentive network (Sun et al., 2023), with the difference that Sun et al. (2023) choose their selective parameters before training and keep them fixed, while our selective parameters ($a_i$ in Eq. (28)) are trained jointly with the rest of the model.

We evaluate the performance of our formulation against the GPT-2 architecture (Radford et al., 2019) and its linearized version obtained by simply removing the softmax (LinAtt). Our architectures LION-S is trained with a trainable linear layer to obtain input-dependent selectivity, i.e., $a_i = \log(\sigma(\mathbf{W}_a\mathbf{x}_i + b))$ in Eq. (28). Note that our model do not use absolute positional encodings. We train our models in the OpenWebText corpus (Gokaslan & Cohen, 2019). We evaluate the architectures in the 124 M parameter setup. Our implementation is based on nanoGPT[2]. We use the default GPT-2 hyperparameters and train our models for 8 days in 4 NVIDIA A100 SXM4 40 GB GPUs.

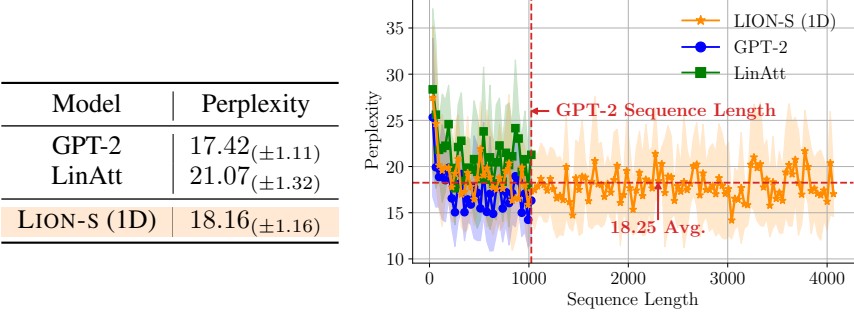

| Model | Perplexity |
|---|---|
| GPT-2 | $17.42_{(\pm1.11)}$ |
| LinAtt | $21.07_{(\pm1.32)}$ |
| LION-S (1D) | $18.16_{(\pm1.16)}$ |

(a) OpenWebText PPL

(b) Perplexity vs. sequence length

Figure 4: *Causal Language Modelling results in the GPT-2 128M size. (a)* Perplexity in the OpenWebText dataset. *(b)* Perplexity vs. sequence length in OpenWebText. Our models improve over the LinAtt baseline (Katharopoulos et al., 2020) while obtaining similar performance to the GPT baseline and being able to extrapolate to larger context lengths than the one used during training.

In Figure 4 we can observe LION-S (1D) significantly improve over the LinAtt baseline, while obtain perplexity close to GPT-2. The lack of absolute positional encodings allows LION-S (1D) to scale to larger sequence lengths than the one used during training.

In Figure 5 we evaluate the latency and memory of LION-S (1D) in three modes: Attention, Attention + KV cache and RNN. While the three modes have the same output, the RNN formulation allows to save computation from previous token generations to require constant memory and latency for generating the next token. Our results align with the findings of Sun et al. (2023), showing that efficient models in training and inference, with a strong performance (up to a small degradation) can be obtained.

---

[2]https://github.com/karpathy/nanoGPT

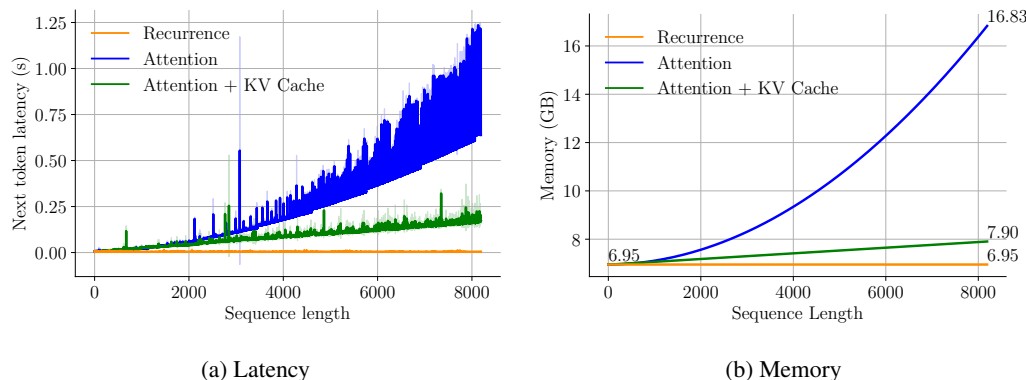

(a) Latency (b) Memory

Figure 5: *Efficiency of the* LION-S *(1D) framework in the next-token generation task.* In *(a)* and *(b)* we measure respectively the latency and memory to generate the next token in the sentence. We compare three generation modes: Attention, Attention with KV cache and the Recurrence formulation. While all three produce the same output, the Recurrence formulation is the most efficient, requiring constant memory and latency to generate the next token.

## B  DETAILED RELATED WORK

### B.1  STATE SPACE MODELS AND TRANSFORMERS

State Space Models, such as S4 (Gu et al., 2022) and S5 (Smith et al., 2023), advanced efficient sequence modeling with linear complexity. Mamba (Gu & Dao, 2024) and Mamba-2 (Dao & Gu, 2024) introduced selective mechanisms within SSMs, achieving strong language modeling performance. Recently, many recurrent models for language have been proposed, e.g., xLSTM (Beck et al., 2024), RWKV (Peng et al., 2024). While RNNs for autoregressive modelling are prevalent, bidirectional models are less explored. Hydra (Hwang et al., 2024) extends Mamba to bidirectional settings using quasiseparable matrix mixers. VisionMamba (Zhu et al., 2024b) employs two separate SSMs to pass over images. However, these works are not equivalent to bidirectional attention. LION adopts a different approach: instead of extending SSMs, we derive equivalence between bidirectional attention with learnable mask and bidirectional RNNs.

Since the pioneering works (Tsai et al., 2019; Katharopoulos et al., 2020), many works have been proposed to enhance linearized attention, including learnable relative positional encoding (Dai et al., 2019), gate mechanisms (Peng et al., 2021; Han et al., 2024; Ma et al., 2022), FFT for kernelized attention (Luo et al., 2021), decay terms in RetNet (Sun et al., 2023), and variants with enhanced expressiveness (Arora et al., 2024; Zhang et al., 2024; Yang et al., 2024). These works focus on causal attention and cannot be directly applied with bidirectionality, while we explicitly write bidirectional attention as bidirectional RNN combined with selectivity, enhancing performance and providing a principled framework for parallel training and linear-time inference in non-causal tasks.

### B.2  LINEAR RECURRENT MODELS SUMMARY

As noted in Qin et al. (2022), scaling the attention can lead to performance instability; therefore, many linear recurrent models avoid scaling the attention matrix. Consequently, we categorize these models into two groups: scaled and non-scaled. We discuss how various choices of parameters and their corresponding operational functions result in different well-known SSMs and Transformers at Table 5.

### B.3  PARALLEL TRAINING AND EFFICIENT INFERENCE

For linear recurrent models, efficient training is ideally achieved either by employing a form similar to $\mathbf{Y} = \text{softmax}\left(\mathbf{Q}\mathbf{K}^{\top}\right)\mathbf{V}$ or by using techniques like parallel scan, as utilized by many SSMs (e.g., Mamba, S5) (Blelloch, 1990). We will cover both techniques in the following sections.

Table 5: *Overview of recent linear recurrent models applied to autoregressive language modeling.* The $-$ mark indicates models without scaling, as they lack $\alpha_i$ and $\beta_i$ and do not scale attention scores. The $\times$ denotes matrix multiplication, $\odot$ represents the Hadamard product, and $*$ signifies the scalar product. All these models are used for autoregressive language modeling. To our knowledge, bi-directional SSMs or linear recurrent models with connections to Transformers, other than LION-LIT and LION-S, do not exist The order is approximately chronological.

| Model | Recurrence Parameters | | | | Operations | | Scaled | $\rightleftarrows$ |
|---|---|---|---|---|---|---|---|---|
| | $\Lambda_i$ | $\alpha_i$ | $\beta_i$ | $\gamma_i$ | $\bullet$ | $\star$ | | |
| Linear Trans (Katharopoulos et al., 2020) | $\mathbf{I}$ | 1 | 1 | 1 | $\times$ | $\times$ | ✓ | ✗ |
| DeltaNet (Schlag et al., 2021) | $\mathbf{I} - \gamma_i \mathbf{k}_i \mathbf{k}_i^\top$ | $-$ | $-$ | $\gamma_i$ | $*$ | $\times$ | ✗ | ✗ |
| S4/S5 (Gu et al., 2022; Smith et al., 2023) | $\mathbf{e}^{(-(\delta \mathbf{1}^\top) \odot \exp(\mathbf{A}))}$ | $-$ | $-$ | $\boldsymbol{B}$ | $\odot$ | $\odot$ | ✗ | ✗ |
| Gated RFA (Peng et al., 2021) | $g_i$ | $g_i$ | $1 - g_i$ | $1 - g_i$ | $*$ | $*$ | ✓ | ✗ |
| RetNet (Sun et al., 2023) | a | $-$ | $-$ | 1 | $*$ | $*$ | ✗ | ✗ |
| Mamba (S6) (Gu & Dao, 2024) | $\mathbf{e}^{(-(\delta_i \mathbf{1}^\top) \odot \exp(\mathbf{A}_i))}$ | $-$ | $-$ | $\boldsymbol{B}_i$ | $\odot$ | $\odot$ | ✗ | ✗ |
| GLA (Yang et al., 2023) | $\mathrm{DIAG}(g_i)$ | $-$ | $-$ | 1 | $*$ | $\times$ | ✗ | ✗ |
| RWKV (Peng et al., 2024) | $\mathrm{DIAG}(g_i)$ | $-$ | $-$ | 1 | $*$ | $\times$ | ✗ | ✗ |
| xLSTM (Beck et al., 2024) | $f_i$ | $f_i$ | $i_i$ | $i_i$ | $*$ | $*$ | ✓ | ✗ |
| Mamba-2 (Dao & Gu, 2024) | $a_i$ | $-$ | $-$ | 1 | $*$ | $*$ | ✗ | ✗ |
| LION-LIT **(ours)** | 1 | 1 | 1 | 1 | $\odot$ | $*$ | ✓ | ✓ |
| LION-S **(ours)** | $e^{-a_i}$ | $e^{-a_i}$ | 1 | 1 | $\odot$ | $*$ | ✓ | ✓ |

**Parallel training in transformers.** As illustrated in equation $\mathbf{Y} = \mathrm{softmax}\left(\mathbf{Q}\mathbf{K}^\top\right)\mathbf{V}$ of the Transformer, vectorization over the sequence is crucial to avoid sequential operations, where the model iterates over the sequence, leading to extensive training times (Vaswani et al., 2017). Parallelizing the operations across the sequence length for linear recurrent models ideally should take a form similar to (Katharopoulos et al., 2020; Sun et al., 2023):

$$\mathbf{Y} = \mathbf{M} * \left(\phi(\mathbf{Q})\phi(\mathbf{K})^\top\right)\mathbf{V} \tag{29}$$

Here, $\mathbf{M}$ represents a mask generated from the interaction of recurrent model parameters ($\Lambda_i$, $\gamma_i$, $\alpha_i$, $\beta_i$). Attention scores can be scaled before or after applying the mask $\mathbf{M}$ and during inference the scaling can be done by using the scaling state $\mathbf{z}_i$. The symbol $*$ indicates the operation in-which mask is applied to the attention. Equation (29) highlights the importance of carefully selecting operations and parameters to ensure parallelizability during training. The mask $\mathbf{M}$ is a lower diagonal mask in case of autoregressive models (Ma et al., 2022).

**Parallel Scan.** Most SSMs utilize the state matrix $\Lambda_i$ as a full matrix, with the $\star$ operation defined as matrix multiplication. Consequently, the output of each layer cannot be represented as in (29). This limitation becomes evident when applying recurrence over the discrete sequence in (3), leading to the output:

$$\mathbf{y}_i = \bar{\boldsymbol{C}}_i^\top \sum_{j=1}^{i} \left( \prod_{k=j+1}^{i} \bar{\boldsymbol{A}}_k \right) \bar{\boldsymbol{B}}_j \mathbf{x}_j, \tag{30}$$

which requires matrix multiplications for $\bar{\boldsymbol{A}}_k$ across all tokens between $i$ and $j$, resulting in substantial memory requirements during training.

To mitigate this issue, SSMs adopt the parallel scan approach (Blelloch, 1990; Ladner & Fischer, 1980), which enables efficient parallelization over sequence length. Initially introduced in S5 (Smith et al., 2023), this method has a time complexity of $\mathcal{O}(L \log L)$. However, Mamba (Gu & Dao, 2024) improves upon this by dividing storage and computation across GPUs, achieving linear scaling of $\mathcal{O}(L)$ with respect to sequence length and enabling parallelization over the state dimension $N$. Ideally, a model should achieve complete parallelization in training without sequential operations, maintain a memory requirement for inference independent of token count, and have linear complexity. Table 1 summarizes various training and inference strategies, along with their complexity and memory demands.

Linear recurrent models (Sun et al., 2023; Katharopoulos et al., 2020) employ attention during training and recurrence during inference, placing them in the last category of Table 1. To our knowledge, an

exact mapping between attention and bidirectional recurrence does not exist; thus, naive forward and backward recurrence cannot be theoretically equated to the attention formulations in (29) and $\mathbf{Y} = \text{softmax}\left(\mathbf{QK}^\top\right)\mathbf{V}$.

### B.4 ARCHITECTURAL DIFFERENCES IN AUTOREGRESSIVE LINEAR RECURRENT MODELS

**Multi-head attention and state expansion.** Another difference between various linear recurrent models, particularly SSMs and transformers, is how they expand single-head attention or SSM recurrence (3) to learn different features at each layer, akin to convolutional neurons in CNNs (He et al., 2016). Transformers achieve this through multi-head attention, while SSMs like Mamba and Mamba-2 (Gu & Dao, 2024; Dao & Gu, 2024) use state expansion also known as Single-Input Single-Output (*SISO*) framework to enlarge the hidden state. In *SISO* framework, the input $\mathbf{x}_i$ in (3) is a scalar and recurrence is applied to all elements in the hidden state independently (Smith et al., 2023), allowing for parallelization during inference and training.

In contrast, simplified SSMs like S5 employ a Multiple-Input Multiple-Output (*MIMO*) approach, where $\mathbf{x}_i$ is a vector, which aligns them more closely with RNN variants like LRU (Orvieto et al., 2023) that are successful in long-range modeling (Smith et al., 2023). However, the *SISO* framework continues to be effective in Mamba models for language modeling (Dao & Gu, 2024).

**Rule of Positional Encoding.** The parameter $\mathbf{\Lambda}_i$ serves as a gating mechanism (Yang et al., 2023; Gu & Dao, 2024) and can also be interpreted as relative positional encoding (Sun et al., 2023). For instance, in an autoregressive model, considering $\mathbf{\Lambda}_i$ as scaler, the mask $\mathbf{M}$ can be defined as follows:

<table>
<tr><td align="center">**SELECTIVE MASK**</td><td align="center">**FIXED MASK**</td></tr>
</table>

$$\mathbf{M}_{ij} = \begin{cases} \Pi_{k=i}^{j+1}\lambda_k & i \geq j \\ 0 & i < j \end{cases} \qquad (31) \qquad\qquad \mathbf{M}_{ij} = \begin{cases} \lambda^{i-j} & i \geq j \\ 0 & i < j \end{cases} \qquad (32)$$

In this context, the selective mask (where $\mathbf{\Lambda}_i = \lambda_i$ varies for each token) is used in architectures like Mamba (Gu & Dao, 2024), while the fixed mask (where $\mathbf{\Lambda}_i = \lambda$ is constant across all tokens) is implemented in architectures like RetNet (Sun et al., 2023). In both cases, the mask $\mathbf{M}_{ij}$ provides rich relative positional encoding between tokens $i$ and $j$. Its structure reinforces the multiplication of all $\mathbf{\Lambda}_k$ elements for $k \in [j, \ldots, i]$, while the selectivity allows the model to disregard noisy tokens, preventing their inclusion in the attention matrix for other tokens.

In contrast, linear recurrent models such as Linear Transformer (Katharopoulos et al., 2020) set $\mathbf{\Lambda}_k = 1$, resulting in $\mathbf{M}$ functioning as a standard causal mask, similar to those used in generative transformers (Kojima et al., 2022). This necessitates the injection of positional information into the sequence, which is achieved using the traditional positional encoding employed in transformers (Vaswani et al., 2017). In this framework, each element of the input data sequence is represented as $\mathbf{x}_i = \mathbf{f}_i + \mathbf{t}_i$, where $\mathbf{f}_i$ denotes the features at time $i$ and $\mathbf{t}_i$ represents the positional embedding. However, this traditional positional encoding has been shown to be less informative compared to relative positional encoding (Su et al., 2024), which is utilized in other linear recurrent models where $\mathbf{\Lambda}_k \neq 1$.

### B.5 MEMORY ALLOCATION IN LION DURING FORWARD AND BACKWARD RECURRENCES

During the forward and backward recurrences, as illustrated in Figure 6, each recurrence saves its corresponding output vector for each token, along with the scaling factor $c$, to generate the final output. Once the backward recurrence reaches a token that the forward recurrence has already passed, it can directly calculate the output $\mathbf{y}_i$ for that token, as $c_i^F$ and $\mathbf{y}_i^F$ have already been computed during the forward pass. Furthermore, the backward recurrence can overwrite the final output in the same memory cell where $\mathbf{y}_i^F$ was stored, since both outputs share the same dimensions. This approach keeps memory allocation consistent with the forward pass, and the time required to process the sequence remains similar to that of autoregressive models, as both recurrences can traverse the sequence in parallel.

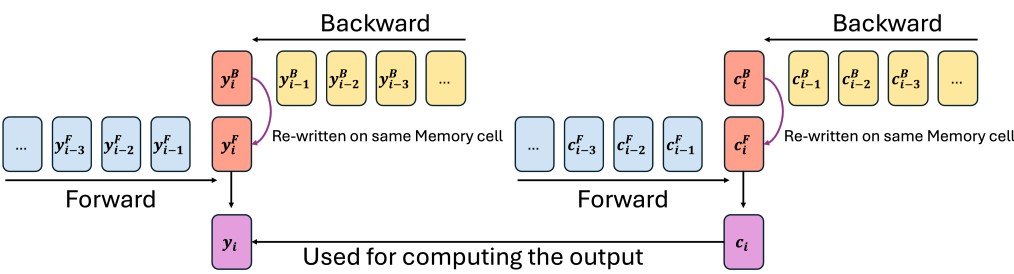

Figure 6: *Memory allocation in* LION *during Forward and Backward recurrences.* The efficient way of re-using the memory during inference is explained.

### B.6 ZERO-ORDER HOLD DISCRETIZATION

Below we explain the zero-order hold discretization derived by Kalman (1960). An LTI system can be represented with the equation:

$$\dot{\mathbf{h}}(t) = A\mathbf{h}(t) + B\mathbf{x}(t), \tag{33}$$

which can be rearranged to isolate $\mathbf{h}(t)$:

$$\dot{\mathbf{h}}(t) - A\mathbf{h}(t) = B\mathbf{x}(t). \tag{34}$$

. By multiplying the equation by $e^{-At}$, we get

$$e^{-At}\dot{\mathbf{h}}(t) - e^{-At}A\mathbf{h}(t) = e^{-At}B\mathbf{x}(t) \tag{35}$$

Since $\frac{\partial}{\partial t}e^{At} = Ae^{At} = e^{At}A$, Eq. (35) can be written as:

$$\frac{\partial}{\partial t}\left(e^{-At}\mathbf{h}(t)\right) = e^{-At}B\mathbf{x}(t). \tag{36}$$

After integrating both sides and simplifications, we get

$$e^{-At}\mathbf{h}(t) = \int_0^t e^{-A\tau}B\mathbf{x}(\tau)\,d\tau + \mathbf{h}(0). \tag{37}$$

By multiplying both sides by $e^{At}$ to isolate $\mathbf{h}(t)$ and performing further simplifications, at the end we get

$$\mathbf{h}(t) = e^{At}\int_0^t e^{-A\tau}B\mathbf{x}(\tau)\,d\tau + e^{At}\mathbf{h}(0). \tag{38}$$

To discretize this solution, we can assume sampling the system at even intervals, i.e. each sample is at $kT$ for some time step $T$, and that the input $\mathbf{x}(t)$ is constant between samples. To simplify the notation, we can define $\mathbf{h}_k$ in terms of $\mathbf{h}(kT)$ such that

$$\mathbf{h}_k = \mathbf{h}(kT). \tag{39}$$

Using the new notation, Eq. (38) becomes

$$\mathbf{h}_k = e^{\mathbf{A}kT}\mathbf{h}(0) + e^{\mathbf{A}kT}\int_0^{kT} e^{-\mathbf{A}\tau}\mathbf{B}\mathbf{x}(\tau)\,d\tau. \tag{40}$$

Now we want to express the system in the form:

$$\mathbf{h}_{k+1} = \tilde{\mathbf{A}}\mathbf{h}_k + \tilde{\mathbf{B}}\mathbf{x}_k. \tag{41}$$

To start, let's write out the equation for $\mathbf{x}_{k+1}$ as

$$\mathbf{h}_{k+1} = e^{\mathbf{A}(k+1)T}\mathbf{h}(0) + e^{\mathbf{A}(k+1)T}\int_0^{(k+1)T} e^{-\mathbf{A}\tau}\mathbf{B}\mathbf{x}(\tau)\,d\tau. \tag{42}$$

After multiplying by $e^{\mathbf{A}T}$ and rearranging we get

$$e^{\mathbf{A}(k+1)T}\mathbf{h}(0) = e^{\mathbf{A}T}\mathbf{h}_k - e^{\mathbf{A}(k+1)T}\int_0^{kT} e^{-\mathbf{A}\tau}\mathbf{B}\mathbf{x}(\tau)\,d\tau. \tag{43}$$

Plugging this expression for $\mathbf{x}_{k+1}$ in Eq. (42) yields to

$$\mathbf{h}_{k+1} = e^{\mathbf{A}T}\mathbf{h}_k - e^{\mathbf{A}(k+1)T}\left(\int_0^{kT} e^{-\mathbf{A}\tau}\mathbf{B}\mathbf{x}(\tau)\,d\tau + \int_0^{(k+1)T} e^{-\mathbf{A}\tau}\mathbf{B}\mathbf{x}(\tau)\,d\tau\right), \tag{44}$$

which can be further simplified to

$$\mathbf{h}_{k+1} = e^{\mathbf{A}T}\mathbf{h}_k - e^{\mathbf{A}(k+1)T}\int_{kT}^{(k+1)T} e^{-\mathbf{A}\tau}\mathbf{B}\mathbf{x}(\tau)\,d\tau. \tag{45}$$

Now, assuming that $\mathbf{x}(t)$ is constant on the interval $[kT, (k+1)T)$, which allows us to take $\mathbf{B}\mathbf{x}(t)$ outside the integral. Moreover, by bringing the $e^{\mathbf{A}(k+1)T}$ term inside the integral we have

$$\mathbf{h}_{k+1} = e^{\mathbf{A}T}\mathbf{h}_k - \int_{kT}^{(k+1)T} e^{\mathbf{A}((k+1)T-\tau)}\,d\tau\,\mathbf{B}\mathbf{x}_k. \tag{46}$$

Using a change of variables $v = (k+1)T - \tau$, with $d\tau = -dv$, and reversing the integration bounds results in

$$\mathbf{h}_{k+1} = e^{\mathbf{A}T}\mathbf{h}_k + \int_0^T e^{\mathbf{A}v}\,dv\,\mathbf{B}\mathbf{x}_k. \tag{47}$$

Finally, if we evaluate the integral by noting that $\frac{d}{dt}e^{\mathbf{A}t} = \mathbf{A}e^{\mathbf{A}t}$ and assuming $\mathbf{A}$ is invertible, we get

$$\mathbf{h}_{k+1} = e^{\mathbf{A}T}\mathbf{h}_k + \mathbf{A}^{-1}\left(e^{\mathbf{A}T} - \mathbf{I}\right)\mathbf{B}\mathbf{x}_k. \tag{48}$$

Thus, we find the discrete-time state and input matrices:

$$\tilde{\mathbf{A}} = e^{\mathbf{A}T} \tag{49}$$

$$\tilde{\mathbf{B}} = \mathbf{A}^{-1}\left(e^{\mathbf{A}T} - \mathbf{I}\right)\mathbf{B}. \tag{50}$$

And the final desecrate state space representation is:

$$\mathbf{h_k} = e^{\mathbf{A}T}\mathbf{h}_{k-1} + \mathbf{A}^{-1}\left(e^{\mathbf{A}T} - \mathbf{I}\right)\mathbf{B}_k\mathbf{x}_k. \tag{51}$$

As in case of LION-S (similar to choice of mamba2 Dao & Gu (2024)) the matrix $\mathbf{A}$ is identity while the time step $T$ is selective and equal to $a_i$. And simply for LION-S scenario the term $Bx(t)$ will change into $\mathbf{k}_i\mathbf{v}_i^\top$ therefor considering Linear Transformer as continuous system like:

$$\mathbf{S}'_{(t)} = \mathbf{S}_{(t)} + \mathbf{k}_{(t)}\mathbf{v}_{(t)}^\top, \tag{52}$$

$$\mathbf{z}_{(t)} = \mathbf{z}_{(t)} + \mathbf{k}_{(t)}, \tag{53}$$

$$\tag{54}$$

By applying the ZOH discritization the final descreate LION-S will be equal to:

$$\text{DISCRETE}$$

$$\mathbf{S}_i = e^{a_i}\mathbf{S}_{i-1} + (e^{a_i} - 1)\mathbf{k}_i\mathbf{v}_i^\top, \tag{55}$$

$$\mathbf{z}_i = e^{a_i}\mathbf{z}_{i-1} + (e^{a_i} - 1)\mathbf{k}_i, \tag{56}$$

And it applies to both directions forward and backward.

## C  PROOFS

### C.1  PROOF OF PROP. 3.2: DUALITY BETWEEN LINEAR RECURRENCE AND ATTENTION

Considering the following recurrence:

$$\mathbf{S}_i = \lambda_i \mathbf{S}_{i-1} + \mathbf{k}_i \mathbf{v}_i^\top, \tag{57}$$

$$\mathbf{z}_i = \lambda_i \mathbf{z}_{i-1} + \mathbf{k}_i, \tag{58}$$

$$\text{SCALED} : \mathbf{y}_i = \frac{\mathbf{q}_i^\top \mathbf{S}_i}{\mathbf{q}_i^\top \mathbf{z_i}} \tag{59}$$

We can calculate each output $\mathbf{y}_i$ recursively as below:

$$\mathbf{S}_1 = \mathbf{k}_1 \mathbf{v}_1^\top, \ \ \mathbf{z}_1 = \mathbf{k}_1, \ \ \mathbf{y}_1 = \mathbf{v}_1 \tag{60}$$

$$\mathbf{S}_2 = \mathbf{k}_2 \mathbf{v}_2^\top + \lambda_1 \mathbf{k}_1 \mathbf{v}_1^\top, \ \ \mathbf{z}_2 = \mathbf{k}_2 + \lambda_1 \mathbf{k}_1, \ \ \mathbf{y}_2 = \frac{\mathbf{q}_2^\top (\mathbf{k}_2 \mathbf{v}_2^\top + \lambda_1 \mathbf{k}_1 \mathbf{v}_1^\top)}{\mathbf{q}_2^\top (\mathbf{k}_2 + \lambda_1 \mathbf{k}_1)} \tag{61}$$

$$\mathbf{S}_3 = \mathbf{k}_3 \mathbf{v}_3^\top + \lambda_1 \mathbf{k}_2 \mathbf{v}_2^\top + \lambda_2 \lambda_1 \mathbf{k}_1 \mathbf{v}_1^\top, \ \ \mathbf{z}_3 = \mathbf{k}_3 + \lambda_1 \mathbf{k}_2 + \lambda_2 \lambda_1 \mathbf{k}_1, \ \ \mathbf{y}_3 = \frac{\mathbf{q}_3^\top (\mathbf{k}_3 \mathbf{v}_3^\top + \lambda_1 \mathbf{k}_2 \mathbf{v}_2^\top + \lambda_2 \lambda_1 \mathbf{k}_1 \mathbf{v}_1^\top)}{\mathbf{q}_3^\top (\mathbf{k}_3 + \lambda_1 \mathbf{k}_2 + \lambda_2 \lambda_1 \mathbf{k}_1)} \tag{62}$$

$$\Rightarrow \mathbf{y}_i = \frac{\mathbf{q}_i^\top (\sum_{j=1}^i \mathbf{M}_{ij}^C \mathbf{k}_j \mathbf{v}_j^\top)}{\mathbf{q}_i^\top (\sum_{j=1}^i \mathbf{M}_{ij}^C \mathbf{k}_j)}, \qquad \mathbf{M}_{ij}^C = \begin{cases} \Pi_{k=i}^{j+1} \lambda_k & i \geq j \\ 0 & i < j \end{cases} \tag{63}$$

This can be shown in a vectorized form as:

$$\mathbf{Y} = \text{SCALE}(\mathbf{Q}\mathbf{K}^\top \odot \mathbf{M}^C)\mathbf{V} \tag{64}$$

Where SCALE is the scaling function which scaled the attention matrix with respect to each row or can also be written as:

$$\text{SCALE}(\mathbf{A})_{ij} = \frac{\mathbf{A}_{ij}}{\sum_{j=1}^L \mathbf{A}_{ij}} \tag{65}$$

Similarly if the SCALE is applied before masking we have:

$$\mathbf{Y} = \big(\text{SCALE}(\mathbf{Q}\mathbf{K}^\top \odot \mathbf{M}_{\text{CAUSAL}}) \odot \mathbf{M}\big)\mathbf{V} \tag{66}$$

With $\mathbf{M}_{\text{CAUSAL}}$ being the causal mask used in autoregressive models (Kojima et al., 2022). This vectorized form is equivalent to:

$$\mathbf{y}_i = \frac{\mathbf{q}_i^\top (\sum_{j=1}^i \mathbf{M}_{ij} \mathbf{k}_j \mathbf{v}_j^\top)}{\mathbf{q}_i^\top (\sum_{j=1}^i \mathbf{k}_j)}, \qquad \mathbf{M}_{ij} = \begin{cases} \Pi_{k=i}^{j+1} \lambda_k & i \geq j \\ 0 & i < j \end{cases} \tag{67}$$

And the recurrence for this vectorized form can be written as:

$$\mathbf{S}_i = \lambda_i \mathbf{S}_{i-1} + \mathbf{k}_i \mathbf{v}_i^\top, \tag{68}$$

$$\mathbf{z}_i = \mathbf{z}_{i-1} + \mathbf{k}_i, \tag{69}$$

$$\text{SCALED} : \mathbf{y}_i = \frac{\mathbf{q}_i^\top \mathbf{S}_i}{\mathbf{q}_i^\top \mathbf{z_i}} \tag{70}$$

### C.2 FORWARD AND BACKWARD RECURRENCES THEORETICAL DETAILS

Considering the following recurrence:

$$\mathbf{S}_i = \lambda_i \mathbf{S}_{i-1} + \mathbf{k}_i \mathbf{v}_i^\top, \tag{71}$$

$$\mathbf{z}_L = \sum_{i=1}^L \mathbf{k}_i \tag{72}$$

$$\mathbf{y}_i = \frac{\mathbf{q}_i^\top \mathbf{S}_i}{\mathbf{q}_i^\top \mathbf{z_L}} \tag{73}$$

This recurrence is the same as recurrence (68) but with $\mathbf{z}_L$ being fixed to the summation of all keys in the sequence, therefor the output $\mathbf{y}_i$ can simply be written as:

$$\mathbf{y}_i = \frac{\mathbf{q}_i^\top \left( \sum_{j=1}^i \mathbf{M}_{ij} \mathbf{k}_j \mathbf{v}_j^\top \right)}{\mathbf{q}_i^\top \mathbf{z}_L}, \qquad \mathbf{M}_{ij} = \begin{cases} \Pi_{k=i}^{j+1} \lambda_k & i \geq j \\ 0 & i < j \end{cases} \tag{74}$$

By replacing the $\mathbf{z}_i = \sum_{j=1}^i \mathbf{k}_j$ in the denominator of equation (70) with $\mathbf{z}_L$. Therefore in vectorized form, it will become:

$$\mathbf{Y} = (\mathbf{A}^C \odot \mathbf{M})\mathbf{V} \tag{75}$$

With $\mathbf{A}^C$ being:

$$\mathbf{A}^C = \begin{pmatrix} \frac{\mathbf{q}_1^\top \mathbf{k}_1}{\mathbf{q}_1^\top \mathbf{z}_L} & & & \\ \frac{\mathbf{q}_2^\top \mathbf{k}_1}{\mathbf{q}_2^\top \mathbf{z}_L} & \frac{\mathbf{q}_2^\top \mathbf{k}_2}{\mathbf{q}_2^\top \mathbf{z}_L} & & \\ \frac{\mathbf{q}_3^\top \mathbf{k}_1}{\mathbf{q}_3^\top \mathbf{z}_L} & \frac{\mathbf{q}_3^\top \mathbf{k}_2}{\mathbf{q}_3^\top \mathbf{z}_L} & \frac{\mathbf{q}_3^\top \mathbf{k}_3}{\mathbf{q}_3^\top \mathbf{z}_L} & \\ \vdots & \vdots & \vdots & \ddots \\ \frac{\mathbf{q}_L^\top \mathbf{k}_1}{\mathbf{q}_L^\top \mathbf{z}_L} & \frac{\mathbf{q}_L^\top \mathbf{k}_2}{\mathbf{q}_L^\top \mathbf{z}_L} & \cdots & \frac{\mathbf{q}_L^\top \mathbf{k}_L}{\mathbf{q}_L^\top \mathbf{z}_L} \end{pmatrix}$$

Importantly this equation can be written as:

$$\mathbf{Y} = \left( \text{SCALE}(\mathbf{Q}\mathbf{K}^\top) \odot \mathbf{M} \right)\mathbf{V} \tag{76}$$

which despite equation (66) scaling is applied over the whole sequence not for the causal part of the sequence. The matrix $\mathbf{A}^C$ is helpful for driving the recurrent version of LION for Forward and Backward recurrences and the mask here $\mathbf{M}$ is equal to LION's forward mask $\mathbf{M}^F$ in equation (16). As shown in (16) the forward recurrence for the causal part of the attention can be presented as $\mathbf{Y}^B = \mathbf{A}^F \odot \mathbf{M}^F$ the matrix $\mathbf{A}^F$ can be created simply by using matrix $\mathbf{A}^C$ as bellow:

$$\underbrace{\begin{pmatrix} \frac{1}{2}\frac{\mathbf{q}_1^\top \mathbf{k}_1}{\mathbf{q}_1^\top \mathbf{z}_L} & & & \\ \frac{\mathbf{q}_2^\top \mathbf{k}_1}{\mathbf{q}_2^\top \mathbf{z}_L} & \frac{1}{2}\frac{\mathbf{q}_2^\top \mathbf{k}_2}{\mathbf{q}_2^\top \mathbf{z}_L} & & \\ \frac{\mathbf{q}_3^\top \mathbf{k}_1}{\mathbf{q}_3^\top \mathbf{z}_L} & \frac{\mathbf{q}_3^\top \mathbf{k}_2}{\mathbf{q}_3^\top \mathbf{z}_L} & \frac{1}{2}\frac{\mathbf{q}_3^\top \mathbf{k}_3}{\mathbf{q}_3^\top \mathbf{z}_L} & \\ \vdots & \vdots & \vdots & \ddots \\ \frac{\mathbf{q}_L^\top \mathbf{k}_1}{\mathbf{q}_L^\top \mathbf{z}_L} & \frac{\mathbf{q}_L^\top \mathbf{k}_2}{\mathbf{q}_L^\top \mathbf{z}_L} & \cdots & \frac{1}{2}\frac{\mathbf{q}_L^\top \mathbf{k}_L}{\mathbf{q}_L^\top \mathbf{z}_L} \end{pmatrix}}_{\mathbf{A}^F} = \underbrace{\begin{pmatrix} \frac{\mathbf{q}_1^\top \mathbf{k}_1}{\mathbf{q}_1^\top \mathbf{z}_L} & & & \\ \frac{\mathbf{q}_2^\top \mathbf{k}_1}{\mathbf{q}_2^\top \mathbf{z}_L} & \frac{\mathbf{q}_2^\top \mathbf{k}_2}{\mathbf{q}_2^\top \mathbf{z}_L} & & \\ \frac{\mathbf{q}_3^\top \mathbf{k}_1}{\mathbf{q}_3^\top \mathbf{z}_L} & \frac{\mathbf{q}_3^\top \mathbf{k}_2}{\mathbf{q}_3^\top \mathbf{z}_L} & \frac{\mathbf{q}_3^\top \mathbf{k}_3}{\mathbf{q}_3^\top \mathbf{z}_L} & \\ \vdots & \vdots & \vdots & \ddots \\ \frac{\mathbf{q}_L^\top \mathbf{k}_1}{\mathbf{q}_L^\top \mathbf{z}_L} & \frac{\mathbf{q}_L^\top \mathbf{k}_2}{\mathbf{q}_L^\top \mathbf{z}_L} & \cdots & \frac{\mathbf{q}_L^\top \mathbf{k}_L}{\mathbf{q}_L^\top \mathbf{z}_L} \end{pmatrix}}_{\mathbf{A}^C} - \underbrace{\begin{pmatrix} \frac{1}{2}\frac{\mathbf{q}_1^\top \mathbf{k}_1}{\mathbf{q}_1^\top \mathbf{z}_L} & & & \\ & \frac{1}{2}\frac{\mathbf{q}_2^\top \mathbf{k}_2}{\mathbf{q}_2^\top \mathbf{z}_L} & & \\ & & \frac{1}{2}\frac{\mathbf{q}_3^\top \mathbf{k}_3}{\mathbf{q}_3^\top \mathbf{z}_L} & \\ & & & \ddots \\ & & & & \frac{1}{2}\frac{\mathbf{q}_L^\top \mathbf{k}_L}{\mathbf{q}_L^\top \mathbf{z}_L} \end{pmatrix}}_{\mathbf{D}^F}$$

Or equivalently:

$$\mathbf{Y}^F = \mathbf{A}^F \odot \mathbf{M}^F = (\mathbf{A}^C - \mathbf{D}^F) \odot \mathbf{M}^F \tag{77}$$

Since the diagonal values of the mask $\mathbf{M}^F$ are all ones and the matrix $\mathbf{D}^F$ is diagonal, we have:

$$\mathbf{Y}^F = (\mathbf{A}^C - \mathbf{D}^F) \odot \mathbf{M}^F = \mathbf{A}^C \odot \mathbf{M}^F - \mathbf{D}^F \tag{78}$$

As $\mathbf{A}^C \odot \mathbf{M}^F$ corresponds to linear recurrence shown at (74). The vectorized form (78) can be presented as linear recurrence:

$$\mathbf{y}_i = \frac{\mathbf{q}_i^\top \left( \sum_{j=1}^i \mathbf{M}_{ij} \mathbf{k}_j \mathbf{v}_j^\top \right)}{\mathbf{q}_i^\top \mathbf{z}_L} - \frac{1}{2}\frac{\mathbf{q}_i^\top \mathbf{k}_i}{\mathbf{q}_i^\top \mathbf{z}_L}, \qquad \mathbf{M}_{ij} = \begin{cases} \Pi_{k=i}^{j+1} \lambda_k & i \geq j \\ 0 & i < j \end{cases} \tag{79}$$

This is equivalent to the linear recurrence presented in equation (73). The same theoretical approach applies to the backward recurrence, leading to the following linear recurrence for both recurrences:

$$\mathbf{S}_i^F = \lambda_i \mathbf{S}_{i-1}^F + \mathbf{k}_i \mathbf{v}_i^\top, \quad (80a) \qquad \mathbf{S}_i^B = \lambda_{L-i} \mathbf{S}_{i-1}^B + \mathbf{k}_{L-i+1} \mathbf{v}_{L-i+1}^\top, \qquad (81a)$$

$$\mathbf{y}_i^F = \frac{\mathbf{q}_i^\top \mathbf{S}_i^F}{\mathbf{q}_i^\top \mathbf{z}_L} - \frac{1}{2} \frac{\mathbf{q}_i^\top \mathbf{k}_i}{\mathbf{q}_i^\top \mathbf{z}_L} \quad (80b) \qquad \mathbf{y}_{L-i+1}^B = \frac{\mathbf{q}_{L-i+1}^\top \mathbf{S}_i^B}{\mathbf{q}_{L-i+1}^\top \mathbf{z}_L} - \frac{1}{2} \frac{\mathbf{q}_{L-i+1}^\top \mathbf{k}_{L-i+1}}{\mathbf{q}_{L-i+1}^\top \mathbf{z}_L} \qquad (81b)$$

However, the above equation requires access to the summation of scaling values $\mathbf{z}_L$. A naive approach would involve adding an additional scaling recurrence alongside the forward and backward recurrences to compute the summation of all keys in the sequence. This approach, however, is inefficient, as it complicates the process. While the forward and backward recurrences can traverse the sequence in parallel to obtain the forward and backward recurrences outputs $\mathbf{Y}^F$ and $\mathbf{Y}^B$, the scaling recurrence must be computed prior to these recurrences because both the forward and backward recurrences computations rely on the final scaling value $\mathbf{z}_L$ to generate their outputs.

### C.3   EFFICIENT AND SIMPLE METHOD FOR SCALING ATTENTION DURING INFERENCE

As shown in previous section scaled attention matrix can be formulated as two recurrences (80) and (81) with an additional recurrence to sum all the keys ($\mathbf{z}_L$). This section we will proof how to avoid an extra scaling recurrence by simple modifications to equation (80) and (81).

Considering having a scaling recurrence as part of forward and backward recurrence we will have:

$$\mathbf{S}_i^F = \lambda_i \mathbf{S}_{i-1}^F + \mathbf{k}_i \mathbf{v}_i^\top, \quad (82a) \qquad \mathbf{S}_i^B = \lambda_{L-i} \mathbf{S}_{i-1}^B + \mathbf{k}_{L-i+1} \mathbf{v}_{L-i+1}^\top, \qquad (83a)$$

$$\mathbf{z}_i^F = \mathbf{z}_{i-1}^F + \mathbf{k}_i \quad (82b) \qquad \mathbf{z}_i^B = \mathbf{z}_{i-1}^B + \mathbf{k}_{L-i+1} \qquad (83b)$$

$$c_i^F = \mathbf{q}_i^\top \mathbf{z}_i^F - \frac{1}{2} \mathbf{q}_i^\top \mathbf{k}_i \quad (82c) \qquad c_i^B = \mathbf{q}_{L-i+1}^\top \mathbf{z}_i^B - \frac{1}{2} \mathbf{q}_{L-i+1}^\top \mathbf{k}_{L-i+1} \qquad (83c)$$

$$\mathbf{y}_i^F = \mathbf{q}_i^\top \mathbf{S}_i^F - \frac{1}{2} \mathbf{q}_i^\top \mathbf{k}_i \mathbf{v}_i \qquad \mathbf{y}_{L-i+1}^B = \mathbf{q}_{L-i+1}^\top \mathbf{S}_i^B - \frac{1}{2} \mathbf{q}_{L-i+1}^\top \mathbf{k}_{L-i+1} \mathbf{v}_{L-i+1}^\top \quad (83d)$$
$$(82d)$$

The equations above are similar to the previous ones, with the addition of scalar states $c^F$ and $c^B$ for the backward and forward recurrences, respectively. During each recurrence, the outputs $\mathbf{y}_i^F$ and $\mathbf{y}_i^B$, along with the scalars $c_i^F$ and $c_i^B$, are saved for each token to construct the final output of each layer. *It is also important to note that there is no need to save $\mathbf{z}^F$ and $\mathbf{z}^B$ for each token; these states can simply be overwritten in memory.* The final output of each layer is equal to:

$$\mathbf{y}_i = \frac{\mathbf{y}_i^F + \mathbf{y}_i^B}{c_i^F + c_i^B} \tag{84}$$

Where $\mathbf{y}_i^F$ and $\mathbf{y}_i^B$ can be written as:

$$\mathbf{y}_i^F = \mathbf{q}_i^\top \left( \sum_{j=1}^i \mathbf{M}_{ij}^F \mathbf{k}_j \mathbf{v}_j^\top \right) - \frac{1}{2} \mathbf{q}_i^\top \mathbf{k}_i \mathbf{v}_i \;\;,\;\; \mathbf{y}_i^B = \mathbf{q}_i^\top \left( \sum_{j=i}^L \mathbf{M}_{ij}^B \mathbf{k}_j \mathbf{v}_j^\top \right) - \frac{1}{2} \mathbf{q}_i^\top \mathbf{k}_i \mathbf{v}_i \tag{85}$$

So the addition $\mathbf{y}_i^F + \mathbf{y}_i^B$ is equal to:

$$\mathbf{y}_i^F + \mathbf{y}_i^B = \mathbf{q}_i^\top \left( \sum_{j=1}^i \mathbf{M}_{ij}^F \mathbf{k}_j \mathbf{v}_j^\top \right) + \mathbf{q}_i^\top \left( \sum_{j=i}^L \mathbf{M}_{ij}^B \mathbf{k}_j \mathbf{v}_j^\top \right) - \mathbf{q}_i^\top \mathbf{k}_i \mathbf{v}_i \tag{86}$$

$$\Rightarrow \mathbf{y}_i^F + \mathbf{y}_i^B = \mathbf{q}_i^\top \left( \sum_{j=1}^i \mathbf{M}_{ij}^F \mathbf{k}_j \mathbf{v}_j^\top + \sum_{j=i}^L \mathbf{M}_{ij}^B \mathbf{k}_j \mathbf{v}_j^\top \right) - \mathbf{q}_i^\top \mathbf{k}_i \mathbf{v}_i \tag{87}$$

Where by considering the mask $\mathbf{M}$ as bellow:

$$\mathbf{M}_{ij} = \begin{cases} \Pi_{k=j}^{i+1}\lambda_k & i > j \\ \Pi_{k=i+1}^{j}\lambda_k & i < j \\ 1 & i = j \end{cases} = \begin{pmatrix} 1 & \lambda_2 & \lambda_2\lambda_3 & \cdots & \lambda_2\cdots\lambda_L \\ \lambda_1 & 1 & \lambda_3 & \cdots & \lambda_3\cdots\lambda_L \\ \lambda_1\lambda_2 & \lambda_2 & 1 & \cdots & \lambda_4\cdots\lambda_L \\ \vdots & \vdots & \vdots & \ddots & \vdots \\ \lambda_{L-1}\cdots\lambda_1 & \lambda_{L-1}\cdots\lambda_2 & \lambda_{L-1}\cdots\lambda_3 & \cdots & 1 \end{pmatrix} \tag{88}$$

The above mask is equal to $\mathbf{M}^F + \mathbf{M}^B - \mathbf{I}$, allowing equation (86) to be rewritten as:

$$\mathbf{y}_i^F + \mathbf{y}_i^B = \mathbf{q}_i^\top (\sum_{j=1}^{i} \mathbf{M}_{ij}^F \mathbf{k}_j \mathbf{v}_j^\top + \sum_{j=i}^{L} \mathbf{M}_{ij}^B \mathbf{k}_j \mathbf{v}_j^\top) - \mathbf{q}_i^{\ \top} \mathbf{k}_i \mathbf{v}_i \tag{89}$$

$$= \mathbf{q}_i^\top (\sum_{j=1}^{L} \mathbf{M}_{ij} \mathbf{k}_j \mathbf{v}_j^\top) + \mathbf{q}_i^{\ \top} \mathbf{k}_i \mathbf{v}_i - \mathbf{q}_i^{\ \top} \mathbf{k}_i \mathbf{v}_i \tag{90}$$

$$= \mathbf{q}_i^\top (\sum_{j=1}^{L} \mathbf{M}_{ij} \mathbf{k}_j \mathbf{v}_j^\top) \tag{91}$$

So we can finally find the output of each layer $\mathbf{y}_i$ as:

$$\mathbf{y}_i = \frac{\mathbf{y}_i^F + \mathbf{y}_i^B}{c_i^F + c_i^B} \xrightarrow{\text{Equation (91)}} \mathbf{y}_i = \frac{\mathbf{q}_i^\top (\sum_{j=1}^{L} \mathbf{M}_{ij} \mathbf{k}_j \mathbf{v}_j^\top)}{c_i^F + c_i^B} \tag{92}$$

It can easily be shown that:

$$c_i^F = \mathbf{q}_i^\top (\sum_{j=1}^{i} \mathbf{k}_j) - \frac{1}{2}\mathbf{q}_i^\top \mathbf{k}_i \quad , \quad c_i^B = \mathbf{q}_i^\top (\sum_{j=i}^{L} \mathbf{k}_j) - \frac{1}{2}\mathbf{q}_i^\top \mathbf{k}_i \tag{93}$$

$$\Rightarrow c_i^F + c_i^B = \mathbf{q}_i^\top (\sum_{j=1}^{L} \mathbf{k}_j) + \mathbf{q}_i^\top \mathbf{k}_i - \frac{1}{2}\mathbf{q}_i^\top \mathbf{k}_i - \frac{1}{2}\mathbf{q}_i^\top \mathbf{k}_i \tag{94}$$

$$\Rightarrow c_i^F + c_i^B = \mathbf{q}_i^\top (\sum_{j=1}^{L} \mathbf{k}_j) + \mathbf{q}_i^\top \mathbf{k}_i - \mathbf{q}_i^\top \mathbf{k}_i = \mathbf{q}_i^\top (\sum_{j=1}^{L} \mathbf{k}_j) = \mathbf{q}_i^\top \mathbf{z}_L \tag{95}$$

So the final output of the layer is:

$$\mathbf{y}_i = \frac{\mathbf{y}_i^F + \mathbf{y}_i^B}{c_i^F + c_i^B} = \frac{\mathbf{q}_i^\top (\sum_{j=1}^{L} \mathbf{M}_{ij} \mathbf{k}_j \mathbf{v}_j^\top)}{\mathbf{q}_i^\top (\sum_{j=1}^{L} \mathbf{k}_j)} \tag{96}$$

Alternatively, in vectorized form, it can be expressed as:

$$\mathbf{Y} = \mathbf{Y}^F + \mathbf{Y}^B = \big(\text{SCALE}(\mathbf{Q}\mathbf{K}^\top) \odot \mathbf{M}\big)\mathbf{V} \tag{97}$$

with $\mathbf{M}$ being the attention mask created by $\lambda_i$s as in equation 88.

## C.4 FLIPPING OPERATION IN BACKWARD RECURRENCE

Here we define the operation which flip the matrices $\mathbf{A}^B, \mathbf{M}^B$ for the reverse reccurence th goal is to find the $F(.)$ such that:

$$
\mathbf{A}^B = \begin{pmatrix} \frac{1}{2}\mathbf{q}_1^\top \mathbf{k}_1 & \mathbf{q}_1^\top \mathbf{k}_2 & \cdots & \mathbf{q}_1^\top \mathbf{k}_L \\ & \frac{1}{2}\mathbf{q}_2^\top \mathbf{k}_2 & \cdots & \mathbf{q}_2^\top \mathbf{k}_L \\ & & \ddots & \vdots \\ & & & \frac{1}{2}\mathbf{q}_L^\top \mathbf{k}_L \end{pmatrix} \rightarrow F(\mathbf{A}^B) = \begin{pmatrix} \frac{1}{2}\frac{\mathbf{q}_L^\top \mathbf{k}_L}{\mathbf{q}_L^\top \mathbf{z}_L} \\ \frac{\mathbf{q}_{L-1}^\top \mathbf{k}_L}{\mathbf{q}_2^\top \mathbf{z}_L} & \frac{1}{2}\frac{\mathbf{q}_{L-1}^\top \mathbf{k}_{L-1}}{\mathbf{q}_2^\top \mathbf{z}_L} \\ \vdots & \vdots & \ddots \\ \frac{\mathbf{q}_1^\top \mathbf{k}_L}{\mathbf{q}_1^\top \mathbf{z}_L} & \frac{\mathbf{q}_1^\top \mathbf{k}_{L-1}}{\mathbf{q}_1^\top \mathbf{z}_L} & \cdots & \frac{1}{2}\frac{\mathbf{q}_1^\top \mathbf{k}_1}{\mathbf{q}_1^\top \mathbf{z}_L} \end{pmatrix} \tag{98}
$$

$$
\mathbf{M}^B = \begin{pmatrix} 1 & \lambda_2 & \lambda_2\lambda_3 & \cdots & \lambda_2\cdots\lambda_L \\ & 1 & \lambda_3 & \cdots & \lambda_3\cdots\lambda_L \\ & & 1 & \cdots & \lambda_4\cdots\lambda_L \\ & & & \ddots & \vdots \\ & & & & 1 \end{pmatrix} \rightarrow F(\mathbf{M}^B) = \begin{pmatrix} 1 \\ \lambda_L & 1 \\ \lambda_L\lambda_{L-1} & \lambda_{L-1} & 1 \\ \vdots & \vdots & \vdots & \ddots \\ \lambda_L\cdots\lambda_2 & \lambda_L\cdots\lambda_3 & \lambda_L\cdots\lambda_4 & \cdots & 1 \end{pmatrix} \tag{99}
$$

The above can be achieved by:

$$
F(\mathbf{A}) = \mathbf{J}_L \mathbf{A} \mathbf{J}_L, \quad \mathbf{J}_L = \begin{pmatrix} & & & 1 \\ & & 1 & \\ & \cdot^{\cdot^{\cdot}} & & \\ 1 & & & \end{pmatrix} \tag{100}
$$

## C.5 MAPPING EXISTING AUTOREGRESSIVE MODELS INTO LION

As noted, other autoregressive recurrent models can also be integrated into our bidirectional framework, benefiting from parallelization during training and fast bidirectional inference. Here, we demonstrate how to map several well-known linear recurrent models into the bidirectional form of LION, along with their corresponding masked attention matrix and inference linear recurrence.

**Linear Transformer** (LION-LIT). According to Katharopoulos et al. (2020) the linear transformer has a recurrence:

$$
\mathbf{S}_i^F = \mathbf{S}_{i-1}^F + \mathbf{k}_i \mathbf{v}_i^\top, \tag{101}
$$

$$
\mathbf{z}_i^F = \mathbf{z}_{i-1}^F + \mathbf{k}_i, \tag{102}
$$

$$
\text{SCALED}: \mathbf{y}_i^F = \frac{\mathbf{q}_i^\top \mathbf{S}_i^F}{\mathbf{q}_i^\top \mathbf{z}_i^F} \tag{103}
$$

$$
\text{NON-SCALED}: \mathbf{y}_i^F = \mathbf{q}_i^\top \mathbf{S}_i^F \tag{104}
$$

As observed, this is a special case of our bidirectional recurrence defined in (24) with $\lambda_i = 1$, as LION resembles the scaled masked attention. In the case of the linear transformer, we require attention without scaling for the recurrence. The vectorized form for the scaled version can then be derived easily as follows:

$$\mathbf{S}_i^{F/B} = \mathbf{S}_{i-1}^{F/B} + \mathbf{k}_i \mathbf{v}_i^\top, \qquad (105)$$

$$\mathbf{z}_i^{F/B} = \mathbf{z}_{i-1}^{F/B} + \mathbf{k}_i \qquad (106)$$

$$c_i^{F/B} = \mathbf{q}_i^\top \mathbf{z}_i^{F/B} - \frac{1}{2}\mathbf{q}_i^\top \mathbf{k}_i, \quad (107)$$

$$\mathbf{y}_i^{F/B} = \mathbf{q}_i^\top \mathbf{S}_i^{F/B} - \frac{1}{2}\mathbf{q}_i^\top \mathbf{k}_i \mathbf{v}_i \qquad (108)$$

$$= \qquad \mathbf{Y} = \text{SCALE}(\mathbf{Q}\mathbf{K}^\top\mathbf{V}) \qquad (109)$$

For the non-scaled variant, we simply remove the scaling state $\mathbf{z}$ as well as the scaling parameter $c$. Consequently, the bidirectional linear transformer, which is equivalent to and parallelizable with attention without scaling, can be expressed as follows:

$$\mathbf{S}_i^{F/B} = \mathbf{S}_{i-1}^{F/B} + \mathbf{k}_i \mathbf{v}_i^\top, \qquad (110)$$

$$\mathbf{y}_i^{F/B} = \mathbf{q}_i^\top \mathbf{S}_i^{F/B} - \frac{1}{2}\mathbf{q}_i^\top \mathbf{k}_i \mathbf{v}_i \qquad (111)$$

$$= \qquad \mathbf{Y} = \mathbf{Q}\mathbf{K}^\top\mathbf{V} \qquad (112)$$

The final output for scaled version can be extracted as $\mathbf{y}_i = \frac{\mathbf{y}_i^B + \mathbf{y}_i^B}{c_i^B + c_i^B}$ for scaled and as $\mathbf{y}_i = \mathbf{y}_i^B + \mathbf{y}_i^B$ for non-scaled version. Variations of linear transformers, such as Performer (Choromanski et al., 2021), which employ different non-linearities $\phi(.)$ for keys and queries, can be adapted to a bidirectional format using the framework established for linear transformers.

**Retentive Network** (LION-RETNET). According to Sun et al. (2023) the forward equation for a retentive network can be written as:

$$\mathbf{S}_i^F = \lambda\mathbf{S}_{i-1}^F + \mathbf{k}_i \mathbf{v}_i^\top, \qquad (113)$$

$$\mathbf{y}_i^F = \mathbf{q}_i^\top \mathbf{S}_i^F \qquad (114)$$

This architecture can also be expanded to bi-directional setting simply by not scaling the attention in our framework and only using the mask with non input-dependent $\lambda_i = \lambda$ values:

$$\mathbf{S}_i^{F/B} = \lambda\mathbf{S}_{i-1}^{F/B} + \mathbf{k}_i \mathbf{v}_i^\top, \qquad (115)$$

$$\mathbf{y}_i^{F/B} = \mathbf{q}_i^\top \mathbf{S}_i^{F/B} - \frac{1}{2}\mathbf{q}_i^\top \mathbf{k}_i \mathbf{v}_i \qquad (116)$$

$$= \qquad \mathbf{Y} = (\mathbf{Q}\mathbf{K}^\top \odot \mathbf{M}^R)\mathbf{V} \qquad (117)$$

Note that: $\mathbf{M}_{ij}^R = \lambda^{|i-j|}$.

**xLSTM** (LION-LSTM). According to Beck et al. (2024) the recurrence for forward recurrence of xLSTM can be written as:

$$\mathbf{S}_i^F = f_i\mathbf{S}_{i-1}^F + i_i\mathbf{k}_i \mathbf{v}_i^\top, \qquad (118)$$

$$\mathbf{z}_i^F = f_i\mathbf{z}_{i-1}^F + i_i\mathbf{k}_i, \qquad (119)$$

$$\mathbf{y}_i^F = \frac{\mathbf{q}_i^\top \mathbf{S}_i^F}{\mathbf{q}_i^\top \mathbf{z}_i^F} \qquad (120)$$

The above recurrence is equivalent to (8) by considering $i_i\mathbf{k}_i$ as a new key. The term $i_i\mathbf{k}_i$ can be easily vectorized by aggregating all $i_i$ values for each token into a vector $\mathbf{i}$. Thus, we can express the vectorized form of the bidirectional xLSTM and its equivalence to attention as follows:

$$\mathbf{S}_i^{F/B} = f_i \mathbf{S}_{i-1}^{F/B} + i_i \mathbf{k}_i \mathbf{v}_i^\top, \tag{121}$$

$$\mathbf{z}_i^{F/B} = f_i \mathbf{z}_{i-1}^{F/B} + i_i \mathbf{k}_i \tag{122}$$

$$c_i^{F/B} = \mathbf{q}_i^\top \mathbf{z}_i^{F/B} - \frac{1}{2}\mathbf{q}_i^\top \mathbf{k}_i, \tag{123}$$

$$\mathbf{y}_i^{F/B} = \mathbf{q}_i^\top \mathbf{S}_i^{F/B} - \frac{1}{2}\mathbf{q}_i^\top \mathbf{k}_i \mathbf{v}_i \tag{124}$$

$$\text{Output: } \mathbf{y}_i = \frac{\mathbf{y}_i^F + \mathbf{y}_i^B}{\max(c_i^F + c_i^B, 1)} \tag{125}$$

$$= \qquad \mathbf{Y} = \text{SCALE'}(\mathbf{Q}(\mathbf{i} \odot \mathbf{K}^\top)) \odot \mathbf{M}^f)\mathbf{V} \tag{126}$$

where the mask $\mathbf{M}^f$ is equal to the LION mask (88) just by replacing $\lambda_i = f_i$. And where operation SCALE' consider the maximum of operation in the denominator as:

$$\text{SCALE'}(\mathbf{A})_{ij} = \frac{\mathbf{A}_{ij}}{\max(\sum_{j=1}^L \mathbf{A}_{ij}, 1)} \tag{127}$$

**Gated RFA** (LION-**GRFA**). Gated RFA (Yang et al., 2023) in autoregressive mode exhibits a recurrence similar to that of xLSTM, with only minor differences:

$$\mathbf{S}_i^F = g_i \mathbf{S}_{i-1}^F + (1 - g_i)\mathbf{k}_i \mathbf{v}_i^\top, \tag{128}$$

$$\mathbf{z}_i^F = g_i \mathbf{z}_{i-1}^F + (1 - g_i)\mathbf{k}_i, \tag{129}$$

$$\mathbf{y}_i^F = \frac{\mathbf{q}_i^\top \mathbf{S}_i^F}{\mathbf{q}_i^\top \mathbf{z}_i^F} \tag{130}$$

Thus, the bidirectional version of the model retains a similar output, achieved by replacing the vector $\mathbf{i}$ in (126) with $1 - \mathbf{g}$, where $\mathbf{g}$ represents the vectorized form of all scalar values $g_i$.

$$\mathbf{S}_i^{F/B} = g_i \mathbf{S}_{i-1}^{F/B} + (1 - g_i)\mathbf{k}_i \mathbf{v}_i^\top, \tag{131}$$

$$\mathbf{z}_i^{F/B} = g_i \mathbf{z}_{i-1}^{F/B} + (1 - g_i)\mathbf{k}_i \tag{132}$$

$$c_i^{F/B} = \mathbf{q}_i^\top \mathbf{z}_i^{F/B} - \frac{1}{2}\mathbf{q}_i^\top \mathbf{k}_i, \tag{133}$$

$$\mathbf{y}_i^{F/B} = \mathbf{q}_i^\top \mathbf{S}_i^{F/B} - \frac{1}{2}\mathbf{q}_i^\top \mathbf{k}_i \mathbf{v}_i \tag{134}$$

$$= \qquad \mathbf{Y} = \text{SCALE}(\mathbf{Q}((1 - \mathbf{g}) \odot \mathbf{K}^\top) \odot \mathbf{M})\mathbf{V} \tag{135}$$

## C.6 GENERATION OF THE MASK

Below we present the Python code used for the creation of the bidirectional mask $\mathbf{M}$ as described in previous sections.

```python
def mask_single_direction(tensor):
    # cumsum = cumulative_sum(tensor)   Definition
    prepend_zeros = zeros(tensor.shape[:-1], 1, dtype=tensor.dtype)
    cumsum = concatenate((prepend_zeros, cumsum), dim=-1)
    A = cumsum[..., 1:].unsqueeze(-2) - cumsum[..., :-1].unsqueeze(-1)
    A = lower_triangle(A.transpose(-1, -2))
    zero_row = zeros(A.shape[:-2], 1, A.shape[-1], dtype=A.dtype)
    A = concatenate((zero_row, A[..., :-1, :]), dim=-2)
    return lower_triangle(exp(A))

def mask_bidirection(vec):
```

```
12    vec_shape = vec.shape
13    A_for = mask_single_direction(vec.unsqueeze(-1).transpose(-1, -2)).
      squeeze()
14    vec_back = concatenate((vec, ones((vec_shape[0], vec_shape[1], 1))),
      dim=-1)
15    A_back = mask_single_direction(vec_back)[:,:,1:].unsqueeze(-1).
      transpose(-1,-2).squeeze()
16    return A_for + A_back - eye(A_for.shape[-1])
```

## C.7 EXPANDING THE DIMENSION OF $a_i$

Similar to other recurrent models, particularly SSM variations, the dimension of $a_i$ can be increased beyond a scalar. When $a_i$ is a scalar, the same mask $\mathbf{M}$ is applied to all elements of the value vector $\mathbf{v}$. However, if we allow $a_i$ to be a vector $\mathbf{a}_i \in \mathbb{R}^d$, the mask matrix transforms into a tensor $\bar{\mathbf{M}} \in \mathbb{R}^{L \times L \times d}$. This tensor can be computed in parallel for each individual value element along the last dimension. The last dimension will then be multiplied using the Hadamard product with the values, resulting in the following vectorized form:

$$\mathbf{Y} = \text{SCALE}(\mathbf{Q}\mathbf{K}^\top \odot \bar{\mathbf{M}}) * \mathbf{V} \tag{136}$$

In this equation, the operation $*$ denotes the Hadamard product applied along the last dimension of the tensor mask $\bar{\mathbf{M}}$ with the value vector $\mathbf{V}$, while the first two dimensions are combined using a standard matrix product. The corresponding code is as follows:

```
1 attn = (Q @ K.transpose(-2, -1))
2 attn = torch.einsum("nhkmd,nhkm->nhkmd", M, attn)
3 attn = scale(attn)
4 x = torch.einsum("nhkmd,nhmd->nhkd", attn, V)
```

## C.8 CHANGING THE ORDER OF PATCHES

When processing images, both the spatial relationships among neighboring pixels and their positions are as critical as the pixel values themselves. Positional embeddings provide a way to incorporate these spatial relationships. A common approach in Transformers involves flattening the image, as illustrated in the left panel of Figure 7. However, we argue that this method of flattening is suboptimal and can be enhanced to include additional contextual information.

Furthermore, in scenarios involving a fully masked setup or RNN-based inference, the sequence in which pixels are processed becomes increasingly important. To address this, we propose a new reordering scheme for pixel values. In the attention module, the pixel values are reordered following the patterns depicted in the center and right panels of Figure 7. Forward and backward passes are then executed based on this new ordering, adhering to established procedures. The outputs from these two passes are subsequently averaged to generate the final result.

We refer to this method as LION-S (v2) throughout the paper. This approach demonstrated a notable improvement in accuracy for image classification tasks while maintaining the efficiency and flexibility inherent to the method. A similar concept has been previously explored in Vision-LSTM (Alkin et al., 2024).

# D ADDITIONAL EXPERIMENTAL VALIDATION

## D.1 CITATIONS FOR LRA BENCHMARKS

The LRA baselines included in Table 2 correspond to Transformer (Vaswani et al., 2017), MEGA and MEGA-chunk (Ma et al., 2022), DSS (Gupta et al., 2022), S4 (Gu et al., 2022), S5 (Smith et al., 2023), Mamba (Gu & Dao, 2024), Local Att. (Vaswani et al., 2017), Sparse Transformer (Child et al., 2019), Longformer (Beltagy et al., 2020), Linformer (Wang et al., 2020), Reformer (Kitaev et al.,

 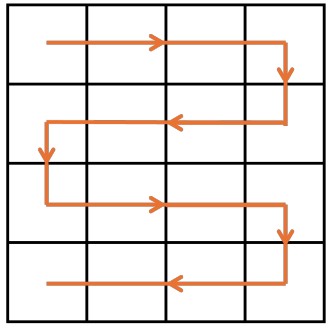 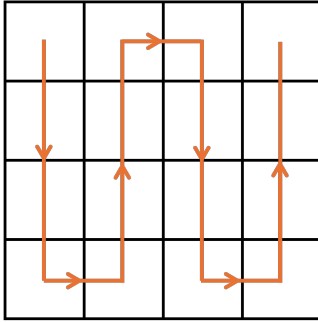

Figure 7: *Reordering of patches.* Left is the naive approach to flatten images, also used in LION-S. Center and right figures are the new approaches applied in LION-S (v2) to consider further spatial information.

2020), Sinkhorn Transformer (Tay et al., 2020a), BigBird (Zaheer et al., 2020), Linear Transformer (Katharopoulos et al., 2020), Performer (Choromanski et al., 2021), FNet (Lee-Thorp et al., 2022), Nyströmformer (Xiong et al., 2021), Luna-256 (Ma et al., 2021) and H-Transformer-1D (Zhu & Soricut, 2021).

### D.2 LRA CONFIGURATIONS FOR LION

For the LRA task, we utilized the same model dimensions as specified in the S5 (Smith et al., 2023) paper, following the guidelines from the S5 GitHub repository[3]. Our state matrix was represented as a vector $\Lambda_i = \lambda_i$, where each element contains a scalar non-input dependent value $e^a$. The value $a$ was initialized based on *HIPPO* theory, alongside the input-dependent $a_i$, as described in main body.

We employed the ADAMW optimizer with an initial learning rate of $5 \times 10^{-4}$ and a cosine learning rate scheduler (Loshchilov & Hutter, 2016). The weights for the queries and keys, as well as the selective component of $\Lambda$, were initialized using a Gaussian distribution with a standard deviation of 0.1. For the values $\mathbf{v}$, we initialized $W_{\mathbf{v}}$ using zero-order hold discretization, represented as $W_{\mathbf{v}}^{\text{init}} = \left(\Lambda^{-1} \cdot (\Lambda - \mathbf{I})\right)$. The non-selective parts of $\Lambda$ were initialized based on the *HIPPO* (Smith et al., 2023) matrix.

### D.3 ABLATION STUDIES ON LRA DATASET

Table 6: *Effects of different parameter choices and non-linearities in* LION-S *on LRA tasks.* Codes: [1] Sigmoid non-linearity was applied to the $\mathbf{k}$ and $\mathbf{q}$ values with unscaled masked attention; [2] ReLU non-linearity was utilized, and the masked attention was scaled; [3] The parameter $a_i$ was selected as a scalar instead of a vector; [4] LION-S model parameters were used without scaling; [5] The attention matrix of LION-S was scaled, but attention values were adjusted without the factor of $\lambda_i$; [6] The selective component of $a_i$ was removed; [7] SoftPlus activation function was employed for the $a_i$ values.

| Model (input length) | ListOps 2048 | Text 2048 | Retrieval 4000 | Image 1024 | Pathfinder 1024 | PathX 16K | Avg. |
|---|---|---|---|---|---|---|---|
| [1] $\phi(x) = \sigma(x)$ w.o scaling | 61.02 | 88.02 | 89.10 | 86.2 | 91.06 | 97.1 | 85.41 |
| [2] $\phi(x) = \text{RELU}(x)$ w. scaling | 36.37 | 65.24 | 58.88 | 42.21 | 69.40 | ✗ | 54.42 |
| [3] $a_i$ only scalar | 36.23 | 60.33 | 60.45 | 58.89 | 70.00 | ✗ | 57.17 |
| [4] LION w.o scaling | 58.76 | 67.22 | 59.90 | 60.0 | 65.51 | ✗ | 62.27 |
| [5] scaled attention w.o mask | 60.12 | 87.67 | 87.42 | 88.01 | 89.23 | ✗ | 82.49 |
| [6] $a_i$ From *HIPPO* w.o selectivity | 60.12 | 88.00 | 89.22 | 83.21 | 91.0 | 96.30 | 84.64 |
| [7] $a_i = \text{SOFTPLUS}(x)$ | 16.23 | 59.90 | 60.00 | 45.12 | 70.07 | ✗ | 50.26 |
| LION-S | **62.25** | **88.10** | **90.35** | **86.14** | **91.30** | **97.99** | **86.07** |

---

[3]https://github.com/lindermanlab/S5

We have observed that bounding the keys and queries significantly enhances the model's ability to solve tasks. This finding is consistent with the observations in Yang et al. (2024). As demonstrated in variation [1], it can successfully tackle the LRA task even without scaling, while the RELU activation fails to do so. Additionally, we found that scaling plays a crucial role, particularly when it comes to scaling the masked attention. The approach used in LION, which scales the attention before applying the mask expressed as $\mathbf{Y} = \text{SCALE}(\mathbf{QK}^\top) \odot \mathbf{M}$ has proven ineffective in addressing the challenging PathX task, as shown in [5]. Furthermore, the modifications implemented in LION-S have demonstrated superior performance compared to all other variations tested.

## D.4  ADDITIONAL EXPERIMENTAL RESULTS FOR THE MLM/GLUE TASKS

In this section and in Table 7, we present our bidirectional language task results in the BASE scale using the BERT pretraining and BERT24 (Izsak et al., 2021) finetuning recipes.

Table 7: *C4 Masked Language Modelling and GLUE results with the BERT pretraining and BERT24 finetuning recipes.* For each column (dataset), the best and the second best results for each model size are highlighted with **bold** and underline respectively.

| Model | MLM Acc. | MNLI | RTE | QQP | QNLI | SST2 | STSB | MRPC | COLA | Avg. |
|---|---|---|---|---|---|---|---|---|---|---|
| BERT | **67.23** | **84.26** | **59.21** | **89.87** | **90.24** | 92.35 | **88.12** | **90.24** | **56.76** | **81.38** |
| LION-LIT | 65.08 | 82.37 | 55.81 | 89.49 | 89.57 | 91.74 | 86.27 | 88.25 | 44.46 | 78.50 |
| LION-S | 66.19 | 82.50 | 57.47 | 89.38 | 87.88 | **92.70** | 82.42 | 82.46 | 53.39 | 78.40 |

## D.5  EXPERIMENTAL DETAILS FOR THE MLM/GLUE TASKS

**Architectures** We train the BASE (110M parameters) and LARGE (336M parameters) model families from the original BERT paper (Devlin et al., 2019). For the LION models, we replace the standard self-attention blocks with LION-LIT/LION-RETNET/LION-S blocks while keeping all hyperparameters the same. For LION-LIT, we incorporate LayerNorm (Ba, 2016) after the attention block to enhance stability. Our implementation is based on the M2 repository (Fu et al., 2023), i.e., `https://github.com/HazyResearch/m2`.

**Pretraining** All our pretraining hyperparameters follow Fu et al. (2023): We employ the C4 dataset (Dodge et al., 2021), a maximum sequence length during pretraining of 128 and a masking probability of 0.3 and 0.15 for the training and validation sets respectively. We train our model for 70,000 steps with a batch size of 4096. We employ the decoupled AdamW optimizer with a learning rate of $8 \cdot 10^{-4}$, $\beta_1 = 0.9$, $\beta_2 = 0.98$, $\epsilon = 10^{-6}$ and weight decay $10^{-5}$. As a scheduler, we perform a linear warm-up for 6% of the training steps and a linear decay for the rest of training until reaching 20% of the maximum learning rate.

Our only change in the pretraining hyperparameters is setting the learning rate to $2 \cdot 10^{-4}$ for the LARGE model family. In our preliminary experiments, we found that training diverged when using a learning rate of $8 \cdot 10^{-4}$ for BERT-LARGE.

For completeness, in Table 7 we present the results with the BERT pretraining[4] and BERT 24 finetuning[5] recipes available in the M2 repository.

**Finetuning** For the GLUE finetuning experiments, we employ four different configurations:

- **BERT24**: Available in Izsak et al. (2021) and the file `https://github.com/HazyResearch/m2/blob/main/bert/yamls/finetune-glue/hf-transformer-finetune-glue-bert-base-uncased.yaml`.

---

[4]`https://github.com/HazyResearch/m2/blob/main/bert/yamls/pretrain/hf-transformer-pretrain-bert-base-uncased.yaml`

[5]`https://github.com/HazyResearch/m2/blob/main/bert/yamls/finetune-glue/hf-transformer-finetune-glue-bert-base-uncased.yaml`

Table 8: GLUE finetuning recipes employed in this work. All recipes finetune on RTE, STSB and MRPC from the weights finetuned in MNLI and the rest from the C4-pretrained weights. All recipes use a sequence length of 128 tokens except BERT24, that uses 256. D. AdamW stands for decoupled AdamW.

| Recipe | Param. | Dataset | | | | | | | |
|---|---|---|---|---|---|---|---|---|---|
| | | MNLI | QNLI | QQP | RTE | SST2 | MRPC | COLA | STSB |
| BERT24 (Izsak et al., 2021) | LR | $5 \cdot 10^{-5}$ | $1 \cdot 10^{-5}$ | $3 \cdot 10^{-5}$ | $1 \cdot 10^{-5}$ | $3 \cdot 10^{-5}$ | $8 \cdot 10^{-5}$ | $5 \cdot 10^{-5}$ | $3 \cdot 10^{-5}$ |
| | WD | $5 \cdot 10^{-6}$ | $1 \cdot 10^{-5}$ | $3 \cdot 10^{-6}$ | $1 \cdot 10^{-6}$ | $3 \cdot 10^{-6}$ | $8 \cdot 10^{-5}$ | $5 \cdot 10^{-6}$ | $3 \cdot 10^{-6}$ |
| | Epochs | 3 | 10 | 5 | 3 | 3 | 10 | 10 | 10 |
| | Optimizer | D. AdamW | D. AdamW | D. AdamW | D. AdamW | D. AdamW | D. AdamW | D. AdamW | D. AdamW |
| M2-BASE (Fu et al., 2023) | LR | $5 \cdot 10^{-5}$ | $5 \cdot 10^{-5}$ | $3 \cdot 10^{-5}$ | $1 \cdot 10^{-5}$ | $3 \cdot 10^{-5}$ | $8 \cdot 10^{-5}$ | $8 \cdot 10^{-5}$ | $8 \cdot 10^{-5}$ |
| | WD | $5 \cdot 10^{-6}$ | $1 \cdot 10^{-5}$ | $3 \cdot 10^{-6}$ | $1 \cdot 10^{-6}$ | $3 \cdot 10^{-6}$ | $8 \cdot 10^{-5}$ | $5 \cdot 10^{-6}$ | $3 \cdot 10^{-6}$ |
| | Epochs | 3 | 10 | 5 | 3 | 3 | 10 | 10 | 10 |
| | Optimizer | D. AdamW | D. AdamW | D. AdamW | D. AdamW | D. AdamW | D. AdamW | D. AdamW | AdamW |
| M2-LARGE (Fu et al., 2023) | LR | $5 \cdot 10^{-5}$ | $5 \cdot 10^{-5}$ | $3 \cdot 10^{-5}$ | $5 \cdot 10^{-5}$ | $3 \cdot 10^{-5}$ | $8 \cdot 10^{-5}$ | $5 \cdot 10^{-5}$ | $8 \cdot 10^{-5}$ |
| | WD | $5 \cdot 10^{-6}$ | $1 \cdot 10^{-6}$ | $3 \cdot 10^{-6}$ | $1 \cdot 10^{-6}$ | $3 \cdot 10^{-6}$ | $8 \cdot 10^{-6}$ | $1 \cdot 10^{-6}$ | $3 \cdot 10^{-5}$ |
| | Epochs | 3 | 10 | 5 | 2 | 3 | 10 | 10 | 8 |
| | Optimizer | D. AdamW | D. AdamW | D. AdamW | AdamW | D. AdamW | D. AdamW | D. AdamW | D. AdamW |
| Modified (Ours) | LR | $10^{-5}$ | $10^{-5}$ | $10^{-5}$ | $10^{-5}$ | $10^{-5}$ | $10^{-5}$ | $10^{-5}$ | $10^{-5}$ |
| | WD | $5 \cdot 10^{-6}$ | $1 \cdot 10^{-6}$ | $3 \cdot 10^{-6}$ | $1 \cdot 10^{-6}$ | $3 \cdot 10^{-6}$ | $8 \cdot 10^{-6}$ | $1 \cdot 10^{-6}$ | $3 \cdot 10^{-5}$ |
| | Epochs | 3 | 10 | 5 | 2 | 3 | 10 | 10 | 8 |
| | Optimizer | D. AdamW | D. AdamW | D. AdamW | AdamW | D. AdamW | D. AdamW | D. AdamW | D. AdamW |

Table 9: Combining positional embeddings with LION-RETNET and LION-S. Both pretrained models improve in the validation MLM acc. when employing positional embeddings.

| Model | Pos. Emb. | MLM Acc. | MNLI | RTE | QQP | QNLI | SST2 | STSB | MRPC | COLA | Avg. |
|---|---|---|---|---|---|---|---|---|---|---|---|
| LION-RetNet | ✗ | 66.62 | 82.85 | 52.49 | 89.63 | 88.43 | 91.86 | 85.96 | 83.94 | 53.58 | 78.59 |
| | ✓ | 66.97 | 83.37 | 54.08 | 89.52 | 88.32 | 92.35 | 83.58 | 79.40 | 54.53 | 78.15 |
| LION-s | ✗ | 67.05 | 83.17 | 53.50 | 89.35 | 88.89 | 93.00 | 37.73 | 77.87 | 53.18 | 72.09 |
| | ✓ | 67.35 | 83.26 | 52.42 | 89.82 | 88.38 | 92.58 | 83.87 | 79.54 | 55.25 | 78.14 |

- **M2-BASE**: Available in Fu et al. (2023), Section C.1 and the file `https://github.com/HazyResearch/m2/blob/main/bert/yamls/finetune-glue/monarch-mixer-finetune-glue-960dim-parameter-matched.yaml`.
- **M2-LARGE**: Available in Fu et al. (2023), Section C.1 and the file `https://github.com/HazyResearch/m2/blob/main/bert/yamls/finetune-glue/monarch-mixer-large-finetune-glue-1792dim-341m-parameters.yaml`.
- **Modified**: Same as M2-LARGE but all learning rates are set to $10^{-5}$.

The recipes are summarized in Appendix D.5. The Modified hyperparameter set was devised as M2-LARGE was found to diverge for BERT-LARGE.

## D.6 ABLATION STUDIES IN THE MLM/GLUE TASKS

**Combining positional embeddings with LION.** We compare the GLUE performance of LION-RETNET and LION-S when including positional embeddings. We pretrain the BASE models and finetune them with the M2-BASE recipe.

In Table 9 we can observe that adding positional embeddings increased the MLM acc. in around 0.3 percentage points. In the GLUE benchmark, we observe that for LION-RETNET performance degraded in 0.44 percentage points, while for LION-S, performance improved in 6.05 percentage points. We attribute this behavior in GLUE to the dependence on the finetuning recipe.

**Recipe selection.** In this section, we select the best finetuning recipe for each model family and size. For the BASE models, we test the M2-BASE and Modified recipes. For the LARGE models, we test the M2-LARGE and Modified recipes.

In Table 10, firstly, we observe that the M2-BASE recipe generally provides a higher GLUE score than the Modified recipe for the BASE models, e.g., 82.25 v.s. 80.26 for the BERT model. Secondly, we observe that for the LARGE model family, the M2-LARGE recipe fails, providing

Table 10: Recipe selection for the GLUE benchmark.

| Model | MLM Acc. | Recipe | MNLI | RTE | QQP | QNLI | SST2 | STSB | MRPC | COLA | Avg. |
|---|---|---|---|---|---|---|---|---|---|---|---|
| BERT | 67.70 | M2-BASE | 84.63 | 64.33 | 89.99 | 89.80 | 92.51 | 86.69 | 89.62 | 60.42 | 82.25 |
| | | Mod. | 83.09 | 58.27 | 89.35 | 89.88 | 92.16 | 86.56 | 87.78 | 55.02 | 80.26 |
| LION-LIT | 65.47 | M2-BASE | 82.50 | 63.47 | 89.72 | 89.27 | 91.74 | 87.18 | 89.37 | 49.22 | 80.31 |
| | | Mod. | 80.88 | 54.95 | 88.80 | 88.83 | 91.32 | 85.42 | 87.07 | 46.98 | 78.03 |
| LION-RETNET | 66.62 | M2-BASE | 82.85 | 52.49 | 89.63 | 88.43 | 91.86 | 85.96 | 83.94 | 53.58 | 78.59 |
| | | Mod. | 80.52 | 52.85 | 88.93 | 88.36 | 91.55 | 82.05 | 84.48 | 49.13 | 77.23 |
| LION-S | 67.05 | M2-BASE | 83.17 | 53.50 | 89.35 | 88.89 | 93.00 | 37.73 | 77.87 | 53.18 | 72.09 |
| | | Mod. | 78.14 | 56.39 | 88.68 | 88.52 | 92.39 | 51.22 | 77.60 | 49.75 | 72.84 |
| BERT$_{LARGE}$ | 69.88 | M2-LARGE | 84.97 | 69.10 | 31.59 | 49.15 | 91.93 | 53.61 | 87.87 | 51.16 | 64.92 |
| | | Mod. | 85.68 | 67.44 | 89.90 | 91.89 | 93.04 | 88.63 | 90.89 | 56.14 | 82.95 |
| LION-LIT $_{LARGE}$ | 67.11 | M2-LARGE | 83.20 | 54.51 | 89.08 | 84.90 | 90.44 | 68.57 | 85.25 | 23.35 | 72.41 |
| | | Mod. | 83.73 | 57.18 | 89.85 | 89.93 | 91.86 | 88.02 | 90.18 | 55.36 | 80.76 |
| LION-RETNET $_{LARGE}$ | 68.64 | M2-LARGE | 83.82 | 52.85 | 41.48 | 53.67 | 91.13 | 36.87 | 82.41 | 45.79 | 61.00 |
| | | Mod. | 83.82 | 60.72 | 89.72 | 89.79 | 92.93 | 87.29 | 89.66 | 56.83 | 81.34 |
| LION-S $_{LARGE}$ | 69.16 | M2-LARGE | 83.71 | 50.04 | 38.81 | 53.98 | 91.59 | 36.98 | 82.29 | 50.27 | 60.96 |
| | | Mod. | 84.38 | 57.69 | 89.57 | 90.30 | 92.93 | 87.68 | 90.57 | 59.54 | 81.58 |

poor performances between 60.96 and 72.41 GLUE points. When reducing the learning rate to $10^{-5}$ (Modified recipe), training is more stable and performance reaches between 80.76 and 82.95 GLUE points. We find that small changes in the finetuning recipe have a large effect in the performance. Our results in standard recipes show that the LION family of models can obtain a high performance without extensive tuning and closely follow the performance of the BERT family models, at 80.31 v.s. 82.25 for the BASE model size and 81.58 v.s. 82.95 for the LARGE model size.

### D.7  ABLATION STUDIES WITH IMAGE CLASSIFICATION

**Resolution vs. Accuracy.** The most common practice in the literature on Vision Transformers is to resize images to $224 \times 224$ even though most of the images in the ImageNet dataset are larger. Since regular Transformers have positional embedding, it is not possible to use a larger resolution during inference than the training. However, since the LION-S architecture does not include any positional embeddings, it can be used with different resolutions. In Figure 8, we present the accuracy of the architectures trained on $224 \times 224$ resolution on the ImageNet dataset at different inference resolutions. As the results illustrate, the abilities of LION-S can be effectively transferred among different resolutions.

**Choice of $\lambda_i$ values.**

In this section, we study the properties of the selectivity parameter $a_i$ on CIFAR-100 dataset. We tested, three cases: (*i*) fixed mask with scalars $a_i = a^i$, (*ii*) vector, input-dependent $\mathbf{a}_i \in \mathbb{R}^d$ (*cf.*, Appendix C.7) and iii) input dependent scalar $\mathbf{a}_i \in \mathbb{R}$. The results, presented in Table 11, show that while the input dependency is beneficial, the expansion of $\mathbf{a}_i$ is not necessary for image tasks. As a result, we employ option three in all image classification tasks, and the end model is called LION-S.

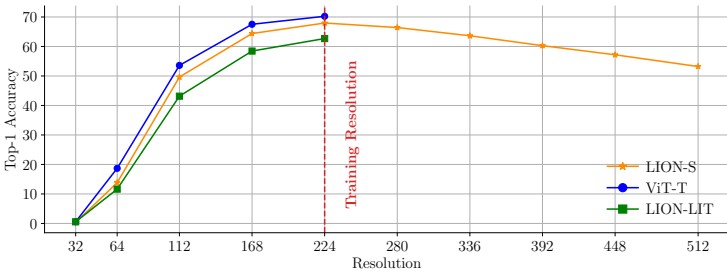

Figure 8: *Top-1 accuracy on Imagenet of the models at different resolutions.* Images are resized at the corresponding resolution and fed into the model. Due to positional embeddings, ViT and LION-LIT models cannot perform with sizes larger than the training size while LION-S can preserve the accuracy for much higher resolutions.

Table 11: *Ablation studies on image classification.* Additional ablations with CIFAR100 dataset to determine the size and input dependency of the selectivity parameter of the model LION-S.

| Models | Top-1 Acc. |
|---|---|
| Fixed mask $a_i = a^i$ | 75.66 |
| Vector $\mathbf{a}_i \in \mathbb{R}^d$ | 67.55 |
| Scalar, input dependent $\mathbf{a}_i \in \mathbb{R}$ (LION-S) | **77.56** |

**Understanding the power of non-linearity, softmax, and positional embeddings.** In Table 12, we present additional ablations on certain design elements of a Vision Transformer. We perform these experiments on CIFAR-100 data using the same hyperparameters with LION-S. We have observed that either nonlinearity or softmax is essential for the model to converge with a nice accuracy. Though positional embedding boosts the accuracy, a mask can easily replace it.

Table 12: *Ablation studies on image classification.* Additional ablations with the CIFAR-100 dataset to understand the contribution of softmax, nonlinearities in a model is presented. Soft., PosEmb and NonLin expresses if softmax, positional embedding, and non-linearity have been applied. ✗ means the model did not converge. The ▢ symbol denotes the adaptation of recurrent models that achieve equivalence to attention during training while utilizing recurrence during inference, as established by our theorem.

| Models | Top-1 Acc. |
|---|---|
| [1] Soft. + PosEmb + NonLin | 73.88 |
| [2] Soft. + PosEmb (ViT-T) | 77.33 |
| [3] Soft. + NonLin | ✗ |
| [4] Soft. | 73.15 |
| [5] PosEmb + Non.Lin (LION-LIT) | 73.61 |
| [6] PosEmb | 68.54 |
| [7] NonLin | 65.28 |
| [8] Base | ✗ |
| Non.Lin + Mask (LION-S) | **77.56** |

Table 13: *Summary of training hyperparameters for image classification tasks.* Corresponding to variations of ViT recipe from Touvron et al. (2020). Nonlinearity is chosen based on performances.

| Models | ViT-T | | | LION-LIT | | | LION-S | | |
|---|---|---|---|---|---|---|---|---|---|
| Datasets | CIFAR-10 | CIFAR-100 | ImageNet | CIFAR-10 | CIFAR-100 | ImageNet | CIFAR-10 | CIFAR-100 | ImageNet |
| Epochs | 1000 | 1000 | 300 | 1000 | 1000 | 300 | 1000 | 1000 | 300 |
| Batch size | 128 | 128 | 3072 | 128 | 128 | 3072 | 128 | 128 | 3072 |
| Learning rate | 5e-4 | 0.001 | 0.003 | 5e-4 | 0.001 | 0.001 | 5e-4 | 0.01 | 0.002 |
| Weight decay | 0.05 | 0.05 | 0.3 | 0.05 | 0.05 | 0.3 | 0.05 | 0.05 | 0.3 |
| Warmup epochs | 5 | 5 | 3 | 5 | 5 | 3 | 5 | 5 | 4 |
| Warmup starting learning rate | 1e-6 | 1e-6 | 5e-4 | 1e-6 | 1e-6 | 5e-4 | 1e-6 | 1e-6 | 5e-4 |
| Dropout | 0 | 0.1 | 0.1 | 0 | 0.1 | 0.1 | 0 | 0.1 | 0.1 |
| Gradient Clip. | ✗ | 1.0 | 1.0 | ✗ | 1.0 | 1.0 | ✗ | 1.0 | 1.0 |
| Nonlinearity | softmax | softmax | softmax | elu() + 1 | elu() + 1 | elu() + 1 | $\frac{SILU(x)}{\|SILU(x)\|}$ | $\frac{SILU(x)}{\|SILU(x)\|}$ | $\frac{SILU(x)}{\|SILU(x)\|}$ |
| Optimizer | AdamW | AdamW | AdamW | AdamW | AdamW | AdamW | AdamW | AdamW | AdamW |
| Scheduler | Cosine | Cosine | Cosine | Cosine | Cosine | Cosine | Cosine | Cosine | Cosine |
| Minimum learning rate | 1e-5 | 1e-5 | 1e-5 | 1e-5 | 1e-5 | 1e-5 | 1e-5 | 1e-5 | 1e-5 |
| Drop path | 0.1 | 0.1 | 0.1 | 0.1 | 0.1 | 0.1 | 0.1 | 0.1 | 0.1 |
| Model EMA | ✓ | ✓ | ✓ | ✓ | ✓ | ✓ | ✓ | ✓ | ✓ |
| Model EMA decay | 0.99996 | 0.99996 | 0.99996 | 0.99996 | 0.99996 | 0.99996 | 0.99996 | 0.99996 | 0.99996 |
| Color jitter | 0.3 | 0.3 | 0.3 | 0.3 | 0.3 | 0.3 | 0.3 | 0.3 | 0.3 |
| Train interpolation | bicubic | bicubic | bicubic | bicubic | bicubic | bicubic | bicubic | bicubic | bicubic |

## D.8 HYPERPARAMETERS FOR TRAINING IMAGE CLASSIFIERS

In Table 13, we present the training hyperparameters for image classification tasks. All experiments were conducted in a single (CIFAR-10, CIFAR-100) or multiple (ImageNet) machines with NVIDIA A100 SXM4 80GB GPUs. The codes for training and evaluating the models are adapted from Touvron et al. (2020) and Wightman (2019).

## D.9 CALCULATION OF NUMBER OF FLOPS

Below we present a theoretical number of FLOPS used in the attention of vision transformers and LION-S during inference where $L$ is the resolution/context length and $D$ is the hidden dimension. Results show that while transformer has $\mathcal{O}(L^2 + LD^2)$ LION-S has $\mathcal{O}(LD^2)$. Note that in this calculation, the exponentials and other nonlinearities are considered as 1 FLOP whereas in reality, the Softmax introduces additional complexities. The same calculations should also apply to other bi-directional models.

The number of FLOPs in the one head of the one layer attention for a vision transformer:

- Calculating $\mathbf{Q}, \mathbf{K}, \mathbf{V}$: $6LD^2$,
- Attention $A = \mathbf{Q}\mathbf{K}^T$: $2L^2D$
- Softmax (assuming 1 FLOP for exp): $2L^2$
- Calculating $\mathbf{Y}$: $2L^2D$
- **TOTAL:** $L(6D^2 + 4LD + 2L)$

The number of FLOPs in the attention module for LION:

- Calculating $\mathbf{Q}, \mathbf{K}, \mathbf{V}, \lambda$: $6LD^2 + 2LD$,
- For each token in one forward/backward recurrence:
  - Updating $\mathbf{S}_i^{F/B}$: $3D^2$
  - Updating $\mathbf{z}_i^{F/B}$: $2D$
  - Calculating $c_i^{F/B}$: $4D + 2$
  - Calculating $\mathbf{y}_i^{F/B}$: $2D^2 + 4D + 1$
  - Total: $5D^2 + 10D + 3$
- L forward + backward recurrences: $2L(5D^2 + 10D + 3)$
- Calculating $\mathbf{Y}$: $2L(D + 1)$
- **TOTAL:** $L(16D^2 + 24D + 7)$

## D.10 DISTILLATION RESULTS OF LION-S

We have also used the same recipe from DeiT distillation Touvron et al. (2021) and distilled the RegNet network into LION-S. We observed that the distillation outperforms the original ViT-Tiny on the ImageNet dataset. The results are shown in the table below:

Table 14: *Distillation results of* LION-S.

| Models | Top-1 Acc. |
|---|---|
| LION-S | 67.95 |
| VIT-Tiny | 70.23 |
| LION-S (Distilled) | **70.44** |

## D.11 TRAINING TIME FOR DIFFERENT MODELS IN VISION EXPERIMENTS

In Table 15, we present the average time per epoch to train each model on the CIFAR-100 dataset with batch size 1024. The same set-up is used in all measurements.

## D.12 ABLATION STUDIES IN MAPPING OF AUTOREGRESSIVE MODELS TO LION FRAMEWORK

Building on the mapping of autoregressive models in Appendix C.5, we conducted additional experiments using LION-RETNET and LION-GRFA. Specifically, we modified the transformer block

Table 15: Training Time per Epoch for Different Models. Best in **bold** and second best is in *italic* form.

| Training Strategy (Model) | Time (s) /Epoch |
|---|---|
| Attention (VIT) | **24.6** |
| Attention (**LION-S**) | 35.8 |
| Attention (**LION-LIT**) | *26.6* |
| Parallel Scan (Hydra) | 43.4 |

of the VIT-Tiny model according to the proposed mapping and evaluated its performance on the CIFAR-100 dataset, maintaining the same training recipes as LION-S. The results, summarized in Table 16, demonstrate that the LION framework facilitates the seamless extension of other autoregressive models to a bi-directional setting, achieving strong performance without requiring additional hyperparameter tuning.

Table 16: *Mapping of autoregressive models to bidirectional setting with* LION *framework.*. These models benefit from the expansion to the bi-directional setting using the LION framework.

| Model | Top-1 Acc. |
|---|---|
| GRFA (Uni-directional) | 71.56 |
| LION-GRFA (Bi-directional) | 73.24 |
| RETNET (Uni-directional) | 72.24 |
| LION-RETNET (Bi-directional) | 75.66 |

### D.13 ABLATION STUDIES ON IMPORTANCE OF BI-DIRECTIONALITY ON IMAGE CLASSIFICATION

To highlight the importance of bi-directionality and demonstrate the versatility of the LION framework, we conducted additional experiments examining the processing directions of the blocks. We evaluated four settings: (i) all blocks process patches in the forward direction only (Forward), (ii) all blocks process patches in the backward direction only (Backward), (iii) odd-numbered blocks process patches in the forward direction while even-numbered blocks process them in the backward direction (Forward-Backward), and (iv) all blocks process patches in both directions (Bi-directional). The results reveal that incorporating both directions improves performance by approximately 4%, while full bi-directionality achieves a significant boost of up to 10%.

Table 17: Results for LION-S and LION-S (v2) with different directional settings on CIFAR-100. Incorporating both directions improves performance by approximately 4%, while full bi-directionality achieves a significant boost of up to 10%.

| Model | Top-1 Acc. |
|---|---|
| LION-S (Forward) | 71.08 |
| LION-S (Backward) | 69.61 |
| LION-S (Forward-backward) | 73.93 |
| LION-S **(Bi-directional)** | **77.56** |
| LION-S (v2) (Forward) | 70.24 |
| LION-S (v2) (Backward) | 70.42 |
| LION-S **(v2) (Bi-directional)** | **80.07** |

