# OpenReview forum: "LION: A bidirectional framework that trains like a Transformer and infers like an RNN"
_ICLR.cc/2025/Conference — Submitted to ICLR 2025_

### Official Review · Reviewer_3WDt · 2024-10-30

**Soundness:** 4
**Presentation:** 4
**Contribution:** 4
**Rating:** 8
**Confidence:** 4

**Summary:**

This paper introduces LION, a novel sequence-to-sequence framework that combines the bidirectionality and parallelized training of Transformers with fast inference of recurrent neural networks.
More specifically this paper at first shows how Linear Attention and Gated Linear attention variants (including Selective State Space Models, such as Mamba-2) can be reformulated as bi-directional RNNs, such that they can use a parallel masked matrix multiply formulation (e.g. Y=scale(QK^T * M) V ) for training similar to transformers while still using the recurrent formulation for inference.

Then, the paper proposes LION-S a novel Gated or Selective Linear Attention variant with a specific parameterization gate or selectivity parameter inspired by zero-order hold discretization and LION-LIT a bi-directional non-gated Linear Attention variant, where the selectivity parameter is set to 1.

Through extensive experiments on the Long Range Arena Benchmark, Masked-Language Modeling, Image classification and Causal-Language Modeling the paper demonstrates that their proposed recurrences LION-S and LION-LIT achieve competitive performance to state of the art Transformer and State Space Models.

**Strengths:**

This paper does an exceptionally good job in presenting the complex formulas and transformations in a very readable and understandable way. It also provides a very good overview of related work and provides lots of insights in relations between those works.
Their experiments are very extensive and the paper also provides careful ablation studies on different design decisions of their LION-S recurrence (e.g. Table 6, Table 7). The authors also provide inference efficiency experiments in the appendix.

I enjoyed reading this paper and believe that it is a good contribution, therefore I recommend to accept this paper and hope the authors can clarify the questions and unclarities below:

**Weaknesses:**

- There are only minor weaknesses such that the Causal Language Modeling and Vision experiments are rather small scale. Nevertheless, the paper demonstrates with these experiments the effectiveness of LION-S also on these compute-intense domains.

- The connection to the continuous domain of LION-S is rather loose and could be elaborated on further.

**Questions:**

- It seems that the paper often uses the terms “attention”, “attention mechanism”, “full-attention” or “attention-formulas” interchangeably, which makes it sometimes hard for reader to disambiguate from the context whether softmax-attention, linear attention or causal (linear) attention is meant (e.g. L. 187, or L. 923). It would improve the readability if the terminology is unified in the paper.

- L. 190 The terminology forward / backward path might be misleading, as the term backward path is usually used for computing the derivatives / delta errors. I would suggest to clarify this distinction at some point or rather use forward / backward recurrence always.
Out of curiosity: Why is it called LION?

- In Table 8, you apply a Non-linearity (elu) to q and k in addition to the LION-S mask. What is the default for LION-S? Only Mask and no non-linearity or mask and non-linearity?

- In equation 85, 88 and 101 is there the mask M missing?

- The exponential gating in the mLSTM (xLSTM) requires a stabilizer (or “max”) state in addition to S and z. See (eq. 15, 16, 17 and  74, 75, 80 in [2]). This is not explicitly addressed in 94 to 96, but the LION bidirectional reformulation should still be applicable.

- Have you tried using just the forward recurrence of LION-S, but reverse the order of the input sequence in consecutive blocks as it was done for example in “Vision-LSTM” [1] ? It would be interesting how this compares to the forward/backward LION-S. The same experiment could be also applicable to masked language modeling in Section 5.2.

[1] Alkin, B., Beck, M., Pöppel, K., Hochreiter, S., & Brandstetter, J. (2024). Vision-LSTM: xLSTM as Generic Vision Backbone. arXiv preprint arXiv:2406.04303

[2] Beck, M., Pöppel, K., Spanring, M., Auer, A., Prudnikova, O., Kopp, M., ... & Hochreiter, S. (2024). xLSTM: Extended Long Short-Term Memory. arXiv preprint arXiv:2405.04517.

---

> ### Author Response · Authors · 2024-11-17
> **Authors Rebuttal to Reviewer 3WDt**
>
> We are glad that you enjoyed our paper and found our contribution valuable to the community! We are happy that our illustrations were helpful and our experiments thorough. Below, we have addressed the remaining questions raised by the reviewer:
>
> ---
>
> **1) Causal Language Modeling and Vision experiments are rather small scale**
>
> We agree with the reviewer appreciate their comment that larger scale models would strenghten our papers. Since pre-training is a compute intensive task, our experiments focus on smaller scales. After the initial submission, we expanded our experiments from tiny models (5.5M) to small scale (21.7M) for vision and from base (110M) to large(340M)  models for causal language task. In the below tables and in Table 3-4 of our paper, you can find the results with these scaled models. In both cases, we observed great benefits with larger models. Our results show that as the model size increases, the gap between LION-S and the transformer architecture gets less and less insignificant.
>
> | Model | MLM Acc. | MNLI   | RTE   | QQP   | QNLI  | SST2  | STSB  | MRPC  | COLA  | Avg.   |
> |-|-|-|-|-|-|-|-|-|-|-|
> | BERT | **69.88** | **85.68** | **67.44** | **89.90** | **91.89** | **93.04** | **88.63** | **90.89** | *56.14* | **82.95** |
> | LION-LIT | 67.11 | 83.73 | 57.18 | *89.85* | 89.93 | 91.86 | *88.02* | 90.18 | 55.36 | 80.76 |
> | LION-S | *69.16* | *84.38* | *57.69* | 89.57 | *90.30* | *92.93* | 87.68 | *90.57* | **59.54** | *81.58* |
>
> | Model| ImageNet|
> |-|-|
> | ViT-S | **72.19**|
> | **LION-LIT** (SMALL) | 69.62            |
> | **LION-S** (SMALL)| _70.86_          |
>
> We believe that these new results further proves the capabilities of the LION architecture.
>
> ---
>
> **2) Continuous domain of LION-S is rather loose**
>
> Thank you for the comment. As also requested by Reviewer sped, we have added details in Appendix Section B.6 explaining how discretization from the continuous domain using zero-order hold filtering leads to LION-S.
>
> ---
>
> **3) Terminology: "attention" and "backwards path"**
>
> Thanks for the suggestion, we have modified the paper to use a unified terminology for attention and to refer backward/forward path as backward/forward recurrence.
>
> We appreciate your curiosity regarding the name "LION." It was chosen to reflect **LI**near Attenti**ON**, and it aligns with the naming conventions used in the community, such as HIPPO, Mamba, Griffin, and others.
>
> ---
>
> **4) Default non-linearity for LION-S? Only Mask and no non-linearity or mask and non-linearity?**
>
> In Table 8, the applied non-linearity is elu()+1 for LION-LIT and silu normalized for LION-S on top of the mask.
>
> ---
>
> **5) Mask $M$ for Linear Transformer Equations 109 and 112 in new pdf version**
>
> Thank you for the detailed notice. The mask in Equation 101 of the original manuscript was missing, and we have now revised this equation for xLSTM, including the extension with the maximum operator as well.
>
> Since the Equations 109 and 112 refer to the Linear Transformer with and without scaling, this model does not use the selective parameter $\lambda_i$; in other words, $\lambda_i = 1$, which results in a mask of all ones. Since the Hadamard product of the mask with the attention matrix does not alter the attention, the scaling does not affect the overall computation.
>
> ---
>
> **6) xLSTM max stabilizer**
>
> We thank you for your careful notice. We have updated our theorem for the xLSTM extension into the bidirectional setting by incorporating the maximum operator in the denominator for scaling in the Appendix C5 line 126 of our new pdf.
>
> ---
>
> **7) Using just forward recurrence but reversing sequence order**
>
> Thanks for the suggestion, we have not tried input sequence reversing but will include it in the final version of the paper.
>
> Regarding Vision-LSTM [1], we considered a similar mechanism to traversal paths to improve locality in masking which we called LION-S (v2) which improved results. For further details of our patch reordering approach, please take a look at Appendix C.8.
>
> | Model | CIFAR-10 | CIFAR-100 | ImageNet |
> |-|--|-|-|
> | ViT-T | *92.84* | 77.33 | **70.23**|
> | **LION-LIT**| 90.05| 73.61| 62.69 |
> | **LION_S** | 93.25       | *77.56*| 67.95|
> | **LION-S (v2)**       | **94.77** | **80.07**| *69.22*|
>
> We also considered larger sizes of the model which we call SMALL using the VIT-S dimensions. For reference, ViT-S is 21.7M while ViT-T is 5.5M. Please refer to the table in our reply to your first concern.
>
> **References:**
>
> [1] Alkin, B., Beck, M., Pöppel, K., Hochreiter, S., & Brandstetter, J. (2024). Vision-LSTM: xLSTM as Generic Vision Backbone. arXiv preprint arXiv:2406.04303
>
> ---
>
> Once again, we would like to express our gratitude to the reviewer for their thoughtful comments and positive feedback. We are pleased to hear that you enjoyed reading our paper and hope that we have addressed all of your concerns. In case of any concerns remain, we would be more than happy to respond.

---

### Official Review · Reviewer_sped · 2024-11-03

**Soundness:** 2
**Presentation:** 2
**Contribution:** 2
**Rating:** 3
**Confidence:** 4

**Summary:**

LION a bidirectional sequence-to-sequence framework that casts attention (Transformers) as RNNs.
The authors propose LION-S, a model which combines LION with Selective SSMs.
LION-S achieves competitive performance with state-of-the-art vision and masked language transformer models while being more resource efficient.

**Strengths:**

The paper's illustrations and equations are written in a way that makes it easier to understand. The colour coding is a nice touch.

**Weaknesses:**

The paper proposes LION emphasizing that the framework casts bidirectional Transformers as bidirectional RNNs.
The paper then introduces LION-S (a model based on state-space models) and focuses on comparisons with Transformers showing efficiency gains and substantial performance improvements. However, this is misleading since the framework LION and LION-S can be viewed simply as SSMs.

By viewing LION as simply SSMs, the claimed benefits of their proposed transformers are not surprising and are common in prior SSMs. For example, (1) "does not require any additional positional encoding, enabling it to extrapolate beyond the context length or resolution during inference.". (2) competitive performance and significantly more efficient than transformers. (3) Solving the Long Range Arena (LRA) task.

More concretely, consider Theorem 3.3 $y_i$'s numerator and denominator:
$$
	y_i^F + y_i^B = q_i^T (S_i^F + S_i^B - k_i v_i)
$$

$$
	c_i^F + c_i^B = q_i^T (z_i^F + z_i^B - k_i)
$$

$S^F, S^B, z^F, z^B$ are simply SSMs. Both $(S^F, z^F)$ and $(S^B, z^B)$ are computable with forward and backward parallel scans respectively.
Essentially the numerator is two SSMs (1 forward and 1 backward) summed together ensuring bidirectionality. The denominator is also two SSMs summed together.

As a result, several experimental results are misleading. LION-S is grouped with "Transformer as Linear Recurrent Model" but it's essentially sums of SSMs with scaling. Viewing it as such, LION-S's similar performance to S4 and S5 is not surprising. Furthermore, its efficiency gains over Transformers are not surprising either.
Notably, Figure 3 (right) efficiency gains are normal since it's essentially comparing SSMs with a Transformer. A more fair comparison in terms of performance efficiency would be to compare with Vision SSM models such as Vision Mamba.



Some other concerns I have are as follows:
 -  In Table 2, S5 results are worse than S4, and Mamba results are very poor compared to S4 and S5. However, one would expect Mamba and S5 to outperform S4 as these methods improved upon S4. It is very surprising to see Mamba perform so poorly and S5's performance significantly worse than what is reported in the original paper. Justification for these results is needed.
 - The paper mentions "Transformers can be represented as SSMs," citing the Mamba-2 paper => This statement is imprecise. The Mamba-2 paper mentions in its footnote: "Technically speaking, these connections only relate to certain flavors of attention." The original Transformer (using Softmax) cannot be represented with a single SSM as it does not have a linear relationship.
 - LION-LIT results are missing from Table 2.
 - LSTM and GRU are described in Table 1 as having bidirectionality. These methods are not bidirectional by default unless stacked in different directions. With a similar logic, the various modern RNNs such as linear transformers and state-space models are also bidirectional by stacking them in different directions. This has already been done in several works such as the S5 paper for their experiments.
 - The paper mentions "SSMs often fall short in tasks requiring dense information processing". This is due to their bottleneck due to also being viewable as an RNN. The way the introduction is framed suggests to me that since LION is based on Transformers, it won't have this issue. However, LION should also have this issue since they are also viewable as RNNs (and are essentially being SSMs in different directions summed together).

**Questions:**

The paper focuses on LION-S as the main contributing method. However, to my understanding, LION-S would use the parallel scan algorithm. However, Figure (1) middle illustrates a matrix formulation during training instead. Could you clarify this?

Long Range Arena's Text dataset is described with an input length of 2048 in Table 2, but in previous papers such as in "S5", the dataset is described with an input length of 4096. Could you clarify if it is indeed different?

Are 27a and 27b coefficients supposed to be $\exp(a_i)$ and $(1 - \exp(a_i))$ instead of $\exp(a_i)$ and $(\exp(a_i) - 1)$? Since $\exp(a_i) = \exp(\log(\sigma(w^T_a x_i + b))) = \sigma(w^T_a x_i + b)$, we have the sum of the two coefficients summing to $1$, ensuring an attention mask. If not, it's not clear to me why the second coefficient would be $(\exp(a_i) - 1)$.

Considering the similarities between LION(-S) with SSMs, what are the conceptual and practical differences compared with existing SSM approaches? In addition, what is the practical benefit of using LION(-S) over these SSMs?

Considering the poor performances of S5 and Mamba, could you explain these discrepancies and provide details on their implementation or experimental setup?

Could you include LION-LIT results or explain why they were omitted in Table 2?

Since the paper labels LSTMs and GRUs as bidirectional when they are traditionally unidirectional, could you include the definition of bidirectionality used in the paper and how it applies consistently across the different models in the table?

---

> ### Author Response · Authors · 2024-11-17
> **Authors Rebuttal to Reviewer sped (1/4)**
>
> Thank you for your review. We believe there has been a misunderstanding regarding the contribution of LION and its training paradigm, which have led to a misevaluation of our work. We have clarified LION's contribution and training approach, and below, we address the reviewer’s comments:
>
>
> ### Misunderstanding of Our Main Contribution
>
> Thank you for the review and the concerns raised. We would like to clarify that the key contribution of our paper and the LION framework has been misunderstood. LION and its variants (such as LION-S) **train like Transformers and do not use parallel scan**. This contribution is clearly reflected in our **title, Figure 1, Table 1, Section 3**, and elsewhere in the paper. Below, we address this point along with other concerns raised by the reviewer:
>
> ---
>
> **Concern 1: "LION can be viewed simply as SSMs, LION-S is two SSMs summed together":**
>
> **LION is not simply an SSM**; it is a framework where training happens with attention (**Theorem 3.3 Eq 25**, i.e., $Y = SCALE(QK^\top ⊙ M)V$) and inference is performed using the equivalent RNN (**Theorem 3.3 Eq 20-24**) in a **bidirectional setting**. Additionally, **LION-S is not simply two SSMs summed together**, as this summation does not equate to full attention (as shown in Figure 2, parts b1 and b2, and Equation 7 of our paper). Instead, LION-S combines our framework with a specific selectivity (inspired by RFA[3] and Mamba-2 [4]), resulting in **a model equivalent to full masked attention**.
>
> Based on the connection outlined in [1], which allows causal Linear Transformers to be trained using attention and inferred as equivalent RNNs (as presented in Equations 4a to 4c of our paper), these models were introduced prior to the development of SSMs [7], and specifically before S5, which introduced parallel scan for SSMs. Causal Linear Transformers and all their variants use Transformer-style training and RNN inference, as demonstrated in Proposition 3.2 of our paper.
>
>
> Moreover, we would like to highlight that the equation provided by the reviewer:
>
> $$ y_i^F + y_i^B = q_i^\top (S_i^F + S_i^B - k_i v_i) $$
>
> poses a problem because the outputs for the forward and backward recurrences are not available for token $i$ at the same time. Therefore, this equation cannot be simply calculated unless all the $q_i$, $k_i$, and $v_i$ values are stored in memory, which contradicts the fundamental motivation behind using SSMs and RNNs (please look at our **response to reviewer tj5X 1st question**). Additionally, the term $k_i v_i$ is mathematically incorrect, as these two vectors cannot be multiplied using matrix multiplication. It is also not a dot product, as it should be a matrix representation to correctly express linear attention [1].
>
> ---
>
> **Concern 2: Difference between LION and SSMs**
>
> LION shares similar properties with mentioned Linear Transformer variants (reffered as Transformers as Linear Recurrent Model in Table 2) and has key differences compared to SSMs, as shown below:
>
> 1) Linear Tranformers, **including LION**, use attention matrix parallelization during training, whereas SSMs use the parallel scan algorithm.
> 2) Linear Tranformers, **including LION**, are presented as RNNs with matrix-valued hidden states $S^{F/B}_i \in \mathbb{R}^{d \times d}$ during inference, while SSMs have vector-valued hidden states $h^{F/B}_i \in \mathbb{R}^{d}$.
> 3) Linear Tranformers, **including LION**, use multi-head attention, while SSMs employ state expansion.
>
> Based on these differences, models like **Linear Transformer**\[1\], **RFA**\[3\], and **RetNet**\[2\] can be considered distinct from SSMs, especially since these models were introduced before the formalization of SSMs. Since LION **uses attention parallelization during training**, **it has matrix-valued hidden state** during inference, and it **uses multi-head attention** instead of state expansion, LION is categorized as "Transformer as Linear Recurrent Model".
>
> Moreover, LION extends these models into a bidirectional setting, where training uses attention parallelization (**without the parallel scan algorithm**), as proven in Theorem 3.3 and shown in Figure 1, and inference is done by a theoretically equivalent bidirectional RNN, as formalized in the same theorem. Additionally, our code is adapted from the original Transformer implementations of [ViT](https://github.com/lucidrains/vit-pytorch/tree/main) and [BERT](https://github.com/HazyResearch/m2/blob/main/bert/README.md), with modifications made only to the attention block to include masking during training (we have included a **torch implementation of the mask in Appendix section C6**), removing the positional embeddings, and adding the recurrent inference. The rest of the model architecture has not been altered. We will share the code upon acceptance.
>
> ---

---

> ### Author Response · Authors · 2024-11-17
> **Authors Rebuttal to Reviewer sped (2/4)**
>
> **Concern 3: Claims about extrapolation, more efficient than Transformers and solving LRA are not surprising**
>
> 1) **Efficiency over Transformers:** We would like to clarify that the novelty of our work is not the efficiency compared to Transformers. Rather, we demonstrate that LION enables the efficiency of Linear Transformers in a bidirectional setting, unlocking the efficient potential of the RNN format in this context. Since the efficiency of SSMs and RNNs during inference has already been established, we are proved that LION, being equivalent to an RNN during inference, is inherently efficient.
>
> 2) **Extrapolation Beyond Context Length Without Additional Positional Embedding:** Similar to the previous point, we do not claim LION is the only model able to extrapolate. Since LION-S does not use positional embeddings and can extrapolate beyond the context length, one might ask to see this ability in experiments, and we provide experimental evidence supporting that LION-S can indeed do this (Figure 3).
>
> 3) **Solving LRA:** Unlike the previous point, LION-S is indeed the first Linear Transformer capable of solving the LRA task, which is a significant result (Table 2). It is important to note that even Mamba, despite its strengths, is not able to solve the LRA task, as we have detailed in our response to the **Concern 5 of this response**.
>
> **Benefits of LION:** The LION framework and the model LION-S enable fast training and parallelization of transformers while maintaining the advantages of RNN/SSMs during inference, which are both faster and more efficient. It combines the best of both worlds for bidirectional sequence modeling. As evidence, the training times for different models for CIFAR100 on a single NVIDIA-A100 GPU with batch size 1024 are presented below:
>
>
> |Training Strategy (Model)| Time (s)/Epoch |
> |-|-|
> |Attention (VIT)  | 24.6|
> |Attention (**LION-S**)|35.8   |
> |Attention (**LION-LIT**)|26.6 |
> |Paralell Scan (Hydra)|43.4|
>
>
> This shows that training using attention is significantly faster than SSMs like Hydra, which is considered a state-of-the-art bidirectional SSM.
>
> ---
>
> **Concern 4: Comparison against vision SSMs**
>
> Although LION is not an SSM, for completeness, we compare it with the HYDRA \[8\] model, which is one of the best-performing state-of-the-art vision SSMs. We used HYDRA-T to match the parameter count. Results show that LION-S (v2) outperforms HYDRA on CIFAR-100 and performs very similarly on ImageNet. Note that the hyperparameters were taken from the original paper. LION_S (v2) is our extended approach that reorders patches to include more spatial information into calculations (for further details, please refer to Appendix C.8). As these results were obtained during the rebuttal period, we will continue tuning until our final submission. For the results LION-S is using the same parameters as ViT for training.
>
> | Model| CIFAR-100 | ImageNet|
> |-|-|-|
> | ViT-T |77.33 |**70.23** |
> | HYDRA-T|77.70|   69.60   |
> | LION-LIT | 73.61|62.69   |
> | LION-S|77.56   |   67.95   |
> | LION-S (v2) |**80.07**| 69.22|
>
> ---
>
> **Concern 5: Mamba and S5 results in experiments:**
>
> The poor performance of Mamba in the LRA task is well-recognized within the community and has been **acknowledged by Albert Gu**, the author of Mamba [6], Mamba-2 [4] after community finds that it is not as performant of other SSMs ([as stated in this GitHub issue](https://github.com/state-spaces/mamba/issues/282#issuecomment-2221135197)). Additionally, other papers [12] have reported similar performance for Mamba as we observed in our experiments. Regarding S5, we have reported their results from version 1 of their submission, which were also presented in [9]. We have now added the updated version of the results, both below and in Table 2 of our paper.
>
> |Model| ListOps | Text  | Retrieval | Image | Pathfinder |Path-X | Avg.  |
> |-|-|-|-|-|-|-|-|
> | [S5 version 1](https://arxiv.org/pdf/2208.04933v1)  also reported in [9] (**Table 1**)| 61.00   | 86.51 | 88.26| 86.14 | 87.57| 85.25  | 82.46 |
> |[S5 final version](https://arxiv.org/pdf/2208.04933v3)| 62.15 | 89.31 | 91.40 | 88.00 | 95.33 | 98.58 | 87.46 |
>
> ---
>
> **Concern 6: "Transformers can be represented as SSMs"**
>
> Thanks to reviewer we have emphasized in the footnote of our background section (on page 3) that softmax-based attention cannot be linearized. However, we would like to highlight that the fact that softmax-based attention does not have an exact finite-dimension equivalent in RNN form has been established in the "Transformers are RNNs" paper [1]. Variants of linear recurrent models, such as Performer [5], approximate softmax-based attention by applying Random Fourier Features.
>
> Additionally, we acknowledge that the sentence "Transformers can be presented as SSM" could be seen as imprecise, especially considering Mamba-2’s footnote. However, the title of Mamba-2, "Transformers are SSMs," can be considered even more imprecise than our statement following the same reasoning.

---

> ### Author Response · Authors · 2024-11-17
> **Authors Rebuttal to Reviewer sped (3/4)**
>
> **Concern 7: LRA Text Dataset**
>
> We thank the reviewer for their careful attention to the details of the paper. This was a typo, and it has been corrected in the new version of the paper. The Text dataset has a sequence length of 4096.
>
> ---
>
> **Concern 8: $exp(a_i)-1$ is not clear in equations 27 and 26 (LION-S section)**
>
> We would like to emphasize that the terms $exp(a_i)-1$ and $exp(a_i)$ are not arbitrary choices but are derived from the zero-order-hold (ZOH) discretization of a continuous-time system. This discretization method is well-established in signal processing, as noted in the HIPPO [14] paper and commonly referenced in standard signal processing textbooks such as [13].
> This ZOH discretization also appears in the Mamba paper, where the discretized parameters of ZOH filtering are presented as follows:
> $\hat{B} = (\Delta A)^{-1}(\exp{\Delta A} - I)\Delta B$
> $\hat{A} = \exp{\Delta A}$
> with $\Delta$ representing the time step size. To provide further clarity, we have included a detailed proof of ZOH discretization in Appendix Section B.6.
> Additionally, we find the reviewers' comment—"we have the sum of the two coefficients summing to, ensuring an attention mask"—unclear. Specifically, the term $(\exp{a_i} - 1)$ does not contribute to the creation of an attention mask, and the phrase "ensuring the mask" lacks precision and clarity. Only the parameter $\lambda_i$ contributes to the attention mask [3,4], and a good example of this is xLSTM [11] where the state parameter $\lambda_i$ and $\gamma_i$ (in equation 4 of our paper) are different and do not sum to one.
>
> ---
>
> **Concern 9: Adding LION-LIT on LRA**
>
> The reason for not initially including LION-LIT as a starting point is that it is the bidirectional version of LinearTransformer, which already appears in the table. However, to address the reviewer's concern, we have now added LION-LIT to Table 2 as a result.
>
> | Model   | ListOps | Text  | Retrieval | Image | Pathfinder | PathX | Avg.  |
> |---------|---------|-------|-----------|-------|------------|-------|-------|
> | LION-LIT| 16.78   | 65.21 | 54.00     | 43.29 | 72.78      | ✘     | 50.41 |
> | LION-S  | 62.25   | 88.10 | 90.35     | 86.14 | 91.30      | 97.99 | 86.07 |
>
> This addition further supports that the choice of LION-S is indeed significantly effective.
>
> ---
>
> **Concern 10: LSTM and GRU categorization**
>
> Thank you for the comment, which helped us clarify the definition and distinction of bidirectionality. We define bidirectional architectures as those that were initially introduced as bidirectional models. As a result, we have revised Table 1 of our paper to reflect this distinction, placing GRU and LSTM in the unidirectional category, as they were originally designed for unidirectional sequence modeling. We have separated these models into a different row of the table and added models like ELMO [10], which were designed as bidirectional RNNs, in a separate row.
>
> ---
>
> **Concern 11: SSMs dense information processing**
>
> Thank you for the comment. We would like to clarify that all linear attention-based models can be represented as RNNs [15], as the linear attention matrix is of rank $d$, where $d$ is the model hidden dimension, while softmax-based attention has rank $L$ due to the non-linear operation (exp) applied to the attention matrix. However, the reason for representing these models as RNNs is not just because of the rank of the matrix. We also want to highlight that studies like Performer [5] approximate softmax attention and can still be represented as an RNN. For clarity of our paper and claims and in response to the reviewer's suggestion, we have removed the sentence in our final manuscript.
>
> ---
>
> Since some of the weaknesses are not itemized below, we have mentioned which concerns correspond to the relative questions/weaknesses in our response.
>
> **W1: Answered in Concern 5**
>
> **W2: Answered in Concern 6**
>
> **W3: Answered in Concern 9**
>
> **W4: Answered in Concern 10**
>
> **W5: Answered in Concern 11**
>
> ---
>
> **Q1: Answered in Concern 1 and 2**
>
> **Q2: Answered in Concern 7**
>
> **Q3: Answered in Concern 8**
>
> **Q4: Answered in Concern 2**
>
> **Q5: Answered in Concern 5**
>
> **Q6: Answered in Concern 9**
>
> **Q7: Answered in Concern 10**
>
> ---

---

> ### Author Response · Authors · 2024-11-17
> **Authors Rebuttal to Reviewer sped (4/4)**
>
> **References**
>
> \[1\]: Transformers are RNNs: Fast Autoregressive Transformers with Linear Attention. Angelos Katharopoulos, Apoorv Vyas, Nikolaos Pappas, François Fleuret. arXiv:2006.16236, 2020.
>
> \[2\]: Retentive Network: A Successor to Transformer for Large Language Models. Yutao Sun, Li Dong, Shaohan Huang, Shuming Ma, Yuqing Xia, Jilong Xue, Jianyong Wang, Furu Wei. arXiv:2307.08621, 2023.
>
> \[3\]: Random Feature Attention. Hao Peng, Nikolaos Pappas, Dani Yogatama, Roy Schwartz, Noah A. Smith, Lingpeng Kong. arXiv:2103.02143, 2021.
>
> \[4\]: Transformers are SSMs: Generalized Models and Efficient Algorithms Through Structured State Space Duality. Tri Dao, Albert Gu. arXiv:2405.21060, 2024.
>
> \[5\]: Rethinking Attention with Performers. Krzysztof Choromanski, Valerii Likhosherstov, David Dohan, Xingyou Song, Andreea Gane, Tamas Sarlos, Peter Hawkins, Jared Davis, Afroz Mohiuddin, Lukasz Kaiser, David Belanger, Lucy Colwell, Adrian Weller. arXiv:2009.14794, 2022.
>
> \[6\]: Mamba: Linear-Time Sequence Modeling with Selective State Spaces. Albert Gu, Tri Dao. arXiv:2312.00752, 2024.
> \[7\]: Efficiently Modeling Long Sequences with Structured State Spaces. Albert Gu, Karan Goel, Christopher Ré. arXiv:2111.00396, 2022.
>
> \[8\]: Hydra: Bidirectional State Space Models Through Generalized Matrix Mixers. Sukjun Hwang, Aakash Lahoti, Tri Dao, Albert Gu. arXiv:2407.09941, 2024.
>
> \[9\]: Liquid Structural State-Space Models. Ramin Hasani, Mathias Lechner, Tsun-Hsuan Wang, Makram Chahine, Alexander Amini, Daniela Rus. arXiv:2209.12951, 2022.
>
> \[10\]: Deep Contextualized Word Representations. Matthew E. Peters, Mark Neumann, Mohit Iyyer, Matt Gardner, Christopher Clark, Kenton Lee, Luke Zettlemoyer. arXiv:1802.05365, 2018.
>
> \[11\]: xLSTM: Extended Long Short-Term Memory. Maximilian Beck, Korbinian Pöppel, Markus Spanring, Andreas Auer, Oleksandra Prudnikova, Michael Kopp, Günter Klambauer, Johannes Brandstetter, Sepp Hochreiter
>
> [12]: State Space Models as Foundation Models: A Control Theoretic Overview. Carmen Amo Alonso, Jerome Sieber, and Melanie N. Zeilinge
>
> [13]: Raymond A DeCarlo. Linear systems: A state variable approach with numerical implementation. Prentice-Hall, Inc., 1989.
>
> [14]: HiPPO: Recurrent Memory with Optimal Polynomial Projections: Albert Gu, Tri Dao, Stefano Ermon, Atri Rudra, and Christopher Re
>
> [15]: Parallelizing Linear Transformers with the Delta Rule over Sequence Length: Songlin Yang, Bailin Wang, Yu Zhang, Yikang Shen, Yoon Kim
>
> ---
>
> We hope that our clarifications have provided a clearer understanding of our contributions and that we have successfully addressed the reviewer's concerns. In case the reviewer has any remaining concerns, we are more than willing to address them

---

> ### Comment · Reviewer_sped · 2024-11-22
>
> I appreciate the author's efforts in providing a comprehensive rebuttal.
>
> > The poor performance of Mamba in the LRA task is well-recognized within the community and has been acknowledged by Albert Gu
>
> Mamba's author stated, "we believe it should perform well on data such as text and not as well on data such as images (e.g., the Image/Pathfinder tasks)." However, the recent xLSTM paper (accepted at NeurIPS) reports achieving 99.2% on Pathfinder with Mamba, a significant improvement over the 69.26% reported here. This raises some ambiguity about Mamba's performance expectations.
>
> Additionally, Mamba's author claims that performance on Retrieval is comparable to S4. However, this does not align with the results presented in the current work—Mamba achieves 72.14%, while S4 reaches 89.46%. Furthermore, the xLSTM paper reports a 90.2% performance for Mamba on Retrieval. Given these discrepancies, the Mamba results presented here seem quite unusual.
>
> > Comparison against vision SSMs
>
> LION-S (v2) introduces a new masking mechanism designed to encourage local information.
>  - Figure 7 suggests the use of two different masking mechanisms. How is the decision made on which one to apply?
>  - Are the Hydra experiments using the same masking mechanism as LION-S (v2)? If not, this could give LION-S (v2) an unfair advantage in comparison.
>  - The results in the Hydra paper indicate that Hydra significantly outperforms ViT on ImageNet. However, the numbers presented here show Hydra underperforming ViT. Could you provide some insight into why this discrepancy occurs?
>
> > Linear Tranformers, including LION, use attention matrix parallelization during training, whereas SSMs use the parallel scan algorithm.
> > torch implementation of the mask in Appendix section C6
>
> The PyTorch implementation of the mask in C6 uses cumsum (cumulative sum). While the paper argues that attention matrix parallelization is employed, there is still a parallel scan actually running under the hood. This scan computes the prefix sums (cumulative sums) in parallel, as an iterative approach would be too slow.
>
> ---
>
> n terms of similarities, LION-S utilizes an input-dependent gating mechanism, which aligns it with methods that also employ such gating, including xLSTM, Gateloop, Mamba's S6, GLA-Transformer, and LRU. However, a direct comparison with these methods is notably absent.
>
> While I agree with the authors' assertion that LION-S is not merely a sum of two SSMs, I apologize for framing it that way. However, I find it somewhat misleading that LION-S is described as a Transformer, with the discussion focusing exclusively on comparisons to Transformer-based models. Many of the input-dependent gating methods mentioned above are quite similar to the recurrence mechanisms described in Section 4. Furthermore, many of these input-dependent gating methods would outperform most of the Transformer models listed in Table 2 (e.g., LRU’s performance on Long Range Arena), and vision models based on these methods would likely be more efficient, as indicated by the results in Table 3.
>
> Additionally, many of these methods can generate a similar mask when applied bidirectionally (and normalized), which could lead to a result similar to a "Transformer" architecture. Appendix C.5 demonstrates this in the context of the proposed LION framework. It is also worth noting that the memory usage of these input-dependent gating methods likely leads to GPU usage comparable to what is shown in Figure 3.
>
> Since the primary focus of the paper is LION-S (not LION-LIT, which is based on the Linear Transformer), and since the core component of LION-S—the input-dependent recurrence—closely resembles these existing input-dependent gating methods, it would be natural for the paper to include comparisons with methods like xLSTM, Gateloop, Mamba's S6, GLA-Transformer, and LRU. I appreciate the inclusion of Hydra in the experiments, as I believe it provides a more fair comparison, given that it too is based on an input-dependent gating RNN.
>
> In my view, presenting direct experimental comparisons with these input-dependent gating methods would make the comparisons more comprehensive and fair. For example, including a memory comparison with Hydra (and/or Vision Mamba) in Figure 3 would help provide a clearer and more balanced evaluation.

---

> ### Author Response · Authors · 2024-11-22
> **Authors rebuttal to Reviewer sped (1/3)**
>
> Dear reviewer sped,
>
> Thanks for your response. We answer to your points as follows:
>
>
>
> - **P1: Regarding Mamba's performance on LRA**
>
> Thank you for your comment. We would like to highlight that our results for Mamba are taken from [1,6], we have updated Table 2 of our paper based on the results requested by the reviewer, specifically from Table 6 in the xLSTM paper [2]. However, we would like to point out two key differences:
>
> 1. **Mamba's performance is still lower than that of LION-S**.
> 2. Neither us nor other authors managed to make Mamba solve the PathX problem, while LION-S is able to solve this task. Please note that the **Mamba LRA results from xLSTM [2] are incomplete as PathX and Text are missing**.
>
> ---
>
> - **P2: Comparison against vision SSMs**
>
> 1. **Which mask to apply at LION-S(v2):** As we have mentioned in line 1498 of our Appendix C.8, we do not choose between these masks and we average two masks given in Figure 7 b and c.
> 2. **Does Hydra have the same mask:** We would like to emphasize that this mask, which is our contribution, is specific to our architecture. Since Hydra uses a matrix-form state $A$, we do not see a straightforward way of applying this technique to Hydra. As mentioned in the Mamba-2 paper [7], only the scale-valued $A$ can be represented as masked attention.
> 3. **Hydra results compared to VIT:** We appreciate the acknowledgement of our extra efforts towards including the results of this concurrent work (released less than 4 months before submission deadline, [please note last FAQ in ICLR 2025 reviewer guidelines](https://iclr.cc/Conferences/2025/ReviewerGuide)). However, we would like to highlight that in the Hydra paper [3], the ViT-Base family is considered. In our experiments, we consider the ViT Tiny and Small families. We reproduced Hydra based on their [repository](https://github.com/goombalab/hydra) for Tiny scale and reported the findings. Due to our limited resources, we are only able to run Hydra in the Tiny scale.
>
> ---
>
> - **P3: LION-S uses cumsum (parallel scan)**
>
> We would like to clarify that LION-S employs `cumsum`, that uses a parallel scan, for efficiently creating the mask. On the other hand, SSMs employ the parallel scan as the key ingredient to parallelize their recurrence formula. The main parallelization mechanism in LION-S is equivalent to the one in the transformer, i.e., through computing the attention matrix.
>
> It is worth noting that the `cumsum` operation is not unique to LION-S. Other prominent Transformer models, such as the [SLiM Performer official implementation](https://github.com/google-research/google-research/blob/master/performer/models/slim_performer/pytorch/numerator_and_denominator.py) [5], [Llava Hugging Face transformers implementation](https://github.com/huggingface/transformers/blob/d9e6f307e71b5108a7882ec00ffcc0d0eb316cb7/src/transformers/models/llava/modeling_llava.py) [4], and the [x-transformers library](https://github.com/lucidrains/x-transformers), also use `cumsum` in their training and they use attention parallelization.
>
> Moreover, we have demonstrated that our training time is significantly faster compared to parallel scan and is comparable to the full VIT training time, as shown in the table provided in our **initial response to your concern 3**.
>
> Therefore, we believe that the use of `cumsum` (in our case for mask creation) which is indeed common in many Transformer implementations under the hood and should not be considered neither as applying parallel scan training **nor as a reason for rejecting our approach** since it does not change the fact that our general training structure follows a transformer training.

---

> ### Author Response · Authors · 2024-11-22
> **Authors rebuttal to Reviewer sped (2/3)**
>
> - **P4: xLSTM, Gateloop, Mamba's S6, GLA-Transformer, and LRU are not included in the paper**
>
>
>     Thank you for your comments. Since our experiments are extensive across multiple domains, we believe the reviewer was referring to the absence of certain models in the LRA task results. We would like to address each baseline in detail:
>
>     - **Mamba (S6)**: We have already included Mamba in the LRA results in Table 2 (line 389 of our paper), and we have now included the new version mentioned by the reviewer.
>     - **xLSTM**: Thanks to the reviewer, we have added the xLSTM results to Table 2 of our paper for the LRA task.
>     - **GateLoop**: We appreciate your comment. However, we have not found results for GateLoop on the LRA benchmark. Additionally, the GateLoop paper does not include experimental results for the LRA task. We are willing to include it if the reviewer can provide a reference of the results.
>     - **LRU**: Thank you for pointing this out. We have added the LRU baseline to Table 2 as requested.
>
> For vision tasks, we would like to emphasize that models such as LRU and GateLoop have not been extended into the bi-directional setting. We include Hydra in our comparison, despite its release as pre-print on arxiv just four months ago. According to the [reviewer's guidelines](https://iclr.cc/Conferences/2025/ReviewerGuide) last FAQ point, comparing against Hydra may not be ideal, but we included it to extend our study in the rebuttal. Hydra is one of the few models that claims bi-directionality, which aligns with the scope of our contribution, making it relevant for comparison despite the timing of its release.
>
> Moreover, since Hydra is already outperforming other vision models like Vision Mamba, S4-VIT, and Hyena-VIT, comparing against Hydra provides a comprehensive evaluation, especially given the limited time during the rebuttal.
>
>
> ---
>
>
> - **P5: ""Many of these input-dependent gating methods would outperform most of the Transformer models listed in Table 2 ... Additionally, many of these methods can generate a similar mask when applied bidirectionally (and normalized), which could lead to a result similar to a "Transformer" architecture""**
>
> As also mentioned by the reviewer, "generating a similar mask" is **the key contribution of our paper**. Specifically, we introduce a novel method for generating the mask and extending existing models into their bi-directional setting, as detailed in Appendix C.5. The models we compare against do not have such a mask in their architecture for the bi-directional case, nor do they have the capability to train in a bi-directional setting, which remains **the core contribution of our work**.
>
> Asking to extend these models into a bi-directional setting and then comparing them against LION-S is not entirely fair, as we would essentially be comparing our own contribution to itself. In any case, both approaches are based on the LION framework, and the central focus of our work is extending existing models into the bi-directional setting, which represents a significant and novel contribution to the field.
>
> LION-S is a representative example of using this framework with a selective mask, demonstrating the potential of gating mechanisms in LION framework.
>
>
> ---
>
> - **P6: "Memory comparison with Hydra"**
>
> We have added the memory consumption of Hydra in Figure 3. For this comparison, we used the memory efficient implementation in the [Official Hydra repo](https://github.com/goombalab/hydra/issues/5). In below table, we present the inference GPU usages of models in terms of GB. It can be observed that Hydra requires a memory nearly 2.5$\times$ that of LION-S/LIT for high resolutions.
>
>
> | Model   | 224     | 512     | 738     | 1024    | 1248    | 2496    |
> |---------|---------|---------|---------|---------|---------|---------|
> | ViT     | 0.21 | 3.48   | 13.79 | 49.90 | OOM     | OOM     |
> | Hydra   | 0.70   | 2.84  | 5.86 | 11.34 | 16.85 | 59.80     |
> | LION-S/LIT  | 0.21  | 1.03  | 2.13  | 4.13   | 6.14  | 24.51 |
>
> ---
>
> ## Final remarks
>
> We appreciate the reviewer's efforts and feedback, which helped us improve our work by incorporating additional baselines such as Mamba results from the Appendix of xLSTM and LRU. These additions have certainly enhanced the quality of our paper. However, we believe it is unfair to assign a score of 3 and recommend rejection solely based on these points. If the concerns raised have been addressed, we kindly ask the reviewer to consider revising the rating accordingly.
>
> ---

---

> > ### Author Response · Authors · 2024-11-22
> > **Authors rebuttal to Reviewer sped (3/3)**
> >
> > **References**
> >
> > [1] Amo Alonso et al., State Space Models as Foundation Models: A Control Theoretic Overview, arXiv preprint arXiv:2403.16899, 2024.
> >
> > [2] Beck et al., xLSTM: Extended Long Short-Term Memory, NeurIPS 2024
> >
> > [3] Hwang et al., Hydra: Bidirectional State Space Models Through Generalized Matrix Mixers, arXiv preprint arXiv:2407.09941, July 16th 2024
> >
> > [4] Liu, H., Li, C., Wu, Q., & Lee, Y. J., Visual Instruction Tuning, arXiv preprint arXiv:2304.08485, April 18th 2023.
> >
> > [5] Likhosherstov, V., Choromanski, K., Davis, J., Song, X., & Weller, A., Sub-Linear Memory: How to Make Performers SLiM, arXiv preprint arXiv:2012.11346, December 21st 2020.
> >
> > [6] Qin, Z., Yang, S., Sun, W., Shen, X., Li, D., Sun, W., & Zhong, Y., HGRN2: Gated Linear RNNs with State Expansion, arXiv preprint arXiv:2404.07904, April 17th 2024.
> >
> > [7] Dao, T., & Gu, A., Transformers are SSMs: Generalized Models and Efficient Algorithms Through Structured State Space Duality, arXiv preprint arXiv:2405.21060, May 28th 2024.

---

> > > ### Comment · Reviewer_sped · 2024-11-23
> > >
> > > I would like to thank the authors for their in-depth responses.
> > >
> > > ## Regarding Hydra
> > > I agree with the authors that this work is concurrent. However, I would like to clarify that **the method referenced in the review was Vision Mamba**, a well-established model (~9 months before the ICLR deadline) and not a contemporary approach. The authors report that Hydra "is already outperforming other vision models like Vision Mamba, S4-VIT, and Hyena-VIT," and that comparing against Hydra provides a comprehensive evaluation. While this is a valid claim, the Hydra paper asserts that Hydra significantly outperforms ViT on ImageNet, yet the results presented here do not support this comparison, raising ambiguity about Hydra's relative performance against other baselines.
> > >
> > > In contrast, Vision Mamba is well-established, bidirectional, and significantly outperforms ViT. The scale of the Vision Mamba models (e.g., Vim-Ti with 7M parameters) is also more comparable to the models in this paper (5.5M parameters, which is relatively small).
> > >
> > > ## Summary regarding Score of 3
> > >
> > > The framing of the paper's experimental section leads one to focus heavily on comparisons between Transformers and LION-S, particularly in terms of performance and memory efficiency. However, the empirical results themselves are expected given the current literature.
> > >
> > > LION-S is based on a linear recurrent model with input-dependent gating, which is made bidirectional to enable training like a Transformer. It is already well-established in the literature (e.g., LRU, S5, Vision Mamba) that linear recurrent models, including bidirectional variants, can substantially outperform Transformers in both performance and memory efficiency across the tasks discussed in the paper. As such, the results reaffirm what has been shown in prior work.
> > >
> > > In my view, the more interesting aspect of this paper lies in its proposed framework (LION) and its potential advantages and implications, though there is very limited analysis provided in this regard.
> > >
> > >
> > > > Asking to extend these models into a bi-directional setting and then comparing them against LION-S is not entirely fair, as we would essentially be comparing our own contribution to itself. In any case, both approaches are based on the LION framework, and the central focus of our work is extending existing models into the bi-directional setting, which represents a significant and novel contribution to the field.
> > >
> > > I respectfully disagree. The paper presents two distinct methodological contributions:
> > >  - A framework (LION) that enables the modification of existing models into bidirectional, Transformer-like architectures.
> > >  - A new linear recurrent model (LION-S) that leverages this framework.
> > >
> > > As the paper introduces the LION framework (with mathematical formulations in Appendix C.5) to modify existing models into a bidirectional Transformer-like structure, natural questions arise: What are the advantages of modifying these methods (xLSTM, Retentive, Gated RFA) with the LION framework (LION-LSTM, LION-RET, or LION-GRFA in Appendix C.5)? How does this modification impact performance? If the performance gain is marginal, then what other advantages does LION offer (e.g., attention mask interpretability)?
> > >
> > > Additionally, since the paper mentions that leveraging LION these modified models are trained like a Transformer, it raises the question of whether training models like xLSTM, Retentive Networks, or Gated RFA using the LION framework (resulting in models like LION-LSTM, LION-RET, and LION-GRFA) is more efficient. If so, what are the runtime and memory characteristics during training?
> > >
> > > Instead of focusing solely on the new LION-S model, the paper would be stronger if it focused more on the analyses and comparisons of those existing methods that could be modified using the LION framework. Specifically, it is not clear to me why a new model, LION-S was introduced rather than focusing on adapting other models like LION-LSTM, LION-RET, or LION-GRFA. However, since a new linear recurrent model is proposed that LION is applied to, it makes sense to compare it with existing linear recurrent models which LION can be applied to (e.g., LION-LSTM, LION-RET, and LION-GRFA).
> > > Although the paper evaluates LION-LIT, Linear Transformer on which the model is based is outdated and performs very poorly on LRA (as shown in Table 1), limiting the insights gained from these comparisons.
> > >
> > > Due to the points raised above, I maintain my score of 3.

---

> ### Author Response · Authors · 2024-11-24
> **Authors rebuttal to Reviewer sped (1/4)**
>
> Dear reviewer sped, thanks for engaging actively in the discussion which we highly appreciate and helped us to add more analysis and details to our work. Since many points have been raised during this rebuttal period, let us start by summarizing the concerns you raised and have been addressed, to then answer to your remaining concerns.
>
> | **Reviewer's concern** | **Origin**              | **Action** |
> |-------------|-------------------------|------------|
> | LION-S is two SSMs summed together | Original Review | [Showed that it's not; in authors first response (1/4)](https://openreview.net/forum?id=VxJjw52dZu&noteId=s4SEACr0Bc)            |
> | Difference between LION and SSMs | Original Review | [Addressed in authors first response (1/4)](https://openreview.net/forum?id=VxJjw52dZu&noteId=s4SEACr0Bc)           |
> | Length extrapolation, LION-S being more efficient than Transformers and solving LRA are not surprising | Original Review | [Explained in authors first response (2/4) + training times evidence](https://openreview.net/forum?id=VxJjw52dZu&noteId=DiyPbUmZ2N)       |
> | Missing performance of Mamba and S5 results in LRA | Original Review | Added to Table 2         |
> | "Transformers can be represented as SSMs" is misleading | Original Review | [Clarified in authors first response (2/4)](https://openreview.net/forum?id=VxJjw52dZu&noteId=DiyPbUmZ2N) and background section footnote (page 4)      |
> | $\text{exp}(a_i) - 1$ is not clear in equations 27 and 26 (LION-S section) | Original Review | [Clarified in authors first response (3/4)](https://openreview.net/forum?id=VxJjw52dZu&noteId=imSXYnsf3X) and Appendix B.6|
> | The performance of LION-LIT on LRA is missing | Original Review | Added to Table 2 |
> | Are LSTMs and GRU bidirectional? | Original Review | Modified Table 1 |
> | SSMs dense information processing | Original Review | [Addressed in authors first response (3/4)](https://openreview.net/forum?id=VxJjw52dZu&noteId=imSXYnsf3X) and removed sentence from manuscript|
> | LION-S also uses the parallel scan in 'cumsum' | Reviewer First Response | [Addressed in authors second response (2/3)](https://openreview.net/forum?id=VxJjw52dZu&noteId=WacW6Sw0yp)           |
> | xLSTM, Gateloop, Mamba's S6, GLA-Transformer, and LRU are not included in the paper | Reviewer First Response | Added to Table-2 |
> | Missing memory comparison with Hydra | Reviewer First Response | [Added to authors second response (2/3)](https://openreview.net/forum?id=VxJjw52dZu&noteId=EMbMwgIf44) + Figure 3|
> | How is the LION-s (v2) mask applied| Reviewer First Response | [Added to authors second response (1/3)](https://openreview.net/forum?id=VxJjw52dZu&noteId=WacW6Sw0yp) & Appendix C.8|
> |Can Hydra use the same mask as LION-S(v2)| Reviewer First Response | [Added to authors second response (1/3)](https://openreview.net/forum?id=VxJjw52dZu&noteId=WacW6Sw0yp)|
>
> We would also like to emphasize that some of the concerns raised have already been addressed in the appendix of both the original and revised versions of the paper, and we have simply pointed them out for clarity. By addressing these concerns, we believe the quality of the paper has increased. We kindly ask you to acknowledge that the previous concerns have been addressed.
>
> The remaining concerns raised by the reviewer regarding the Vision Mamba and Hydra baselines, variants of LION (such as LION-RETNET), training time comparisons, and the focus of our paper on the LION framework are addressed below:
>
> ---
>
> **(1) The method referenced in the review was Vision Mamba**
>
> Thanks for the clarification. Since the reviewer believes that Vision Mamba is an important baseline to include in our study, we have added the results for Vision Mamba Tiny and Small in our new experimental results in Table 4 of our paper, based on their work [1]. Note that there are salient differences in their training setup, including data augmentation and other novelties, which are tailored towards improving the accuracy (**which is not our focus nor our contribution here**). However, we will still continue working on reproducing the Vision Mamba results and aim to include the reproduction by the end of the rebuttal period.
>
> Moreover, the key advantage of LION lies in its efficient training with reduced time and memory resources. Our framework enables linear recurrent models like Linear Transformer, RETNET, and selective variants to also benefit from fast/parallelized training of Transformers. Our experiments illustrate that these models perform as well as vision models and outperform VIT in several tasks.
>
>
> ---

---

> ### Author Response · Authors · 2024-11-24
> **Authors rebuttal to Reviewer sped (2/4)**
>
> **(2) Comparing against Hydra provides a comprehensive evaluation. Hydra paper asserts that Hydra significantly outperforms ViT on ImageNet, yet the results presented here do not support this comparison**
>
> As we mentioned in our previous response and as reflected in our paper (Table 4), the **scale of Hydra that we reproduced is "Tiny," while the only scale represented in the original Hydra manuscript is "Base."** Therefore, our results are not in contradiction with the Hydra paper, as they never claimed that "Tiny" size for Hydra outperforms VIT, and it does not exist in their publication. Moreover, after a careful reading, we failed to find a direct comparison of Hydra and Vision Mamba [1] in the Hydra paper.
>
> Furthermore, since Hydra was suggested by reviewer tj5X, and considering the short rebuttal period and our limited resources, we decided to include Hydra. Hydra is more aligned with our study since it is applied to both vision and bi-directional language modeling tasks. For comparison, we have also added the Vision Mamba results from their paper in Table 4 of our manuscript.
>
> ---
>
> **(3) The paper presents two distinct methodological contributions:**
>
> **- A framework (LION) that enables the modification of existing models into bidirectional, Transformer-like architectures.**
> **- A new linear recurrent model (LION-S) that leverages this framework.**
>
> **We have revised our main claims in the paper to better align with the study. Now, we introduce LION-S as a representative of our framework with a selective mask**, ensuring the focus remains on our framework rather than solely on LION-S. Additionally, we clarified that LION-S serves as a running example illustrating how our framework works for both fixed and selective masks. As represented in our introduction now our main claims are clarified as:
>
> - We propose a theoretical framework LION (Theorem 1), which expresses bidirectional Transformers as bidirectional RNNs, enabling efficient inference for long sequences while benefiting from well-established Transformer training (cf., Table 1).
> - Our theoretical framework offers the foundations to transform a wide class of autoregressive recurrent models (cf., Section 2) into their bidirectional counterparts.
> - We propose three main running examples of our framework, inspired by prior work, namely:
>   1. **LION-LIT**: Scaled attention without masking, a bidirectional extension of Linear Transformer.
>   2. **LION-RETNET**: Fixed masked scaled attention with scalar and learnable state parameter $\gamma$, an extension of RETNET into the bidirectional setting.
>   3. **LION-S**: Selective masked scaled attention with input-dependent mask $\lambda_i$, inspired by the selectivity of Mamba-2.
> - Through extensive experiments in the Long Range Arena, Vision Tasks, and Masked Language Modeling, we have demonstrated the capabilities of the LION framework and the models built upon it, as outlined above.
>
>
> ---

---

> > ### Author Response · Authors · 2024-11-24
> > **Authors rebuttal to Reviewer sped (3/4)**
> >
> > **(4) If the performance gain is marginal, then what other advantages does LION offer**
> >
> > We would like to highlight the advantages  of the LION framework as presented in our paper:
> >
> > - **Faster Training Speed:** LION enables training using attention which trains significantly faster than parallel scan, as demonstrated by LION-LIT, which is **1.7× faster** than parallel scan used in SSM-based models (Also mentioned in table in our response to your concern 5 of this review).
> > - **Bi-directional Extension of Linear Transformers:** Our framework extends existing linear transformers and their masked versions into the bi-directional setting, preserving all the advantages initially claimed for the uni-directional case. Please note the impact of the linear transformers had in the literature and our work in essence completes the bidirectional picture in a complementary fashion.
> > - **Extensive Evaluation:** We have evaluated three main running examples of LION-S, LION-RETNET, and LION-LIT, and shown that they are highly performant for MLM and vision tasks, even outperforming VIT.
> > - **Solid Performance on Large-Scale Vision and Language tasks:** We have run extensive experiments on large-scale tasks to demonstrate the robustness and effectiveness of our framework, validating its real-world applicability.
> > - **LRA Performance of LION-S:** LION-S outperforms other selective variants, such as Mamba, and non-selective models like Linear Transformer on the Long Range Arena benchmark. Unlike other linear recurrent models, LION-S is also capable of solving the Path-X problem.
> > - **LION-S (v2):** Thanks to the LION, we introduced new masks that are better suited for image locality, and LION-S has outperformed the classical ViT in the same training setup.
> > - **No intense fine-tuning for LRA, Vision, and Language Modeling tasks:**
> >   - In the Long Range Arena (LRA) tasks, LION-s results were achieved using the same model dimensions as the original Linear Transformer baseline, without extensive fine-tuning for each individual task, unlike baselines such as S5.
> >   - In the vision experiments, we only replaced the attention block in ViT with our LION block and did not alter any other parameters apart from adjusting the learning rate. Additionally, we did not use data augmentation, unlike models like Vision Mamba and Hydra, which extensively fine-tuned data augmentation to achieve higher scores.
> >   - For the Masked Language Modeling (MLM) tasks, we did not perform heavy fine-tuning for each task, unlike Hydra, which modified the training paradigm of the original BERT model to achieve higher GLUE score.
> >
> > All of the above experiments sufficiently demonstrate the usability of LION and its variants across different settings, confirming that it works effectively.
> >
> > ---
> >
> > **(5) What are the runtime and memory characteristics during training?**
> >
> > Thanks for the comment, we have already included the run times of these models for the vision task in our response to your concern 3, but we would like to highlight it here as well. Additionally, memory requirements during training and FLOPs are included in the appendix.
> > |Training Strategy (Model) |Time(s) /Epoch|
> > |-|-|
> > |Attention (VIT)|24.6|
> > |Attention (**LION-S**)| 35.8|
> > |Attention (**LION-LIT**)| 26.6|
> > |Attention (**LION-RETNET**)|28.4|
> > |Parallel Scan (Hydra)|43.4|
> >
> > As seen above, **LION-S trains 20% and LION-RETNET trains 52% faster** compared to parallel scan used in Hydra and Vision Mamba, showing the considerable advantage of our training paradigm over parallel scan. The training advantages are expected to increase as the model dimensions grow.
> >
> > ---
> >
> > **(6) LION-LSTM, LION-RET, or LION-GRFA are missing**
> >
> > Thank you for the suggestion. We believe that adding LION-LSTM and LION-GRFA would enhance the thoroughness of our experimental analysis. However, we would like to point out that we have already included LION-RETNET as an example of a fixed mask strategy for training in Appendix D.6. In response to the reviewer’s request, we have now added the new baseline (LION-RETNET) to emphasize the flexibility of our framework in both MLM and vision tasks. The results have been updated in Table 3 and Table 4 of our paper, and we have also adjusted our claims in the introduction accordingly.
> >
> > LION-LSTM and LION-GRFA can be considered more specific subcategories of LION-S, where the $K$ values are multiplied by an additional gating mechanism. These models still follow the core principles of LION-S but introduce selective gating for further refinement of the attention mechanism.
> >
> > Due to the inclusion of LION-LSTM and LION-GRFA as specific subcategories of LION-S, we have revised the conclusion of our paper to better align with the focus of our numerical evidence on the LION framework. This revision highlights the core contributions of LION, including the extension of linear transformers into bi-directional models and the significant improvements in training efficiency and performance across a variety of tasks.

---

> > > ### Author Response · Authors · 2024-11-24
> > > **Authors rebuttal to Reviewer sped (4/4)**
> > >
> > > ---
> > >
> > > **(7) The results reaffirm what has been shown in prior work**
> > >
> > >
> > > We sincerely disagree with the reviewer’s point on the following grounds:
> > >
> > > 1) As highlighted in our rebuttal and general response, the results for LION clearly demonstrate the advantages of its inference mechanism, which behaves **similarly to SSMs/RNNs**. Our key claim is that LION is faster during training, which we further emphasized in our responses during rebuttal. Empirical evidence supporting this claim is provided in the table of concern 5 of this response.
> > >
> > > 2) These advantages have been well-established in the context of uni-directional linear recurrent models. However, for linear transformers and their selective variants, such as LION-S these benefits had not previously been shown, especially in the bidirectional setting. LION-LIT, now LION-RETNET in the main text and LION-S, as representative examples, demonstrate this superiority during inference for bidirectional models.
> > >
> > > ---
> > >
> > > ### Our concerns with this review process
> > >
> > > Thank you for your review. We appreciate the time and effort you have taken to engage with our work. However, we would like to respectfully point out a few concerns we had regarding the evaluation:
> > >
> > > It seems that some of the points we addressed in our previous responses were not acknowledged. Despite our extensive additions to the manuscript such as new baselines, theoretical proofs, and updated measurements, these efforts were not sufficiently recognized in the evaluation. There appears to be a shift to new concerns with each new response, without taking into account the ongoing clarifications and adjustments we've made in the paper. The framing of our contribution seems to have been based on a misunderstanding, which has led to some subjective conclusions that might not fully reflect the essence of our work.
> > >
> > > We believe that a more thorough discussion of these points would lead to a fairer and more balanced evaluation of our contribution. We would be happy to provide any additional clarifications if needed.
> > >
> > > ---
> > >
> > > **References**
> > >
> > > [1] Vision Mamba: Efficient Visual Representation Learning with Bidirectional State Space Model, Lianghui Zhu, Bencheng Liao, Qian Zhang, Xinlong Wang, Wenyu Liu, Xinggang Wang

---

> > > > ### Comment · Reviewer_sped · 2024-11-24
> > > >
> > > > I would like to once again thank the authors for their ongoing discussion and apologize for not explicitly acknowledging their responses to my earlier concerns. I truly appreciate the time and effort the authors have dedicated to addressing these points.
> > > >
> > > > It took me some time to identify my core/fundamental concern with the paper (as outlined in the "Summary regarding score of 3"), and I recognize this was partly due to my inexperience. I apologize for any confusion caused by this.
> > > >
> > > > To conclude, I would like to clarify my main point regarding the "Summary regarding score of 3":
> > > >
> > > > Since the paper highlights the benefits of LION as a framework, I would **expect to see comparative analyses between the original methods** (e.g., LIT, RETNET, S, xLSTM) **and their LION counterparts** (e.g., LION-LIT, LION-RETNET, LION-S, LION-xLSTM). These comparisons should focus on **key factors such as training memory, runtime, inference memory/runtime, and performance**. For example, does the attention matrix in LION require quadratic in memory during training, introducing the same scalability issues as Transformers (and making the original methods less scalable)?
> > > >
> > > > I do not see direct apples-to-apples comparison in the paper. Instead, the emphasis is placed on comparing LION-S with Transformer variants and variants of methods using parallel scan. This raises ambiguity whether the benefits presented are truly attributable to the LION framework itself or the underlying method.
> > > >
> > > > If the authors can provide direct comparisons of the original methods alongside their LION-augmented versions, clearly demonstrating the benefits of their framework (improvements in training memory, runtime, inference memory/time, and/or performance), I would be convinced and happy to revise my score.

---

> > > > > ### Author Response · Authors · 2024-11-24
> > > > > **Authors rebuttal to Reviewer sped (1/2)**
> > > > >
> > > > > Dear Reviewer sped,
> > > > >
> > > > > We sincerely thank you for acknowledging that your previous concerns have been addressed. We truly appreciate your willingness and efforts to contribute to our paper.
> > > > >
> > > > > We also thank you for the clarification on your core/fundamental concern. Below, we will continue to address the key points that might have led to the misunderstanding of our contributions.
> > > > >
> > > > > ---
> > > > >
> > > > > We will quote your core/fundamental concern here:
> > > > >
> > > > > "I would expect to see comparative analyses between the original methods (e.g., LIT, RETNET, S, xLSTM) and their LION counterparts (e.g., LION-LIT, LION-RETNET, LION-S, LION-xLSTM). These comparisons should focus on key factors such as training memory, runtime, inference memory/runtime, and performance."
> > > > >
> > > > > We begin by stressing that there is no **"original method"** called LIT or RETNET  designed for **bi-directional sequence modeling**. Indeed, Linear Transformers, such as LinTrans [2], RETNET [3], and GRFA [4], are originally designed to be trained as Transformers and inferred as RNNs **only in the causal sequence modeling**.
> > > > >
> > > > > The LION framework extends the existing paradigm by enabling **training like a Transformer** (i.e., parallelized training with attention) and **inference as an RNN** in the **bi-directional setting**.
> > > > >
> > > > > As shown in our paper and during rebuttal, simply applying two recurrent models for the forward and backward recurrences **does not result in full attention**, and therefore does not allow parallelization during training (see Figure 2 and Observation 3.1).
> > > > >
> > > > > Using recurrent models for forward and backward recurrences would be efficient during inference. However, training them naively results in sequential processing (similar to traditional RNNs e.g., GRU) which is inefficient and causes slow training.
> > > > >
> > > > > The main motivation for the invention of Transformers was to **parallelize training** (as demonstrated in Table 1 of the original Transformer paper [1] and our own paper). Sequential training, particularly for long sequences, is inefficient and slow. Linear Transformers (e.g., Linear Transformer [2], RETNET [3], etc.) were originally designed for causal sequence modeling to achieve **the best of both worlds** (as quoted in "Transformers are RNNs" [2]) — parallel training and RNN inference. However, these models cannot be trained as Transformers and inferred as RNNs **in the bi-directional context**. LION is the framework that extends these models to enable training like Transformers and inference like RNNs in a bi-directional setting. Therefore, there is no "original LIT" or "RETNET" for bi-directional tasks, which is the focus of our work.
> > > > >
> > > > >
> > > > > So, we cannot compare the original RETNET or LIT for bidirectional tasks simply because their original training strategy does not work (shown in Observation 3.1 of our paper) due to 2 main issues 1) double counting of diagonal and 2) incorrect scaling. We address these problems in LION by **introducing modifications specifically designed for the bi-directional setting** to enable parallelized training with attention and inference like RNNs for Linear Transformers.
> > > > >
> > > > >
> > > > > An alternative method for training introduced by SSMs, and utilized by models like xLSTM, is **the parallel scan**. This approach allows for parallelized training, overcoming the limitations of sequential processing. The primary reason we *compare against parallel scan is that, for models like xLSTM and their variants, there is no other alternative for parallelizing training*—**except for LION**. Also, Linear Transformers, such as Performer, LIT, RETNET, and others, do not employ **parallel scan** for training in causal sequence modeling. Instead, they rely on **attention parallelization** for efficient training. However, **without** the **LION framework**, this parallelization **cannot be applied to the bi-directional setting** (shown in observation 1.3 of our paper).
> > > > >
> > > > > Hence, LION elevates these models, originally designed for causal settings, into the bi-directional domain, enabling them to train efficiently using attention while maintaining the benefits of RNN-like inference.
> > > > >
> > > > > In the case of comparing the original versions of LIT, RETNET, GRFA, and others, it is important to note that these models were originally designed exclusively for **causal sequence modeling**. Therefore, comparing their **bi-directional setting**, which is represented by LION, with the **original causal setting** is not valid because LION will inherently show superiority. This is because the original models are unidirectional, while LION enables bidirectional training and inference, which is a key advantage.
> > > > >
> > > > > Training these models using **sequential training** would result in significantly longer training times because they are not parallelizable in this form. Sequential training goes against the very motivation behind the creation of Linear Transformers, which was to enable **parallelized training** (as demonstrated by the Transformer model).

---

> > > > > > ### Author Response · Authors · 2024-11-24
> > > > > > **Authors rebuttal to Reviewer sped (2/2)**
> > > > > >
> > > > > > In summary, both options—comparing LION’s bi-directional setting with the original models in their causal form or training these models sequentially—fundamentally contradict the original goal of Linear Transformers, which is to facilitate parallel training with attention. **LION is the framework that enables bidirectionally for these models**, allowing them to train efficiently in parallel over the sequence with attention, while still benefiting from RNN-like inference in the bi-directional case.
> > > > > >
> > > > > > We hope that we have successfully clarified the contribution of our work and addressed the reviewers' concerns. However, we are still uncertain about what the reviewer refers to as the "original" case of Linear Transformers, as such a concept does not exist in the context of bi-directional sequence modeling. Linear Transformers, in their original form, were designed for causal sequence modeling and do not have a bi-directional counterpart unless extended by frameworks like LION.
> > > > > >
> > > > > > ---
> > > > > >
> > > > > > **References:**
> > > > > >
> > > > > > [1] Attention Is All You Need. Ashish Vaswani, Noam Shazeer, Niki Parmar, Jakob Uszkoreit, Llion Jones, Aidan N. Gomez, Lukasz Kaiser, Illia Polosukhin
> > > > > >
> > > > > > [2] Transformers Are RNNs: Fast Autoregressive Transformers with Linear Attention. Angelos Katharopoulos, Apoorv Vyas, Nikolaos Pappas, François Fleuret. 2020.
> > > > > >
> > > > > > [3] Retentive Network: A Successor to Transformer for Large Language Models. Yutao Sun, Li Dong, Shaohan Huang, Shuming Ma, Yuqing Xia, Jilong Xue, Jianyong Wang, Furu Wei. 2023.
> > > > > >
> > > > > > [4] Random Feature Attention. Hao Peng, Nikolaos Pappas, Dani Yogatama, Roy Schwartz, Noah A. Smith, Lingpeng Kong. 2021.

---

> > > > > > > ### Author Response · Authors · 2024-11-26
> > > > > > >
> > > > > > > **Dear Reviewer sped,**
> > > > > > >
> > > > > > > We would like to share our latest results, regarding the forward-only and backward-only recurrences of LION-S and LION-S (v2). Specifically, we trained LION-S and LION-S (v2) in a single direction for tasks that are inherently bi-directional. The results are presented below:
> > > > > > >
> > > > > > > ### Results for LION-S (Forward and Backward Only) Without Patch Ordering
> > > > > > >
> > > > > > > | Method                | Score  |
> > > > > > > |-----------------------|--------|
> > > > > > > | LION-S (Forward only) | 71.08  |
> > > > > > > | LION-S (Backward only)| 69.61  |
> > > > > > > | **LION-S (Bi-directional)** | **77.56** |
> > > > > > >
> > > > > > > ### Results for LION-S (v2) (Forward and Backward Only) With Modified Patch Ordering
> > > > > > >
> > > > > > > | Method                        | Score  |
> > > > > > > |-------------------------------|--------|
> > > > > > > | LION-S (v2) (Forward only)    | 70.24  |
> > > > > > > | LION-S (v2) (Backward only)   | 70.42  |
> > > > > > > | **LION-S (v2) (Bi-directional)** | **80.07** |
> > > > > > >
> > > > > > > These results demonstrate that both LION-S and LION-S (v2) achieve approximately **10% higher accuracy** in their bi-directional configurations compared to their forward-only and backward-only counterparts. This improvement aligns with expectations, as these models are inherently designed for bi-directional tasks, whereas the unidirectional configurations are more suitable for causal or sequential tasks.
> > > > > > >
> > > > > > > Furthermore, **as per your request**, we have trained adaptations of **LION-GRFA** and **LION-RETNET** for bi-directional tasks on the CIFAR-100 dataset. The results are as follows:
> > > > > > >
> > > > > > > ### Results for LION-GRFA and LION-RETNET
> > > > > > >
> > > > > > > We have trained the **uni-directional models of GRFA and RETNET** on CIFAR-100 and compared their performance to the bidirectional LION counterparts (i.e., the extensions of these models in the bidirectional setting). The results are presented below:
> > > > > > >
> > > > > > > | Method         | Score  |
> > > > > > > |----------------|--------|
> > > > > > > | GRFA           |71.56  |
> > > > > > > | LION-GRFA      |**73.24**|
> > > > > > >
> > > > > > > | Method         | Score  |
> > > > > > > |----------------|--------|
> > > > > > > | RETNET         | 72.24  |
> > > > > > > | LION-RETNET    | **75.66**  |
> > > > > > >
> > > > > > > These results demonstrate that bi-directionality significantly improves the performance of both RETNET and GRFA in the task of image classification.
> > > > > > >
> > > > > > > We hope this provides further clarification and empirical evidence supporting the effectiveness of our framework.
> > > > > > >
> > > > > > > ---

---

> > > > > > > > ### Author Response · Authors · 2024-12-02
> > > > > > > > **Final Day of Discussion Period for ICLR 2025**
> > > > > > > >
> > > > > > > > Dear Reviewer sped,
> > > > > > > >
> > > > > > > > As there is only **one day remaining in the extended ICLR discussion period**, we would like to express our appreciation for your engagement and feedback during this process. Your comments have led to the inclusion of new experimental and theoretical results in our paper.
> > > > > > > >
> > > > > > > > As you mentioned, clarifying the distinction between Linear Transformers like RETNET and their LION counterparts would lead to an increase in your score.
> > > > > > > >
> > > > > > > > **We believe we have successfully addressed this by explaining that LION is the first framework that extends Linear Transformers into the bi-directional setting**. Additionally, we have provided supporting experimental results that demonstrate the comparison between original Linear Transformers like **GRFA and RETNET and their LION counterparts**.
> > > > > > > >
> > > > > > > > We would be grateful if, upon review, you could consider increasing your score, as you previously mentioned, since we believe we have addressed your concerns.
> > > > > > > >
> > > > > > > > Once again, thank you for your constructive feedback and engagement during the rebuttal period.
> > > > > > > >
> > > > > > > > Best regards,
> > > > > > > >
> > > > > > > > Authors

---

### Official Review · Reviewer_z393 · 2024-11-04

**Soundness:** 2
**Presentation:** 3
**Contribution:** 2
**Rating:** 5
**Confidence:** 4

**Summary:**

They show that applying two linear attention mechanisms in two directions would not be equivalent to an bidirectional Transformer. Hence, they propose to scale masked linear attention in a different way to make it more similar to birectional full attention. Furthermore, they propose LION-S that combines LION with the selective mechanism of SSMs.

**Strengths:**

1. They try to find an equivalent recurrence for the scaled attention, which is more accurate than linear attention.
2. The visualization of upper/lower triangular matrix is helpful.

**Weaknesses:**

1. Some equations/notations are not clear enough. For example, in Eq 6 and 7, it looks like equations A1/A2 are without masks while B1 and B2 are with masks, but this is not explained, and lambda is not defined.
2. LION-Lit performs worse than BERT or ViT, and the gap is not negligible. If LION is equivalent to full attention, the difference should not be that large.
3. The advantage of adding selectivity is not well explained. Table 2 only shows LION-S results, but it does not compare it with LION.
4. The definitions of the architectural elements of figure 1 should be added to its caption.

**Questions:**

1. Is this method limited to bidirectional models? Most generative models are not bidirectional. How does it apply to single-directional models?
2. Other than the theoretical improvement in complexity, how much is the improvement in reduction in latency, computation FLOPs, etc? Figure 3 only shows GPU memory for image classification. How does GPU memory change on language tasks?

---

> ### Author Response · Authors · 2024-11-17
> **Authors Rebuttal to Reviewer z393 (1/2)**
>
> We thank the reviewer for their comments. Below, we have addressed the points raised in their feedback:
>
> ---
>
> **1. Some equations/notations are not clear enough.**
>
> We thank the reviewer for the helpful pointer. We have clarified in the text that in Eq. 6, $a_1/a_2$ are without the mask, and in Eq. 7, $b_1/b_2$ correspond to the masked attention. Additionally, $\lambda_i$ is the scalar version of $\Lambda_i$, as described in Eq. 5a. We have added the exact definition of $\lambda_i$ to the paper and further clarified Equations 6 and 7.
>
> ---
>
> **LION-Lit performs worse than BERT or ViT, and the gap is not negligible.**
>
> Thank you for the important notice. As outlined in Equations 20-25 and lines 336-339, LION-LIT is the bidirectional sequence modeling equivalent of the Linear Transformer.
> LION-LIT is formulated as *Scale*($QK^\top$), whereas models like ViT and BERT are formulated with *Softmax*($QK^\top$). Thus, LION-LIT is equivalent to linear attention but does not apply softmax, as  it prevents the the attention matrix from being linearized, as discussed in Theorem 3.3 of our paper (lines 145-154) and in more detail in Appendix B.3. It was observed in autoregressive and unidirectional models [1] that Linear Attention can be less performant than softmax-based attention. However, Linear Attention offers greater efficiency, as it can be expressed in an RNN format, enabling faster inference and lower memory usage [1,2]. To address the reviewer's concern, we have carried out additional experiments on the GLUE benchmark with BERT-large models, which indicate that the gap between LION-LIT and BERT is becoming smaller by increasing the model dimension to LARGE from Base as bellow:
>
>
> | Model   (LARGE)              | MLM Acc. | MNLI   | RTE   | QQP   | QNLI  | SST2  | STSB  | MRPC  | COLA  | Avg.   |
> |-----------------------|----------|--------|-------|-------|-------|-------|-------|-------|-------|--------|
> | BERT | **69.88** | **85.68** | **67.44** | **89.90** | **91.89** | **93.04** | **88.63** | **90.89** | *56.14* | **82.95** |
> | LION-LIT | 67.11 | 83.73 | 57.18 | *89.85* | 89.93 | 91.86 | *88.02* | 90.18 | 55.36 | 80.76 |
> | LION-S | *69.16* | *84.38* | *57.69* | 89.57 | *90.30* | *92.93* | 87.68 | *90.57* | **59.54** | *81.58* |
>
> ---
>
> **3. The advantage of adding selectivity is not well explained. Table 2 only shows LION-S results, but it does not compare it with LION.**
>
> Thank you for the suggestion. We have now included the LION-LIT results (which is a bi-directional version of Linear Transformer), which **indeed support our claim of improvement by using the selectivity in LION-S**. As observed, the Linear Attention variants of linear recurrent models do not perform as well as selective ones, such as Mamba-2 \[2\] or Gated-RFA \[3\]. By incorporating selectivity, we introduce LION-S.  The results of LION-LIT are as follows:
>
> |Model|ListOps|Text|Retrieval|Image|Pathfinder|PathX|Avg.|
> |-|-|-|-|-|-|-|-
> |LION-LIT|16.78|65.21|54.00|43.29|72.78|✘|50.41|
> |LION-S|62.25|88.10|90.35|86.14|91.30|97.99|86.07|
>
> The LION-LIT results are presented in revised Table 2, clearly demonstrating the effectiveness of the selective parameters and their choices.
>
> ---
>
> **4. The definitions of the architectural elements of figure 1 should be added to its caption.**
>
> Thank you for the helpful pointer. The definitions have been added to the caption in our revised paper.
>
> ---
>
> **5. Is this method limited to bidirectional models? Most generative models are not bidirectional. How does it apply to single-directional models?**
>
> We have already evaluated our selective parameter choice (LION-S 1D) in Appendix A.1 for the single-directional scenario of generative language modeling (in our first and latest submission), showing significant improvement over Linear Attention, with performance close to GPT-2 [4] in terms of perplexity (PPL). Additionally, since many variants of Transformers and SSMs have already been explored for autoregressive tasks (as shown in Table 5 of the Appendix), we believe that extending these models to their bidirectional alternatives is the key missing piece, which constitutes the main contribution of our paper. We will repost it here:
>
> | Model            | Perplexity (PPL)           |
> |------------------|-----------------------|
> | GPT-2            | **17.42**  |
> | LinAtt   (LION-Lit 1D)        | 21.07  |
> | **LIONs (1D)**    | *18.16*  |
>
> As seen, LION-S significantly outperforms Linear Attention and achieves performance comparable to GPT-2.

---

> ### Author Response · Authors · 2024-11-17
> **Authors Rebuttal to Reviewer z393 (2/2)**
>
> **6. Other than the theoretical improvement in complexity, how much is the improvement in reduction in latency, computation FLOPs, etc? Figure 3 only shows GPU memory for image classification. How does GPU memory change on language tasks?**
>
> Thank you for the question. In Appendix A.1, Figure 5 we have shown the latency and GPU memory of different gerneration modes on causal language modelling task. We also added the memory usage graph for non-causal language task in Figure 3. For number of FLOPs theorethical calculations are included in Appendix D.9. The calculations show that while transformer requires $O(L^2+Ld^2)$ FLOPS, LION-S needs only $O(Ld^2)$ with $d$ being the model dimension and $L$ the sequence length. The same calculations apply to the non-causal language task where $L$ corresponds to context length.
>
> ---
>
> **References**
>
> \[1\]: Transformers are RNNs: Fast Autoregressive Transformers with Linear Attention. Angelos Katharopoulos, Apoorv Vyas, Nikolaos Pappas, François Fleuret. arXiv:2006.16236, 2020.
>
> \[2\]: Transformers are SSMs: Generalized Models and Efficient Algorithms Through Structured State Space Duality. Tri Dao, Albert Gu. arXiv:2405.21060, 2024.
>
> \[3\]: Random Feature Attention. Hao Peng, Nikolaos Pappas, Dani Yogatama, Roy Schwartz, Noah A. Smith, Lingpeng Kong. arXiv:2103.02143, 2021.
>
> \[4\]: Language Models are Unsupervised Multitask Learners. Alec Radford, Jeff Wu, Rewon Child, David Luan, Dario Amodei, Ilya Sutskever. 2019.
>
> ---
>
> We again extend our appreciation for the reviewer's comments and concerns and hope that our efforts to address these concerns have been effective. In case the reviewer has any remaining concerns, we are happy to address them.

---

> > ### Author Response · Authors · 2024-11-24
> > **Deadline for author-reviewer discussion approaching**
> >
> > We would like to gently remind you that the **author-reviewer discussion period for our manuscript is approaching.**
> >
> > We appreciate your valuable contributions. **We believe that our responses address the concerns raised.** If you have any further suggestions or concerns, please let us know.
> >
> > Best,
> >
> > Authors

---

> ### Author Response · Authors · 2024-12-02
> **Final Day of Discussion Period for ICLR 2025**
>
> ---
>
> Dear Reviewer z393,
>
> We hope you are doing well. As today **marks the last day of the extended discussion period for ICLR 2025**, we would like to kindly request your final feedback on our submission. We sincerely appreciate your thoughtful comments, and we have made every effort to address your concerns in our rebuttal.
>
> If there are any remaining questions or clarifications needed, we would be more than happy to respond. **We would greatly appreciate it if you could review the rebuttal and provide your final thoughts by the end of the discussion period.**
>
> Additionally, if you feel that we have adequately addressed all concerns, **we would greatly appreciate it if you could consider increasing the score accordingly.**
>
> Best regards,
>
> Authors
>
> ---

---

### Official Review · Reviewer_tj5X · 2024-11-04

**Soundness:** 3
**Presentation:** 4
**Contribution:** 2
**Rating:** 3
**Confidence:** 5

**Summary:**

The paper presents *LION*, a bidirectional sequence modeling framework based on State Space-like Models (SSMs) such as Linear Attention and Mamba. Supplementing the standard parallel vectorized implementation, LION enables output computation through two recurrences, designed to preserve output fidelity with the parallelized approach. The main contributions are as follows:

1. Design linear recurrences that correspond to Linear Attention (LA) aka LION-LIT
2. Add Mamba's selectivity to those recurrences aka LION-S.
3. Empirically test the model on the LRA task, GLUE, and image classification
4. The resulting model does not require position embeddings and can be extrapolated beyond the training length.

**Strengths:**

1. The paper is very well written and quite illustrative which makes it an easy read.
2. The goals and the methodology proposed in the paper are easy to understand.
3. The paper shows that bidirectional linear attention (and its selective variant) can be written in a recurrent form.

**Weaknesses:**

**1: Recurrence Form of Linear Attention is a straightforward extension**
The claim that Linear Attention (LA) can be expressed as a recurrence seems like a straightforward calculation, given it is already known that causal LA can be written as a recurrence. To extend this to bidirectional settings, the numerator is simply a forward recurrence, $\sum_{j \leq i} q_i^Tk_j$, combined with a backward recurrence, $\sum_{j \geq i} q_i^Tk_j$, and a correction term $-q_i^Tk_j$ to account for double-counting. The same approach works with the denominator. While this is mathematically valid, the recurrence structure derived here seems like a straightforward extension of known results.


**2: LION's Continuous-Domain Selectivity is derived in Mamba**

The continuous selectivity mechanism in LION seems to be derived in Mamba’s formulation (see Eq. (4) in [1]). Mamba originally derived this expression into the form used here and then simplified it, i.e. they start with $\mathbf{\bar{B}} = (\Delta \mathbf{A})^{-1} (\exp(\Delta \mathbf{A}) - \mathbf{I}) \Delta \mathbf{B}$, and drop $(\Delta \mathbf{A})^{-1} (\exp(\Delta \mathbf{A}) - \mathbf{I})$ to simplify the term and notice no drop in performance. LION chooses the $\mathbf{A}$ matrix as identity which reduces the above expression to LION's (27a)


**3: No Position Encodings required is a consequence of using SSMs**
The claim that this model achieves independence from position encoding and can hence be extrapolated to longer sequences is a standard characteristic of all State Space Models, which implicitly capture positional information through recurrence. This has been recognized and documented within the community (see [GitHub issue #29](https://github.com/state-spaces/mamba/issues/29)). I feel that is not a contribution of LION but SSMs at large.

**4: Practicality of Recurrent Inference**

I am unsure where the recurrence form in the paper may have practical utility. For language modeling using recurrence makes imminent sense because it helps in streaming out the output data to the user in real time. However, in case of bidirectional modeling the parallel implementation already achieves $O(L)$ space and time complexity which is the same as LION

The only limited application I can see this being used is if the memory constraints are such that we have $O(L)$ memory but the constant is probably not big enough, then instead of computing it in a purely recurrent fashion which would be very slow, it is standard practice in SSMs to processes input in chunks, using:
  - **Input:** [SSM's State, subset of contiguous tokens]
  - **Output:** [Updated SSM's State, corresponding output tokens]

 Each chunk can be computed using the parallel form under available memory without needing to fully unroll the recurrence which would be very slow on GPUs. I would like to note that the above approach is well-established in the SSM community.

**5: Experimental Validity Concerns**

- **GLUE Scores:** The reported GLUE scores seem low; prior works using selective recurrences and adding them without correction factors at 70M parameters has been shown to yield a GLUE score of 80.6 which is higher than LION's 78.5. (See [2])

- **Vision Task Comparisons:** In the vision tasks, the authors only compare their model with ViT-small, which was originally designed for distillation purposes. Training a 22M model directly for vision tasks is impractical; comparisons with a larger model, such as ViT-base (86M parameters), would be more convincing and relevant.

[2]: Hydra: Bidirectional State Space Models Through Generalized Matrix Mixers. Sukjun Hwang, Aakash Lahoti, Tri Dao, Albert Gu

**Questions:**

N/A - see weaknesses

---

> ### Author Response · Authors · 2024-11-17
> **Authors Rebuttal to Reviewer tj5X (1/3)**
>
> We thank the reviewer for their feedback. Below, we have addressed the concerns raised and provided clarifications that we believe may help resolve any misunderstandings and better highlight the contribution of our work.
>
> ---
>
> **1: Recurrence Form of Linear Attention is a straightforward extension**
>
> Thanks for the comment. We would like to stress that the reviewer might have missed key details, causing a misevaluation. For this purpose, we would like to clarify:
> - While our approach may seem straightforward in retrospect, it is not an obvious extension, as it has not been formally addressed before. Theorem 3.3 and the proof in Section 3 provide key insights into expanding matrices to achieve bidirectional equivalence to the RNN format.
> - We consider the straightforward nature of our approach as an advantage, making it easier to understand. However, below we show that the naive extension of Linear Attention (which we believe what the reviewer has in mind) has significantly higher memory requirements than LION, rendering LION a non-obvious extension.
> - In short, compared to the naive extension (*cf.*, below), **LION scales linearly with the model dimension $d$**, whereas the **naive extension** and other alternatives for bidirectionality (as shown in Table 1 of the revised PDF) exhibit memory demands that **scale with $d^2$**.
>
> **Naive extension vs LION's Linear Memory Requirement Relative to Model Dimension:**
>
> Given this review, we see that an important benefit of LION, regarding its memory requirement, particularly in comparison to the naive extension of Linear Transformers into a bidirectional setting, has not been sufficiently highlighted. The following formulation for Linear Attention, presented in the "Transformers are RNNs" section, serves as the basis for comparison:
>
> $S_i = S_{i-1} + k_i v_i^\top$
>
> $z_i = z_{i-1} + k_i$
>
> $y_i = \frac{q_i^\top S_i}{q_i^\top z_i}$
>
> A naive extension of this approach, accounting for both forward and backward states, would be:
>
> $S_i^{F/B} = S_{i-1}^{F/B} + k_i v_i^\top$
>
> $z_i^{F/B} = z_{i-1}^{F/B} + k_i$
>
> *OUTPUT*: $y_i = \frac{q_i^\top (S^F_i + S^B_i - k_i v_i^\top)}{q_i^\top (z^F_i + z^B_i - k_i)}$
>
> We demonstrate below that this formulation is highly inefficient in terms of memory.
>
> Since the forward and backward states for token $i$ are not available simultaneously (except for the middle token in even-length sequences), both the forward and backward hidden states $S^F_i$ and $S^B_i$ ($O(2d^2L)$), and both scaling states $z^F_i$ and $z^B_i$ ($O(2Ld)$). Additionally, the memory required to store the output of each token ($O(Ld)$) results in a total memory demand of $O(2d^2L + 3Ld)$.
>
> Even with the memory allocation strategy described in Appendix B.5 of our paper, the memory of **above naive extention still scales quadratically with $d$**, **which contradicts the core motivation of using bidirectional RNNs**. The goal of bidirectional RNNs is to store only one output of dimension $d$ for each token, updating the state without requiring quadratic memory demands of $O(Ld^2)$.
>
> To address this, we store the states $c^{F/B}_i$ as scalar values and $y^{F/B}_i$ as vectors of dimension $d$. This approach only requires $O(Ld + L)$ memory units, which is significantly more efficient than the naive extension and **scales linearly with $d$** (Also added to Table 1 of our paper).
>
> In retrospect, LION is easy to understand (i.e., perhaps straightforward but in a good sense) and yet is highly effective (but non-obvious). This is our main contribution (which is also extensible): extending existing autoregressive recurrent models, such as Linear Attention (LA) or RetNet \[1\], into a bidirectional setting, where the attention mechanism is theoretically equivalent to a bidirectional RNN. LION achieves low memory requirements during inference, scaling linearly with the model dimension $d$.
>
> ---
>
> **2: LION's Continuous-Domain Selectivity is derived in Mamba**
>
> Thank you for the comment. We would like to clarify that this discretization is a zero-order hold (ZOH) discretization of continuous-time dynamic systems \[2\], which is also used in Mamba. However, as mentioned in the [HIPPO](https://arxiv.org/pdf/2008.07669) paper on page 21, this discretization was derived much earlier (before Mamba) and has been well-established in signal processing community and can also be found in relevant textbooks [5].
>
> For better understanding of the reader, we have included an explicit mention of this fact along with the proof of this discretization for both continuous state-space dynamics and transformer dynamics in Equations 26 and 27 in the appendix B.6 of our paper for completeness.

---

> ### Author Response · Authors · 2024-11-17
> **Authors Rebuttal to Reviewer tj5X (2/3)**
>
> Similar to Mamba, we have chosen the selectivity of the proposed bidirectional model based on the connection to ZOH discretization, and our results demonstrate that it works for bidirectional sequence modeling as well. Our claim is not that we invented this approach, but rather that it is a traditional method for sampling continuous signals, which has been successfully used in models like Mamba and other SSMs for causal sequence modeling. We are therefore showing that using LION with this approach also works for well-known bidirectional sequence modeling tasks, such as vision and masked language modeling.
>
> ---
>
> **3: No Position Encodings required is a consequence of using SSMs**
>
> We would like to emphasize again that the primary contribution of LION is extending causal Linear Transformers and their selective variants into a bidirectional setting. In this setting, the model trains like a Transformer using attention and infers like an RNN/SSM. As a result, LION leverages the advantages of SSMs/RNNs during inference, such as the elimination of the need for positional embeddings. While we do not claim this as the novelty of our work, we demonstrate that because LION is equivalent to an RNN during inference, it inherently possesses these advantages.
>
> LION train using attention parallelization (which is faster than scan algorithm, please see our response to question 5) while benefiting from the memory and computational efficiency of RNNs/SSMs during inference. Our experimental results provide evidence that, if the model is equivalent to an RNN during inference, it should inherit the advantages of RNNs/SSMs, such as not requiring positional embeddings and the ability to extrapolate.
>
> We demonstrate that LION and its selective variant LION-S (inspired by Mamba) indeed exhibit these features. Aside from positional embeddings, details about efficiency of LION are also provided at our response to concern 3 of reviewer sped.
>
> ---
>
> **4: Practicality of Recurrent Inference**
>
> Thanks for the comment. We would like to clarify that techniques like chunking can also be applied to LION. Since LION uses a scalar-valued state parameter, it is $d$ times more memory-efficient than Mamba and comparable to Mamba-2 with a scalar value $a_i$. This efficiency allows LION to achieve $O(Ld)$ memory requirements without needing chunking. However, chunking can still be applied for parallelization and further acceleration during inference when memory demands are high.
>
> As you can see, LION enables Linear Transformers to be extended into a bidirectional setting, *while retaining all the advantages of linear attention during inference*, such as chunk parallelization, and allowing for parallelized training using attention.
>
> The key advantage of LION over other SSMs that use parallel scan is that LION leverages attention parallelism during training, which is significantly faster than parallel scan, as shown in the table in our response to Question 5 of your review. This results in LION being much faster to train compared to parallel scan-based approaches.
>
>
> ---

---

> ### Author Response · Authors · 2024-11-17
> **Authors Rebuttal to Reviewer tj5X (3/3)**
>
> **5: Experimental Validity Concerns**
>
> * **GLUE Scores**: In our initial results we had employed the original [pretraining](https://github.com/HazyResearch/m2/blob/main/bert/yamls/pretrain/hf-transformer-pretrain-bert-base-uncased.yaml) and [finetuning](https://github.com/HazyResearch/m2/blob/main/bert/yamls/finetune-glue/hf-transformer-finetune-glue-bert-base-uncased.yaml) recipes from the **M2-BERT repository**. We have updated our results with the [pretraining](https://github.com/HazyResearch/m2/blob/main/bert/yamls/pretrain/monarch-mixer-pretrain-960dim-parameter-matched.yaml) and [finetuning](https://github.com/HazyResearch/m2/blob/main/bert/yamls/finetune-glue/monarch-mixer-finetune-glue-960dim-parameter-matched.yaml) recipes employed in the M2 models for the parameter-matching BASE and LARGE sizes. **We would like to highlight that this repository differs from Hydra's recipe, which is based on [MosaicBERT](https://mosaicbert.github.io/).**
>     * It is worth noting that these hyperparameters are highly tuned for the M2 model family. For the LARGE models, we even had to reduce the learning rate due to divergence during training, see Appendices D.4-D.5.
>     * Similarly, **Hydra [3] tunes the hyperparameters specifically for their architecture by exploring different learning rates, weight decays and number of epochs** for each GLUE task.
>     * Our computational resources do not allow for such expensive tuning. In our new results, we can observe that both the BERT and LION models obtain respectively a GLUE score of $82.25$ and $80.31$ for the BASE (110M) size and $82.95$ and $81.58$ for the LARGE (340M) size. For additional results, please check Table 3 and Appendices D.4-D.5. The BERT large results are:
>     |Model|MLM Acc.|MNLI|RTE|QQP|QNLI|SST2|STSB|MRPC|COLA|Avg.|
>     |-|-|-|-|-|-|-|-|-|-|-|
>     |BERT|**69.88**|**85.68**|**67.44**|**89.9**|**91.89**|**93.04**|**88.63**|**90.89**|*56.14*|**82.95**|
>     |LION-LIT|67.11|83.73|57.18|*89.85*|89.93|91.86|*88.02*|90.18|55.36|80.76|
>     |LION-S|*69.16*|*84.38*|*57.69*|89.57|*90.30*|*92.93*|87.68|*90.57*|**59.54**|*81.58*|
>
> This demonstrates how closely LION-S aligns with the original BERT, with the gap smaller in the larger model hidden size ($d$), while LION-S offers significant inference speed benefits.
>
> * **Vision Task Comparison**:
>     * We agree with the reviewer that larger scale models would strengthen our paper. Since pre-training is a compute intensive task, our experiments focus on smaller scales. After the initial submission, we expanded our experiments from 5.5M (Vit-T) to 21.7M (ViT-S). In the below table and in the updated Table 4 of our paper, you can find the results with larger models. Our results show that as the model size increases, the gap between LION-S and ViT gets less and less significant from 2.28% to 1.33%.
>     |Model|ImageNet|
>     |-|-|
>     |ViT-S|**72.19**|
>     |**LION-LIT** (SMALL) |69.62|
>     |**LION-S** (SMALL)|_70.86_|
>     * To further improve the performance of LION architecture, we introduced LION-S (v2) that changes the order of patches to capture spatial information. For details of the implementation, please see the Appendix C.8.
>     * Additionally, we have tried distillation to tiny LION-S model using DeiT framework [4] and we have observed that distilled LION-S can outperforms ViT-T. We believe that this result further proves the capabilities of the LION architecture.
>     |Model|ImageNet|
>     |-|-|
>     |ViT-T|70.23|
>     |**LION-S** (TINY)|67.95|
>     |**LION-S (v2)** (TINY)|69.22|
>     |**LION-S-Distil** (TINY) |**70.44**|
>     * Moreover, we have added the training times for each model using the CIFAR100 data, highlighting that training with attention (like ViT & LION) is **significantly faster** than using parallel scan. This underscores the motivation for training like a Transformer while inferring like an RNN.
>     |Training Strategy (Model)|Time (s)/Epoch|
>     |-|-|
>     |Attention (VIT) |24.6|
>     |Attention (**LION-S**) |35.8|
>     |Attention (**LION-LIT**)|26.6|
>     |Parallel Scan (Hydra) |43.4|
> ---
> **References**
>
> [1]: Retentive Network: A Successor to Transformer for Large Language Models. Yutao Sun, Li Dong, Shaohan Huang, Shuming Ma, Yuqing Xia, Jilong Xue, Jianyong Wang, Furu Wei. We
>
> [2]: Rudolph Emil Kalman. A new approach to linear filtering and prediction problems. 1960.
>
> [3]: Hydra: Bidirectional State Space Models Through Generalized Matrix Mixers. Sukjun Hwang, Aakash Lahoti, Tri Dao, Albert Gu
>
> [4]: Training Data-Efficient Image Transformers & Distillation Through Attention. Hugo Touvron, Matthieu Cord, Matthijs Douze, Francisco Massa, Alexandre Sablayrolles, Hervé Jégou. arXiv:2012.12877, 2021.
>
> [5]: Raymond A DeCarlo. Linear systems: A state variable approach with numerical implementation. Prentice-Hall, Inc., 1989.
>
> ---
> We hope that through our evaluations, we have been able to address the concerns raised by the reviewer. In case the reviewer has any remaining concerns, we are happy to address them.

---

> ### Author Response · Authors · 2024-11-24
> **Deadline for author-reviewer discussion approaching**
>
> We would like to gently remind you that the **author-reviewer discussion period for our manuscript is approaching.**
>
> We appreciate your valuable contributions. **We believe that our responses address the concerns raised.** If you have any further suggestions or concerns, please let us know.
>
> Best,
>
> Authors

---

> ### Author Response · Authors · 2024-12-02
> **Final Day of Discussion Period for ICLR 2025**
>
> ---
>
> Dear Reviewer tj5X,
>
> We hope you are doing well. As today **marks the last day of the extended discussion period for ICLR 2025**, we would like to kindly request your final feedback on our submission. We sincerely appreciate your thoughtful comments, and we have made every effort to address your concerns in our rebuttal.
>
> If there are any remaining questions or clarifications needed, we would be more than happy to respond. **We would greatly appreciate it if you could review the rebuttal and provide your final thoughts by the end of the discussion period.**
>
> Additionally, if you feel that we have adequately addressed all concerns, **we would greatly appreciate it if you could consider increasing the score accordingly.**
>
> Best regards,
>
> Authors
>
> ---

---

### Author Response · Authors · 2024-11-17
**General Response to All Reviewers**

We sincerely thank all the reviewers for their valuable feedback, which has helped improve our manuscript. Based on your suggestions, we have clarified several points about our approach and made the following updates in our proofs and experiments:

### Clarifications:
* **LION** is a framework that trains like transformers with **full linear attention** including their selective variants while inferring as **bi-directional RNNs**.
* **LION**'s memory requirements **scale linearly with the model hidden dimension $d$**, in contrast to the naive extension, which scales quadratically with $d^2$ during inference. (Detailed in our response to question 1 of Reviewer tj5X)
* **LION is not simply a (bidirectional) SSM**. LION has key differences with SSMs, including:
    1)  **LION** uses attention matrix parallelization during training, whereas SSMs use the parallel scan algorithm.
    2) **LION** is presented as RNNs with matrix-valued hidden states $S^{F/B}_i \in \mathbb{R}^{d \times d}$ during inference, while SSMs have vector-valued hidden states $h^{F/B}_i \in \mathbb{R}^{d}$.
    3) **LION** uses multi-head attention, while SSMs employ state expansion.
### New Empirical Evaluations and Theoretical Proofs:
* **Masked Language Modeling (MLM)**:
  - We changed the recipe for training (see Table 3 in the main manuscript). And LION-S has achived 81.58% of GLUE score, **which shrinked the gap with BERT model from 2.97% to 1.37%**. Fine-tuning recipes for all baselines and LION-S are provided in Table 8.
  - **Inference GPU memory analysis** for BERT vs LION-S is now included in Figure 3, which indicates that **LIOS-S scales linearly while BERT scales quadratically with sequence length**.
  - We added extensive additional **details on MLM tasks** in Appendix D.5.
  - Extensive ablations  for **LION-RetNet** and **LION-S** have been added in Section D.6 of the Appendix for both scales.

* **Vision Experiments**:
  - A **new version of LION-S (v2)**, with changes to patch orders, is presented in Table 4 and detailed in Appendix C.8. Which significanlty improves the performance of **LION-S achieving a top1 accuracy improvement of 1.27% on imagenet** (and larger on other datasets).
  - We experimented with **larger vision baselines and LION-S** and updated results in Table 4, which shows that LION-S (v1) performance is **70.86% on top-1 accuracy** on imagenet decreasing the performance gap against ViT from 2.28% to 1.33%.
  - **New Distilled LION-S following the DeiT recipe**, LION-S Distilled **outperforms VIT-T on imagenet by 0.2% top1 accuracy** (results in Table 14).
  - Number of **FLOPs** calculations is added in Appendix D.9, which proves that **LION-S scaled linearly with sequence length.**
  - **Training times** for all models is added Appendix D.11, which **higlights the benefit of LION training** over parallel scan algorithm.

* **LRA**:
  - **LION-LIT** results are added to Table 2, which **signifies the advantages of LION-S** by adding the selectivity.
  - The latest version of **S5** is now included in Table 2 as a baseline.

* **Theory**:
    - The proof connecting **LION-S**'s discretization from continuous time system with the **ZOH filtering** (presented in section 4 of paper) has been added in Appendix B.6.

---

All new material added to the **revised manuscript** has been highlighted in **blue text** for better visibility.


Best Regards,

Authors

---

### Author Response · Authors · 2024-11-21
**Gentle Reminder to all Reviewers**

Dear Reviewers,

As the discussion period comes to a close, we want to express our heartfelt gratitude for your time, effort, and thoughtful feedback. We are pleased to share that the latest version of LION-S (v2) on ImageNet has achieved a significant improvement, **outperforming ViT-Small by 1.25%** and achieving the highest accuracy of **73.44% among all baselines**.

We believe we have thoroughly addressed your questions and concerns with the following updates:

- The scale of our experiments $\Rightarrow$ Larger models in Tables 3 and 4.
- Theoretical connections of LION-S to zero-order hold discretization $\Rightarrow$ Appendix B.6.
- New experimental results for vision tasks (e.g., LION-S v2 detailed in Appendix C.8) $\Rightarrow$ Improvement over ViT-Small in Table 4.
- Clarification of the structure of LION and how it differs from SSMs $\Rightarrow$ [General response](https://openreview.net/forum?id=VxJjw52dZu&noteId=L6tZTXH1vI)
- More efficiency comparisons both theoreticaly and empiricaly $\Rightarrow$ Figure 3 and Appendix D.9.
- Adding LION-LIT on the LRA task $\Rightarrow$ Table 2.

We hope these updates and clarifications provide a comprehensive understanding of our work and highlight its significance. If there are any remaining questions or points requiring further clarification, we are happy to address them promptly before the discussion period concludes.

---

### Meta-Review · Area_Chair_tRJt · 2024-12-23

**Metareview:**

This paper proposes a new framework to train bi-directional Transformers to achieve memory-efficient inference through recurrence. While the experimental results are solid, and the method is supported by theories, recurrent inference of Transformers might not significantly impact the current practice. Practitioners (me included) prefer utilizing the parallelism of GPUs to do inference for bidirectional Transformers in parallel to achieve a higher speed than recurrence. Inference speed is critical in practice. Therefore, unless the method can reduce memory cost while maintaining the speed and accuracy, I tend to reject this paper.

**Additional Comments On Reviewer Discussion:**

Reviewers have concerns about the novelty and practical value of this work. Some of the novelty issues are not the main claims of the work, but I do agree that making bi-directional Transformers inference sequentially like RNNs seems useless in practice.

---

### Decision · Program_Chairs · 2025-01-22

Reject